# Exotic $U(1)$ Symmetries, Duality, and Fractons in $3+1$-Dimensional Quantum Field Theory

Nathan Seiberg and Shu-Heng Shao

*School of Natural Sciences, Institute for Advanced Study,*
*Princeton, NJ 08540, USA*

## Abstract

We extend our exploration of nonstandard continuum quantum field theories in $2+1$ dimensions to $3+1$ dimensions. These theories exhibit exotic global symmetries, a peculiar spectrum of charged states, unusual gauge symmetries, and surprising dualities. Many of the systems we study have a known lattice construction. In particular, one of them is a known gapless fracton model. The novelty here is in their continuum field theory description. In this paper, we focus on models with a global $U(1)$ symmetry and in a followup paper we will study models with a global $\mathbb{Z}_N$ symmetry.

# 1 Introduction

Common lore states that the low-energy behavior of every lattice system can be described by a continuum quantum field theory. However, some recently found lattice constructions, including theories of fractons (for reviews, see e.g. [1, 2] and references therein), violate this lore.

Our study was motivated by the question: how can the framework of continuum quantum field theory accommodate these examples?

This paper is the second in a series of three papers addressing this question. The first paper [3] focused on models in $2 + 1$ dimensions, while this paper and [4] study $3 + 1$-dimensional systems. Here we limit ourselves to system whose global symmetry is continuous, and in particular $U(1)$, while [4] will discuss systems based on $\mathbb{Z}_N$. (A followup paper [5] explores additional models.)

Our discussion here (and in [3, 4]) uses a number of new ingredients:

- Not only are these quantum fields theories not Lorentz invariant, they are also not rotational invariant. In [3], the 2+1-dimensional systems preserve only the $\mathbb{Z}_4$ subgroup of the $SO(2)$ rotation group, while here and in [4] only the $S_4$ subgroup of the $SO(3)$ rotations is preserved. $S_4$ is the cubic group generated by 90 degree rotations.

- We continue the investigation of [6, 3], emphasizing the global symmetries of these systems. As always, the discussion of the symmetries is more general than the specific models. The symmetries here are not the usual global symmetries; we refer to them as exotic global symmetries. We also gauge these global symmetries.

- Perhaps the most significant new element is that we consider discontinuous fields. The underlying spacetime is continuous, but we allow discontinuous field configurations. Starting at short distances with a lattice, all the fields are discontinuous there. In standard systems, the fields in the low-energy description are continuous. Here, they are more continuous than at short distances, but some discontinuities remain.

Throughout this paper we will consider only flat spacetime. Space will be either $\mathbb{R}^3$ or a rectangular three-torus $\mathbb{T}^3$. The signature will be either Lorentzian or Euclidean. And when it is Euclidean we will also consider the case of a rectangular four-torus $\mathbb{T}^4$. We will use $x^i$ with $i = 1, 2, 3$ to denote the three spatial coordinates, $x^0$ for Lorentzian time, and $\tau$ for Euclidean time. The spatial vector index $i$ can be freely raised and lowered. When specializing to a particular component of an expression, we will also use $(t, x, y, z)$ to denote the coordinates with $t \equiv x^0, x \equiv x^1, y \equiv x^2, z \equiv x^3$. When we consider tensors, e.g. $A_{ij}$, we will denote specific components as $A_{xy}$, etc.

When space is a three-torus, the lengths of its three sides will be denoted as $\ell^i$ (or explicitly, $\ell^x$, $\ell^y$, $\ell^z$). When we take an underlying lattice into account the number of sites in the three directions are $L^i = \frac{\ell^i}{a}$ (or explicitly, $L^x$, $L^y$, $L^z$).

*Summary of [3]*

Since this paper is a continuation of [3], we will simply review its main results here and refer the interested reader to [3] for the details.

Most of the discussion in [3] focused on the XY-plaquette model [7], whose $2 + 1$-dimensional continuum Lagrangian is [7–13] (related Lagrangians appeared in [14–16])

$$\mathcal{L} = \frac{\mu_0}{2}(\partial_0\phi)^2 - \frac{1}{2\mu}(\partial_x\partial_y\phi)^2$$
$$\phi \sim \phi + 2\pi \, . \tag{1.1}$$

A key fact about the model (1.1) is that the dispersion relation is

$$\omega^2 = \frac{1}{\mu_0\mu}(k_x k_y)^2 \, . \tag{1.2}$$

This means that the low-energy theory includes modes with arbitrarily large $k_x$, provided $k_y$ is small enough. Similarly, it includes modes with arbitrarily large $k_y$, provided $k_x$ is small enough. This is an intriguing UV/IR mixing and it underlies many of the peculiarities of the system.

This model has two dipole global symmetries [3]. They are subsystem symmetries; i.e. they act separately at fixed $x$ or separately at fixed $y$. We referred to these two different symmetries as momentum and winding symmetries. The model and its symmetries are summarized in Table 1.

An essential part of the analysis was the use of discontinuous field configurations. Clearly, we must consider discontinuous fields whose action is finite. More interestingly, we also entertained some discontinuous fields, whose action diverges.[1] For that we had in mind a lattice with lattice spacing $a$. This turns out to be meaningful because these field configurations carry a conserved charge and they lead to the lowest energy states carrying this charge.

Our analysis in [3] concluded that all the states carrying momentum and winding charges have energies of order $\frac{1}{a}$. A conservative approach simply discards them. Yet, we found it interesting to explore their properties as they follow the Lagrangian (1.1). We did emphasize

---

[1]It is well known that the Euclidean path integral is dominated by discontinuous configurations with infinite action. We do not see a relation between this fact and the phenomena we study here.

| Lagrangian | $\frac{\mu_0}{2}(\partial_0\phi)^2 - \frac{1}{2\mu}(\partial_x\partial_y\phi)^2$ | $\frac{\widetilde{\mu}_0}{2}(\partial_0\phi^{xy})^2 - \frac{1}{2\widetilde{\mu}}(\partial_x\partial_y\phi^{xy})^2$ |
|---|---|---|
| dipole symmetry $(\mathbf{1}_0, \mathbf{1}_2)$ | momentum $(J_0 = \mu_0\partial_0\phi, \, J^{xy} = -\frac{1}{\mu}\partial^x\partial^y\phi)$ | winding $(J_0 = \frac{1}{2\pi}\partial_x\partial_y\phi^{xy}, \, J^{xy} = \frac{1}{2\pi}\partial_0\phi^{xy})$ |
| currents | $\partial_0 J_0 = \partial_x\partial_y J^{xy}$ | |
| charges | $Q^x(x) = \oint dy\, J_0 = \sum_\alpha N_\alpha^x \delta(x - x_\alpha)$ $Q^y(y) = \oint dx\, J_0 = \sum_\beta N_\beta^y \delta(y - y_\beta)$ $\oint dx\, Q^x(x) = \oint dy\, Q^y(y)$ | |
| energy | $\mathcal{O}(1/a)$ | |
| number of sectors | $L^x + L^y - 1$ | |
| dipole symmetry $(\mathbf{1}_2, \mathbf{1}_0)$ | winding $(J_0^{xy} = \frac{1}{2\pi}\partial^x\partial^y\phi, \, J = \frac{1}{2\pi}\partial_0\phi)$ | momentum $(J_0^{xy} = \widetilde{\mu}_0\partial_0\phi^{xy}, \, J = -\frac{1}{\widetilde{\mu}}\partial_x\partial_y\phi^{xy})$ |
| currents | $\partial_0 J_0^{xy} = \partial^x\partial^y J$ | |
| charges | $Q_x^{xy}(x) = \oint dy\, J_0^{xy} = \sum_\alpha W_\alpha^x \delta(x - x_\alpha)$ $Q_y^{xy}(y) = \oint dx\, J_0^{xy} = \sum_\beta W_\beta^y \delta(y - y_\beta)$ $\oint dx\, Q_x^{xy}(x) = \oint dy\, Q_y^{xy}(y)$ | |
| energy | $\mathcal{O}(1/a)$ | |
| number of sectors | $L^x + L^y - 1$ | |
| duality map | $\mu_0 = \frac{\widetilde{\mu}}{4\pi^2} \quad \mu = 4\pi^2\widetilde{\mu}_0$ | |

Table 1: Global symmetries and their charges in the $2+1$-dimensional scalar theories $\phi$ and $\phi^{xy}$. The energies of states that are charged under these global symmetries are of order $1/a$.

in [3] that this analysis is not universal and can be contaminated by certain higher derivative corrections to the minimal Lagrangian (1.1), but these corrections do not change the qualitative behavior.

The momentum and winding states have energy of order $\frac{1}{\ell a}$, with $\ell$ the physical size of the system. This means that if we take the large volume limit $\ell \to \infty$ before the continuum limit $a \to 0$, these states have zero energy. They correspond to different superselection sectors in this infinite volume limit. However, if we take the continuum limit $a \to 0$ at fixed volume (with or without taking later the large volume limit $\ell \to \infty$), then these states are heavy.

Surprisingly, the theory based on (1.1) is self-dual. The Lagrangian of the dual field $\phi^{xy}$ is

$$
\begin{aligned}
\mathcal{L} &= \frac{\widetilde{\mu}_0}{2}(\partial_0 \phi^{xy})^2 - \frac{1}{2\widetilde{\mu}}(\partial_x \partial_y \phi^{xy})^2 \\
&\phi^{xy} \sim \phi^{xy} + 2\pi \\
\widetilde{\mu}_0 &= \frac{\mu}{4\pi^2}, \qquad \widetilde{\mu} = 4\pi^2 \mu_0.
\end{aligned}
\tag{1.3}
$$

As in standard T-duality in $1+1$ dimensions, the role of the momentum and winding symmetries is exchanged by the duality. See Tables 1 for details.

Our earlier paper [3] also considered the gauge theory based on the global symmetry of (1.1). This gauge theory had been studied in [17, 8, 18, 19, 10, 11, 20]. (Related models were discussed in [21–28, 15, 29, 16, 30, 31, 13].) The gauge fields are $A_0$ and $A_{xy}$ with the gauge transformation

$$
\begin{aligned}
A_0 &\to A_0 + \partial_0 \alpha, \\
A_{xy} &\to A_{xy} + \partial_x \partial_y \alpha \\
\alpha &\sim \alpha + 2\pi.
\end{aligned}
\tag{1.4}
$$

There are no $A_{xx}, A_{yy}$ components. This theory has a gauge invariant electric field

$$
E_{xy} = \partial_0 A_{xy} - \partial_x \partial_y A_0
\tag{1.5}
$$

and no magnetic field. Its Lagrangian is

$$
\frac{1}{g_e^2} E_{xy}^2 + \frac{\theta}{2\pi} E_{xy}.
\tag{1.6}
$$

In many ways it is similar to an ordinary $U(1)$ gauge theory in $1+1$ dimensions. It has a $\theta$-parameter and no local excitations.

Its spectrum includes excitations with energy of order $g_e^2 \ell a$ with $a$ the lattice spacing and $\ell$ the physical size of the system. In the continuum limit, we take $a \to 0$ with fixed $\ell$. Then these states have zero energy. Alternatively, if we take the large volume limit $\ell \to \infty$ before

| $(2+1)d$ | Lagrangian | spectrum |
|---|---|---|
| scalar theory $\phi$ | $\frac{\mu_0}{2}(\partial_0\phi)^2 - \frac{1}{2\mu}(\partial_x\partial_y\phi)^2$ | gapless local excitations <br> charged states at order $\frac{1}{\mu\ell a}$, $\frac{1}{\mu_0\ell a}$ |
| $U(1)$ tensor gauge theory $A$ | $\frac{1}{g_e^2}E_{xy}^2 + \frac{\theta}{2\pi}E_{xy}$ | no local excitations – gapped <br> charged states at order $g_e^2\ell a$ |
| $\mathbb{Z}_N$ tensor gauge theory | $\frac{N}{2\pi}\phi^{xy}E_{xy}$ | no local excitations – gapped <br> large vacuum degeneracy |

Table 2: Spectra of the continuum field theories discussed in [3]. Depending on the order of limits $a \to 0$ or $\ell \to \infty$, the energy of the charged states goes to zero or infinity.

the continuum limit, they have infinite energy.

In [3] we also considered certain charged states with order $1/a$ different nonzero charges. Such states have energy of order one and the precise value of their energy can be contaminated by higher derivative corrections to the minimal Lagrangian (1.6).

A $\mathbb{Z}_N$ version of the tensor gauge theory was found by Higgsing the $U(1)$ gauge theory using a scalar field $\phi$ (as in (1.1)) with charge $N$. We dualized $\phi$ to $\phi^{xy}$ (as in (1.3)) to find a $BF$-type description

$$\frac{N}{2\pi}\phi^{xy}E_{xy} \tag{1.7}$$

of the $\mathbb{Z}_N$ tensor gauge theory.

The resulting theory turned out to be dual to a non-gauge theory of $\mathbb{Z}_N$ spins interacting around a plaquette [3]. These theories are known as Ising-plaquette theories and they had been studied extensively (see [32] for a review and references therein).

Just as its parent $U(1)$ theory is similar to an ordinary $U(1)$ gauge theory in $1+1$ dimensions, this theory is similar to an ordinary $\mathbb{Z}_N$ gauge theory in $1+1$ dimensions.

We summarize the theories studied in [3] and their spectra in Table 2.

*Outline*

The goal of this paper (and of the later paper [4]) is to extend the discussion in [3] to $3+1$ dimensions. Here we will focus on models with continuous global symmetries analogous to (1.1) and (1.4) and in [4] we will consider $\mathbb{Z}_N$ theories analogous to those of [3].

In Section 2, we will discuss the global symmetries of these systems. Unlike the $2+1$-dimensional systems of [3], here we will have more options for the representations of the spatial rotation group and they lead to several interesting exotic symmetries.

Section 3 will analyze the $3+1$-dimensional version of (1.1). We will refer to it as the $\phi$-theory. The discussion will be similar to that of the $2+1$-dimensional theory. The main difference between them is that the $3+1$-dimensional $\phi$-theory is not selfdual. As in $2+1$ dimensions, we will find momentum and winding states with energy of order $\frac{1}{a}$.

In Section 4, we will consider another non-gauge theory. We will refer to it as the $\hat{\phi}$-theory. This theory differs from the $\phi$-theory in two crucial ways. First, the dynamical field $\hat{\phi}$ is not invariant under rotations. It is in a two-dimensional representation of the cubic group (see Appendix A). Second, unlike the $\phi$-theory, its Lagrangian is second order in spatial derivatives. Again, we will find momentum and winding exotic symmetries and a rich spectrum of states charged under them. The momentum states have energy of order $\frac{1}{a}$ (as in the $\phi$-theory). But the winding states have energies of order $a$. This is unlike the case in the $\phi$-theory, where they are both at $\frac{1}{a}$, and it is also different from the winding states of an ordinary compact scalar whose energies are of order one.

In Sections 5 and 6, we will consider gauge theories associated with the global momentum symmetries of the $\phi$-theory (Section 3) and the $\hat{\phi}$-theory (Section 4), respectively. Therefore, we will denote the gauge fields by $A$ and $\hat{A}$, and we will refer to the theories as the $A$-theory and the $\hat{A}$-theory.

Certain aspects of the gauge theory of $A$ have been discussed in [17, 8, 18, 19, 9, 12] (see [21–28, 15, 29, 16, 30, 10, 11, 31, 13, 20] for related tensor gauge theories). The gauge theory of $\hat{A}$ is related to gauge theories discussed in [8, 12]. These two gauge theories have new exotic global symmetries, analogous to the electric and the magnetic generalized global symmetries of ordinary $U(1)$ gauge theories [33]. And they have subtle excitations carrying these global electric and magnetic charges.

We will show that the $A$-theory is dual to the $\hat{\phi}$-theory and the $\hat{A}$-theory is dual to the $\phi$-theory. In every one of these dual pairs the global symmetries and the spectra match across the duality. (See Table 3 and Table 4.) This is particularly surprising given the subtle nature of the states that are charged under the momentum and winding symmetries of the non-gauge systems and the subtle nature of the states that are charged under the magnetic and the electric symmetries of the gauge systems.

These two dual pairs of theories, $A/\hat{\phi}$ and $\hat{A}/\phi$, will be the building blocks of the $\mathbb{Z}_N$

| Lagrangian | $\frac{\hat{\mu}_0}{12}(\partial_0\hat{\phi}^{i(jk)})^2 - \frac{\hat{\mu}}{2}(\partial_k\hat{\phi}^{k(ij)})^2$ | $\frac{1}{2g_e^2}E_{ij}E^{ij} - \frac{1}{2g_m^2}B_{[ij]k}B^{[ij]k}$ |
|---|---|---|
| | | $E_{ij} = \partial_0 A_{ij} - \partial_i\partial_j A_0$ |
| | | $B_{[ij]k} = \partial_i A_{jk} - \partial_j A_{ik}$ |
| $(\mathbf{2}, \mathbf{3'})$ <br> tensor symmetry | momentum <br> $(J_0^{[ij]k} = \hat{\mu}_0\partial_0\hat{\phi}^{[ij]k}, J^{ij} = \hat{\mu}\partial_k\hat{\phi}^{k(ij)})$ | magnetic <br> $(J_0^{[ij]k} = \frac{1}{2\pi}B^{[ij]k}, J^{ij} = \frac{1}{2\pi}E^{ij})$ |
| currents | $\partial_0 J_0^{[ij]k} = \partial^i J^{jk} - \partial^j J^{ik}$ | |
| charges <br> (4.38) (5.60) | $Q^{[xy]}(z) = \oint dx \oint dy J_0^{[xy]z} = \sum_\gamma W_{z\gamma}\delta(z - z_\gamma)$ <br> $\oint dz Q^{[xy]} + \oint dx Q^{[yz]x} + \oint dy Q^{[zx]y} = 0$ | |
| energy (4.40) (5.61) | $\mathcal{O}(1/a)$ | |
| number of sectors | $L^x + L^y + L^z - 1$ | |
| $(\mathbf{3'}, \mathbf{2})$ <br> tensor symmetry | winding <br> $(J_0^{ij} = \frac{1}{2\pi}\partial_k\hat{\phi}^{k(ij)}, J^{k(ij)} = \frac{1}{2\pi}\partial_0\hat{\phi}^{k(ij)})$ | electric <br> $(J_0^{ij} = \frac{2}{g_e^2}E^{ij}, J^{[ki]j} = \frac{2}{g_m^2}B^{[ki]j})$ |
| currents | $\partial_0 J_0^{ij} = \partial_k(J^{[ki]j} + J^{[kj]i})$ <br> $\partial_i\partial_j J_0^{ij} = 0$ | |
| charges <br> (4.42) (5.45) | $Q^z(x, y) = \oint dz J_0^{xy} = W_z^x(x) + W_z^y(y)$ <br> $(W_z^x(x), W_z^y(y)) \sim (W_z^x(x) + 1, W_z^y(y) - 1)$ | |
| energy (4.45) (5.42) | $\mathcal{O}(a)$ | |
| number of sectors | $2L^x + 2L^y + 2L^z - 3$ | |
| duality map | $\hat{\mu}_0 = \frac{g_m^2}{8\pi^2} \qquad \hat{\mu} = \frac{g_e^2}{8\pi^2}$ | |

Table 3: Global symmetries of the $U(1)$ tensor gauge theory $A$ and its dual $\hat{\phi}$. Above we have only shown charges for some directions, while the others admit similar expressions.

| Lagrangian | $\frac{\mu_0}{2}(\partial_0\phi)^2 - \frac{1}{4\mu}(\partial_i\partial_j\phi)^2$ | $\frac{1}{2\hat{g}_e^2}\hat{E}_{ij}\hat{E}^{ij} - \frac{1}{\hat{g}_m^2}\hat{B}^2$ |
|---|---|---|
| | | $\hat{E}^{ij} = \partial_0\hat{A}^{ij} - \partial_k\hat{A}_0^{k(ij)}$ $\hat{B} = \frac{1}{2}\partial_i\partial_j\hat{A}^{ij}$ |
| $(\mathbf{1},\mathbf{3}')$ dipole symmetry | momentum $(J_0 = \mu_0\partial_0\phi, J^{ij} = -\frac{1}{\mu}\partial^i\partial^j\phi)$ | magnetic $(J_0 = \frac{1}{2\pi}\hat{B}, J^{ij} = \frac{1}{2\pi}\hat{E}^{ij})$ |
| currents | $\partial_0 J_0 = \frac{1}{2}\partial_i\partial_j J^{ij}$ | |
| charges (3.27) (6.59) | $Q_{xy}(z) = \oint dx \oint dy J_0 = \sum_\gamma W_{z\gamma}\delta(z-z_\gamma)$ $\oint dz Q_{xy}(z) = \oint dy Q_{zx}(y) = \oint dx Q_{yz}(x)$ | |
| energy (3.28) (6.60) | $\mathcal{O}(1/a)$ | |
| number of sectors | $L^x + L^y + L^z - 2$ | |
| $(\mathbf{3}',\mathbf{1})$ dipole symmetry | winding $(J_0^{ij} = \frac{1}{2\pi}\partial^i\partial^j\phi, J = \frac{1}{2\pi}\partial_0\phi)$ | electric $(J_0^{ij} = -\frac{2}{\hat{g}_e^2}\hat{E}^{ij}, J = \frac{2}{\hat{g}_m^2}\hat{B})$ |
| currents | $\partial_0 J_0^{ij} = \partial^i\partial^j J$ $\partial^i J_0^{jk} = \partial^j J_0^{ik}$ | |
| charges (3.30) (6.50) | $Q(\mathcal{C}_i^{xy}, z) = \oint_{\mathcal{C}_i^{xy}\in(x,y)}(dx J_0^{zx} + dy J_0^{zy}) = \sum_\gamma W_{i\gamma}^z\delta(z-z_\gamma)$ $\oint dz Q(\mathcal{C}_x^{xy}, z) = \oint dx Q(\mathcal{C}_z^{yz}, x)$ | |
| energy (3.32) (6.51) | $\mathcal{O}(1/a)$ | |
| number of sectors | $2L^x + 2L^y + 2L^z - 3$ | |
| duality map | $\mu_0 = \frac{\hat{g}_m^2}{8\pi^2}$ $\qquad$ $\frac{1}{\mu} = \frac{\hat{g}_e^2}{8\pi^2}$ | |

Table 4: Global symmetries of the $U(1)$ tensor gauge theory $\hat{A}$ and its dual $\phi$. Here $\mathcal{C}_i^{ij}$ is a curve on the $ij$ plane that wraps around the $i$ cycle once but not the $j$ cycle. Above we have only shown charges for some directions, while the others admit similar expressions.

tensor gauge theory in [4], which is the continuum field theory for the X-cube model [34]. More specifically, the $\mathbb{Z}_N$ continuum field theory can arise from Higgsing the $U(1)$ gauge group of $A$ by a charge $N$ matter field $\phi$, or from Higgsing the $U(1)$ gauge group of $\hat{A}$ by a charge $N$ matter field $\hat{\phi}$. The two descriptions are equivalent to each other at long distances.

Appendix A will review the representations of the cubic group and our notation.

## 2    Exotic $U(1)$ Global Symmetries

### 2.1    Ordinary $U(1)$ Global Symmetry and Vector Global Symmetry

Consider a $3 + 1$-dimensional quantum field theory with an ordinary $U(1)$ global symmetry that is associated with a Noether current $J_\mu$. The current conservation equation is

$$\partial^\mu J_\mu = 0 \,, \tag{2.1}$$

or in non-relativistic notation

$$\partial_0 J_0 = \partial^i J_i \,, \tag{2.2}$$

where $i = 1, 2, 3$ is a vector index of $SO(3)$.

This can be generalized to currents in other representations of the rotation group.

One example is the vector global symmetry whose currents are $(J_0^i, J^{ji})$ [6]. The $SO(3)$ representations for the time and space components of the currents are $\mathbf{R}_{\text{time}} = \mathbf{3}$ and $\mathbf{R}_{\text{space}} = \mathbf{1} \oplus \mathbf{3} \oplus \mathbf{5}$, respectively. The current obeys the conservation equation

$$\partial_0 J_0^i = \partial_j J^{ji} \,. \tag{2.3}$$

The currents $(J_0^i, J^{ji})$ can be further restricted by an algebraic condition such as $J^{ij} = -J^{ji}$, so that $(\mathbf{R}_{\text{time}}, \mathbf{R}_{\text{space}}) = (\mathbf{3}, \mathbf{3})$. The conserved charge is

$$Q(\mathcal{C}) = \oint_{\mathcal{C}} n_i J_0^i \,, \tag{2.4}$$

where $\mathcal{C}$ is a closed two-dimensional spatial manifold and $n_i$ is the normal vector to $\mathcal{C}$. This is a non-relativistic one-form global symmetry [6]. If the currents further obey a differential condition

$$\partial_i J_0^i = 0 \,, \tag{2.5}$$

then the dependence of $Q(\mathcal{C})$ on $\mathcal{C}$ becomes topological. This is a relativistic one-form global symmetry [33].

Alternatively, we can restrict $\mathbf{R}_{\mathrm{space}}$ to a singlet $\mathbf{1}$, and the currents obey

$$\partial_0 J_0^i = \partial_i J . \tag{2.6}$$

The conserved charge is

$$Q(\mathcal{C}) = \oint_{\mathcal{C}} dx^i \, J_0^i \tag{2.7}$$

with $\mathcal{C}$ a closed one-dimensional spatial curve. An example realizing the $(\mathbf{R}_{\mathrm{time}}, \mathbf{R}_{\mathrm{space}}) = (\mathbf{3}, \mathbf{1})$ current is a compact boson $\Phi$ in the continuum, $\Phi \sim \Phi + 2\pi$. The current

$$J_0^i = \partial^i \Phi , \qquad J = \partial_0 \Phi \tag{2.8}$$

satisfies the conservation equation (2.6) trivially and the charge $Q(\mathcal{C}) = \oint_{\mathcal{C}} dx^i \partial_i \Phi$ is the winding charge. In this case the currents satisfy a differential condition

$$\partial^i J_0^j = \partial^j J_0^i , \tag{2.9}$$

making the dependence of $Q(\mathcal{C})$ on $\mathcal{C}$ topological.

In the following we will consider more general currents with $\mathbf{R}_{\mathrm{time}}$ in a tensor representation of $SO(3)$ or a subgroup thereof.

## 2.2  $U(1)$ Tensor Global Symmetry

Let the time component of the current be $J_0^I$, where the index $I$ is in the representation $\mathbf{R}_{\mathrm{time}}$ of the rotation group. Denote the spatial component of the current as $J^{iI}$. The currents obey a conservation equation

$$\partial_0 J_0^I = \partial_i J^{iI} . \tag{2.10}$$

We could impose further algebraic constraints on $J^{iI}$ so that it is in a representation $\mathbf{R}_{\mathrm{space}}$ of the rotation group. We will call the symmetry generated by the currents $(J_0^I, J^{iI})$ the $(\mathbf{R}_{\mathrm{time}}, \mathbf{R}_{\mathrm{space}})$ *tensor global symmetry.*

The global symmetry charge is obtained by integrating $J_0^I$ over the entire space

$$Q^I = \int_{\mathrm{space}} J_0^I , \tag{2.11}$$

or a closed subspace $\mathcal{C}$

$$Q(\mathcal{C}) = \oint_{\mathcal{C}} J_0 \tag{2.12}$$

where the index $I$ is contracted with the integral measure and is suppressed. The subspace

is chosen such that the charge is conserved

$$\partial_0 Q(\mathcal{C}) = \oint_{\mathcal{C}} \partial_0 J_0 = \oint_{\mathcal{C}} \partial_i J^i = 0 \,, \tag{2.13}$$

where again the index $I$ is contracted and is suppressed.

Often the time component of the current satisfies some differential condition (such as $\partial_i J_0^i = 0$). This can restrict the dependence on $\mathcal{C}$. Then, $Q(\mathcal{C})$ can be independent of certain changes in $\mathcal{C}$ or even be completely topological. Algebraically, this condition performs a quotient of the space of charges.

As an example, let us take the time component of the current $J_0^{(ij)}$ to be a symmetric tensor of $SO(3)$, i.e. $\mathbf{R}_{\text{time}} = \mathbf{1} \oplus \mathbf{5}$.[2] For the spatial component $J^{k(ij)}$, we impose an algebraic condition

$$J^{(kij)} = 0 \,, \tag{2.14}$$

to restrict its representation $\mathbf{R}_{\text{space}}$ to $\mathbf{3} \oplus \mathbf{5}$.[3]

The currents $(J_0^{(ij)}, J^{k(ij)})$ with $(\mathbf{R}_{\text{time}}, \mathbf{R}_{\text{space}}) = (\mathbf{1} \oplus \mathbf{5}, \mathbf{3} \oplus \mathbf{5})$ obey the conservation equation

$$\partial_0 J_0^{(ij)} = \partial_k J^{k(ij)} \,. \tag{2.16}$$

Using the algebraic equation (2.14) and the conservation law

$$G \equiv \partial_i \partial_j J_0^{(ij)} \tag{2.17}$$

is conserved

$$\partial_0 G = \partial_0 \partial_i \partial_j J_0^{(ij)} = \partial_i \partial_j \partial_k J^{k(ij)} = 0 \,. \tag{2.18}$$

In some applications we also set

$$G \equiv \partial_i \partial_j J_0^{(ij)} = 0 \,. \tag{2.19}$$

We will be particularly interested in a more general case where only the cubic symmetry $S_4$ subgroup of the full rotation symmetry $SO(3)$ is preserved. The vector representation $\mathbf{3}$

---

[2]In this discussion with the $SO(3)$ rotation symmetry, the vector indices $i, j, k$ can be the same. In other parts of the paper where the rotation group is the cubic group $S_4$, the indices $i, j, k$ are never equal, $i \neq j \neq k$.

[3]In general, a tensor $T^{i(jk)}$ is in $\mathbf{3} \otimes (\mathbf{1} \oplus \mathbf{5}) = \mathbf{3} \oplus \mathbf{3} \oplus \mathbf{5} \oplus \mathbf{7}$ of $SO(3)$. The algebraic condition sets the totally symmetric combination in $(\mathbf{3} \otimes \mathbf{3} \otimes \mathbf{3})^S = \mathbf{3} \oplus \mathbf{7}$ to zero, and then the current $J^{k(ij)}$ only includes $\mathbf{3} \oplus \mathbf{5}$. More explicitly, the $\mathbf{3}$ and $\mathbf{5}$ components of $J^{k(ij)}$ are

$$J^{(\mathbf{3})i} = \delta_{jk} J^{i(jk)} \,, \qquad J^{(\mathbf{5})i(jk)} = J^{i(jk)} - \frac{1}{3} \delta^{jk} J^{(\mathbf{3})i} \,. \tag{2.15}$$

of $SO(3)$ reduces to the standard representation $\mathbf{3}$ of $S_4$. On the other hand, the traceless, symmetric representation $\mathbf{5}$ of $SO(3)$ decomposes into $\mathbf{2} \oplus \mathbf{3}'$, where $\mathbf{3}'$ is the tensor product of $\mathbf{3}$ and the sign representation $\mathbf{1}'$ of $S_4$.

The symmetric traceless tensor current $(\mathbf{5}, \mathbf{5})$ of $SO(3)$ splits into several currents under $S_4$. We will be interested in symmetries with the currents $(\mathbf{3}', \mathbf{2})$ and $(\mathbf{2}, \mathbf{3}')$ of $S_4$ and will impose a variant of (2.19). These symmetries will be realized in Section 4 and Section 5.

## $(\mathbf{2}, \mathbf{3}')$ *Tensor Symmetry*

Let us consider a case where we have only one of these two currents. Consider the tensor global symmetry with currents $(J_0^{[ij]k}, J^{ij})$ in the $(\mathbf{2}, \mathbf{3}')$ representations (see Appendix A). We label the components of the representation $\mathbf{3}'$ by two symmetric indices $ij$ with $i \neq j$. The current conservation equation is

$$\partial_0 J_0^{[ij]k} = \partial^i J^{jk} - \partial^j J^{ik} . \tag{2.20}$$

We define a conserved charge operator by integrating over the $ij$-plane:

$$Q^{[ij]}(x^k) = \oint dx^i dx^j \, J_0^{[ij]k} , \qquad (\text{no sum in } i, j) . \tag{2.21}$$

Note that these charges are not independent. Since $J_0^{[ij]k} + J_0^{[jk]i} + J_0^{[ki]j} = 0$ (see Appendix A),

$$\oint dx^k Q^{[ij]} + \oint dx^i Q^{[jk]} + \oint dx^j Q^{[ki]} = 0 . \tag{2.22}$$

On a lattice, there are $L^x + L^y + L^z - 1$ such charges where the $-1$ comes from the condition on their sum.

## $(\mathbf{3}', \mathbf{2})$ *Tensor Symmetry*

Next, consider a different tensor global symmetry with currents $(J_0^{ij}, J^{[ij]k})$ in the $(\mathbf{3}', \mathbf{2})$ representations (see Appendix A). The currents obey the conservation equation

$$\partial_0 J_0^{ij} = \partial_k (J^{[ki]j} + J^{[kj]i}) , \tag{2.23}$$

and we impose the differential constraint

$$G \equiv \partial_i \partial_j J_0^{ij} = 0 . \tag{2.24}$$

For every point $(x^j, x^k)$ on the $jk$-plane, we define a charge operator by integrating along the $x^i$ direction:

$$Q^i(x^j, x^k) = \oint dx^i \, J_0^{jk} \,. \tag{2.25}$$

The charge operator is conserved $\partial_0 Q^i(x^j, x^k) = 0$ because of the conservation equation (2.23) and the fact that the three indices of $J^{kij}$ are all different.[4]

How does the charge operator, say, $Q^z(x, y)$ depend on the coordinates $x, y$? Consider the double derivative

$$\partial_x \partial_y Q^z(x, y) = \oint dz \, \partial_x \partial_y J_0^{xy} = 0 \,, \tag{2.26}$$

where we have used the differential condition (2.24) $\partial_x \partial_y J_0^{xy} = -\partial_z(\partial_x J_0^{xz} + \partial_y J_0^{yz})$. This means that

$$Q^z(x, y) = Q_x^z(x) + Q_y^z(y) \tag{2.27}$$

and only the sum of their zero modes is physical. Similar statements are true for the other $Q^i(x^j, x^k)$.

On the lattice, there are $L^x + L^y - 1$ conserved charges $Q^z$ (where the $-1$ comes from the zero mode), rather than $L^x L^y$ of them. Adding all three directions the number of charges is $2L^x + 2L^y + 2L^z - 3$.

## 2.3 $U(1)$ Multipole Global Symmetry

Next, we further generalize the tensor global symmetry (2.10). Consider a continuum field theory with operators $(J_0^I, J^K)$ where the index $I$ and $K$ are respectively in representation $\mathbf{R}_{\text{time}}$ and $\mathbf{R}_{\text{space}}$ of the spatial rotation group. We assume that the operators satisfy the following identity[5]

$$\partial_0 J_0^I = \partial_{j_1} \partial_{j_2} \cdots \partial_{j_n} J^K f_K^{j_1 j_2 \cdots j_n, \, I} \,, \tag{2.30}$$

---

[4]If we do not have the differential condition (2.24), $G \equiv \partial_i \partial_j J_0^{ij}$ is still conserved at every point, i.e. $\partial_0 G = 0$. If it is spontaneously broken, we have many soft modes. If it is unbroken, then we have a separate conserved charge at every point in space.

[5]It might happen that the operator identity (2.30) can be integrated to

$$\partial_0 \widehat{J}_0^{i, \, I} = \partial_j \widehat{J}^{[ji], \, I} + \partial_{j_1} \partial_{j_2} \cdots \partial_{j_{n-1}} J^K f_K^{j_1 j_2 \cdots j_{n-1} i, \, I} \tag{2.28}$$

with well-defined $\widehat{J}_0^{i, \, I}$ and $\widehat{J}^{[ji], \, I}$. A necessary condition for that is

$$J_0^I = \partial_i \widehat{J}_0^{i, \, I} \,. \tag{2.29}$$

This has the effect of reducing the number of spatial derivatives in the right hand side from $n$ to $n-1$, but adds another operator $\widehat{J}^{[ji], \, I}$, which is not present in (2.30). We will focus on the case (2.30) and assume that it cannot be integrated.

where $f_K^{j_1 j_2 \cdots j_n , \, I}$ is an invariant tensor. There might be further differential conditions on these operators. We will refer to the symmetry generated by the currents $(J_0^I, J^K)$ the $(\mathbf{R}_{\text{time}}, \mathbf{R}_{\text{space}})$ *multipole global symmetry.*

Our characterization of the global symmetry is in terms of the currents and their local conservation equations. This formulation of the symmetry is independent of the global topology of the spacetime. This is to be contrasted with the perspectives in, for example, [15, 16], where the emphasis was on the symmetry charges defined in infinite space.

We now discuss two dipole global symmetries that are compatible with the cubic group $S_4$. These two symmetries will be realized in Section 3 and Section 6.

$(\mathbf{1}, \mathbf{3}')$ *Dipole Symmetry*

Consider currents $(J_0, J^{ij})$ in the $(\mathbf{R}_{\text{time}}, \mathbf{R}_{\text{space}}) = (\mathbf{1}, \mathbf{3}')$ of $S_4$. We label the components of the representation $\mathbf{3}'$ by two symmetric indices $ij$ with $i \neq j$. They obey

$$
\begin{aligned}
\partial_0 J_0 &= \frac{1}{2} \partial_i \partial_j J^{ij} \\
&= \partial_x \partial_y J^{xy} + \partial_z \partial_x J^{zx} + \partial_y \partial_z J^{yz} \, ,
\end{aligned}
\tag{2.31}
$$

where the factor $\frac{1}{2}$ comes from the index contraction of $ij$. There are three kinds of conserved charges, each integrated over a plane:

$$
Q_{ij}(x^k) = \oint dx^i \oint dx^j J_0 \, .
\tag{2.32}
$$

They obey the constraint:

$$
\oint dx Q_{yz}(x) = \oint dy Q_{zx}(y) = \oint dz Q_{xy}(z) \, .
\tag{2.33}
$$

On a lattice, we have $L^x + L^y + L^z - 2$ such charges.

$(\mathbf{3}', \mathbf{1})$ *Dipole Symmetry*

The second dipole symmetry is generated by currents $(J_0^{ij}, J)$ with $(\mathbf{R}_{\text{time}}, \mathbf{R}_{\text{space}}) = (\mathbf{3}', \mathbf{1})$ of $S_4$. They obey the conservation equation:

$$
\partial_0 J_0^{ij} = \partial^i \partial^j J \, ,
\tag{2.34}
$$

and a differential condition

$$
\partial^i J_0^{jk} = \partial^j J_0^{ik} \, .
\tag{2.35}
$$

For any closed curve $\mathcal{C}^{xy}$ on the $xy$-plane, there is a conserved charge

$$Q(\mathcal{C}^{xy}, z) = \oint_{\mathcal{C}^{xy} \in (x,y)} ( dx J_0^{zx} + dy J_0^{zy} ) . \tag{2.36}$$

The differential condition (2.35) implies that the charge $Q(\mathcal{C}^{xy}, z)$ is independent of small deformation of the curve $\mathcal{C}^{xy}$, but depends on the $z$ coordinate. Therefore, on the $xy$-plane, the conserved charges are generated by $Q(\mathcal{C}^{xy}_x, z)$ and $Q(\mathcal{C}^{xy}_y, z)$. Here $\mathcal{C}^{xy}_x$ is a closed curve that wraps around the $x$ direction once but not the $y$ direction, and vice versa. There are similar charges on the $xz$ and $yz$ planes.

Finally, there are constraints among these charges:

$$\oint dz Q(\mathcal{C}^{xy}_x, z) = \oint dx Q(\mathcal{C}^{yz}_z, x) ,$$
$$\oint dz Q(\mathcal{C}^{xy}_y, z) = \oint dy Q(\mathcal{C}^{xz}_z, y) , \tag{2.37}$$
$$\oint dx Q(\mathcal{C}^{yz}_y, x) = \oint dy Q(\mathcal{C}^{xz}_x, y) .$$

On a lattice, we have $2L^x + 2L^y + 2L^z - 3$ such charges.

## 2.4   Gauging Global Symmetries

Let us gauge the multipole global symmetries (2.30) (which include the tensor global symmetries (2.10) as special cases). We couple the currents $J_0^I$ and $J^K$ to background fields. Since for $n > 1$ this is not a standard conserved current, this is not ordinary gauging of a global symmetry. We introduce gauge fields $(A_{0,\,I}, A_K)$ and add to the Lagrangian the minimal coupling

$$A_{0,\,I} J_0^I + (-1)^n A_K J^K . \tag{2.38}$$

Because of (2.30), the terms (2.38) are unchanged when the gauge fields transform as

$$A_{0,\,I} \to A_{0,\,I} + \partial_0 \lambda_I ,$$
$$A_K \to A_K + \partial_{j_1} \partial_{j_2} \cdots \partial_{j_n} \lambda_I f_K^{j_1 j_2 \cdots j_n ,\, I} . \tag{2.39}$$

This means that there is a redundancy in the fields $A_{0,\,I}$ and $A_K$, which generalizes ordinary gauge symmetry (or better stated, ordinary gauge redundancy). We will refer to (2.39) as the gauge symmetry of the system.

Note that the gauge parameter $\lambda_I$ is in the representation $\mathbf{R}_{\text{time}}$. If $J_0^I$ is subject to a differential condition, then integrating by parts shows that some deformations of $\lambda_I$ do not

act on the gauge fields. This means that $\lambda_I$ is itself a gauge field. This is familiar in the case of higher-form global symmetries and their corresponding higher-form gauge fields.

# 3 The $\phi$-Theory

In this section we discuss a $3+1$-dimensional continuum field theory of $\phi$ with dipole global symmetries (2.31) and (2.34). The $\phi$-theory is the continuum limit of the $3+1$-dimensional version of the XY-plaquette model in [7, 3]. Certain aspects of this continuum field theory have been discussed in [7–13].

## 3.1 The Lattice Model

The XY-plaquette model is defined on a three-dimensional spatial, cubic lattice with with a phase variable $e^{i\phi_s}$ at every site $s = (\hat{x}, \hat{y}, \hat{z})$. Let $L^x, L^y, L^z$ be the numbers of sites in the $x, y, z$ directions, respectively. We label the sites by $s = (\hat{x}, \hat{y}, \hat{z})$, with integer $\hat{x}^i = 1, \cdots, L^i$. Let $a$ be the lattice spacing. When we take the continuum limit, we will use $x^i = a\hat{x}^i$ to label the coordinates and $\ell^i = aL^i$ to denote the physical size of the system.

The variable $\phi_s$ is $2\pi$-periodic at each site, $\phi_s \sim \phi_s + 2\pi$. Let $\pi_s$ be the conjugate momentum of $\phi_s$. They obey the commutation relation $[\phi_s, \pi_{s'}] = i\delta_{s,s'}$. The $2\pi$-periodicity of $\phi_s$ implies that the eigenvalues of $\pi_s$ are integers. The Hamiltonian is

$$H = \frac{u}{2}\sum_s (\pi_s)^2 - K\sum_{i<j}\sum_s \cos(\Delta_{ij}\phi_s)\,, \tag{3.1}$$

where $\Delta_{xy}\phi_s \equiv \phi_s - \phi_{s+(1,0,0)} - \phi_{s+(0,1,0)} + \phi_{s+(1,1,0)}$ and similarly for $\Delta_{xz}\phi_s$ and $\Delta_{yz}\phi_s$. The second term in the Hamiltonian is a sum over all the plaquettes in the three-dimensional lattice.

This lattice system has a large number of $U(1)$ global symmetries that grows linearly in the size of the system [7]. For every point $\hat{x}_0$ in the $x$ direction, there is a $U(1)$ global symmetry that acts as

$$U(1)_{\hat{x}_0}: \quad \phi_s \to \phi_s + \varphi\,, \qquad \forall\, s = (\hat{x}, \hat{y}, \hat{z}) \text{ with } \hat{x} = \hat{x}_0\,, \tag{3.2}$$

where $\varphi \in [0, 2\pi)$. Similarly we have $U(1)_{\hat{y}_0}$ and $U(1)_{\hat{z}_0}$ associated with the $y$ and $z$ directions, respectively. There are two relations among these symmetries. The composition of all the $U(1)_{\hat{x}_0}$ transformations with the same $\varphi$ is the same as the composition of all the $U(1)_{\hat{y}_0}$ transformations with the same $\varphi$, and the same as the composition of all the

$U(1)_{\hat{z}_0}$ transformations with the same $\varphi$. This composition rotates all the $\phi_s$'s on the three-dimensional lattice simultaneously. In total, we have $L^x + L^y + L^z - 2$ independent $U(1)$ global symmetries.

## 3.2  Continuum Lagrangian

The continuum limit of the XY-plaquette model is a real scalar field theory with Lagrangian

$$
\begin{aligned}
\mathcal{L} &= \frac{\mu_0}{2}(\partial_0\phi)^2 - \frac{1}{4\mu}(\partial_i\partial_j\phi)^2 \\
&= \frac{\mu_0}{2}(\partial_0\phi)^2 - \frac{1}{2\mu}\left[(\partial_x\partial_y\phi)^2 + (\partial_z\partial_x\phi)^2 + (\partial_y\partial_z\phi)^2\right],
\end{aligned} \tag{3.3}
$$

where $\mu_0$ has dimension 2 and $\mu$ is dimensionless. This is the $3+1$-dimensional version of the $\phi$-theory (1.1) in [3].

The equation of motion is

$$
\mu_0\partial_0^2\phi = -\frac{1}{\mu}\left(\partial_x^2\partial_y^2\phi + \partial_z^2\partial_x^2\phi + \partial_y^2\partial_z^2\phi\right). \tag{3.4}
$$

Locally, the field $\phi$ is subject to the gauge symmetry

$$
\phi(t,x,y,z) \sim \phi(t,x,y,z) + 2\pi w^x(x) + 2\pi w^y(y) + 2\pi w^z(z), \tag{3.5}
$$

where $w^i(x^i) \in \mathbb{Z}$ [3]. Because of this gauge identification, the operators $\partial_i\phi$ are not gauge-invariant, while $e^{i\phi}, \partial_i\partial_j\phi$ with $i \neq j$ are well-defined operators. Globally, the field $\phi$ is not a single-valued function, but a section over a nontrivial bundle with transition functions of the form (3.5). An example of such a nontrivial configuration on a spatial 3-torus is

$$
\phi(t,x,y,z) = 2\pi\left[\frac{x}{\ell^x}\Theta(y-y_0) + \frac{y}{\ell^y}\Theta(x-x_0) - \frac{xy}{\ell^x\ell^y}\right]. \tag{3.6}
$$

We refer the readers to [3] for more discussions on the global issues of the $\phi$ field.

## 3.3  Global Symmetries and Their Charges

We now discuss the exotic global symmetries of the continuum field theory.

### 3.3.1   Momentum Dipole Symmetry

The equation of motion (3.4) implies the $(\mathbf{1}, \mathbf{3}')$ dipole global symmetry (2.31)

$$\partial_0 J_0 = \frac{1}{2}\partial_i\partial_j J^{ij} \tag{3.7}$$

with currents [9]

$$
\begin{aligned}
J_0 &= \mu_0\partial_0\phi\,,\\
J^{ij} &= -\frac{1}{\mu}\partial^i\partial^j\phi\,.
\end{aligned}
\tag{3.8}
$$

We will refer to this symmetry as the *momentum dipole symmetry*. This symmetry is the continuum version of (3.2) on the lattice.

The conserved charges (2.32) are

$$Q_{ij}(x^k) = \mu_0\oint dx^i\oint dx^j\partial_0\phi\,. \tag{3.9}$$

They implement

$$\phi(t, x, y, z) \to \phi(t, x, y, z) + f^x(x) + f^y(y) + f^z(z)\,. \tag{3.10}$$

In (3.5), we gauge the $\mathbb{Z}$ part of the momentum dipole symmetry, so that the global form of the symmetry is $U(1)$ as opposed to $\mathbb{R}$.

### 3.3.2   Winding Dipole Symmetry

Since $\partial_i\phi$ is not a well-defined operator, we do *not* have the ordinary winding global symmetry, whose currents are $J_0^i = \frac{1}{2\pi}\partial^i\phi,\ J = \frac{1}{2\pi}\partial_0\phi$.

Instead, we have a $(\mathbf{3}', \mathbf{1})$ dipole global symmetry (2.34)

$$\partial_0 J_0^{ij} = \partial^i\partial^j J\,, \tag{3.11}$$

with currents

$$
\begin{aligned}
J_0^{ij} &= \frac{1}{2\pi}\partial^i\partial^j\phi\,,\\
J &= \frac{1}{2\pi}\partial_0\phi\,.
\end{aligned}
\tag{3.12}
$$

The currents are subject to the differential condition (2.35)

$$\partial^i J_0^{jk} = \partial^j J_0^{ik}\,. \tag{3.13}$$

We will refer to this symmetry as the *winding dipole symmetry*. Note that this symmetry is not present on the lattice. This is similar to the absence of winding global symmetry in the lattice version of the standard XY model.

The conserved charge (2.36) is

$$Q(\mathcal{C}^{xy}, z) = \frac{1}{2\pi} \partial_z \oint_{\mathcal{C}^{xy} \in (x,y)} ( dx \partial_x \phi + dy \partial_y \phi ) , \tag{3.14}$$

where $\mathcal{C}^{xy}$ is a closed curve on the $xy$-plane. The charges for other directions can be similarly defined.

## 3.4 Momentum Modes

In this subsection we discuss states that are charged under the momentum dipole symmetry (3.8).

We start by analyzing the plane wave solutions in $\mathbb{R}^{3,1}$:

$$\phi = C e^{i\omega t + i k_i x^i} . \tag{3.15}$$

The equation of motion (3.4) gives the dispersion relation

$$\omega^2 = \frac{1}{\mu \mu_0} \left( k_x^2 k_y^2 + k_z^2 k_y^2 + k_y^2 k_z^2 \right) . \tag{3.16}$$

Classically, the zero-energy solutions $\omega = 0$ are those modes with at least two of the three $k_i$'s vanishing. In particular, there are classical zero-energy solutions with $k_x = k_y = 0$ but arbitrarily large $k_z$. The momentum dipole symmetry (3.8) maps one such zero-energy classical solution to another. Therefore, we will call these modes the momentum modes. Classically, the momentum dipole symmetry appears to be spontaneously broken, while the winding dipole symmetry does not act on these plane wave solutions.

Similar to the $\phi$-theory in $2+1$ dimensions, this classical picture turns out to be incorrect quantum mechanically.

Let us quantize the momentum modes of $\phi$:

$$\phi(t, x, y, z) = \phi^x(t, x) + \phi^y(t, y) + \phi^z(t, z) , \tag{3.17}$$

where $\phi^i(t, x^i)$ is point-wise $2\pi$-periodic. They share a common zero mode, which implies

the following gauge symmetry parameterized by $c^x(t), c^y(t)$

$$\phi^x(t,x) \to \phi^x(t,x) + c^x(t), \qquad \phi^y(t,y) \to \phi^y(t,y) + c^y(t), \qquad \phi^z(t,z) \to \phi^z(t,z) - c^x(t) - c^y(t).$$
$$(3.18)$$

The Lagrangian of these modes is

$$
\begin{aligned}
L &= \frac{\mu_0}{2} \oint dx\,dy\,dz \left[ \dot\phi^x(t,x) + \dot\phi^y(t,y) + \dot\phi^z(t,z) \right]^2 \\
&= \frac{\mu_0}{2} \left[ \ell^y \ell^z \oint dx (\dot\phi^x)^2 + \ell^z \ell^x \oint dy (\dot\phi^y)^2 + \ell^x \ell^y \oint dz (\dot\phi^z)^2 \right. \\
&\quad \left. + 2\ell^x \oint dy\,\dot\phi^y \oint dz\,\dot\phi^z + 2\ell^y \oint dx\,\dot\phi^x \oint dz\,\dot\phi^z + 2\ell^z \oint dx\,\dot\phi^x \oint dy\,\dot\phi^y \right]
\end{aligned}
$$
$$(3.19)$$

The conjugate momenta are

$$\pi^i(t,x^i) = \mu_0 \left( \ell^j \ell^k \, \dot\phi^i + \ell^k \oint dx^j \dot\phi^j + \ell^j \oint dx^k \dot\phi^k \right). \tag{3.20}$$

They are the charges of the momentum dipole symmetry

$$Q_{ij}(x^k) = \mu_0 \oint dx^i dx^j \, \partial_0 \phi = \pi^k(x^k). \tag{3.21}$$

The gauge symmetry (3.18) implies that the conjugate momenta satisfy

$$\Pi \equiv \oint dx\,\pi^x = \oint dy\,\pi^y = \oint dz\,\pi^z. \tag{3.22}$$

The Hamiltonian is

$$H = \frac{1}{2\mu_0 \ell^x \ell^y \ell^z} \left[ \sum_i \ell^i \oint dx^i (\pi^i)^2 - 2\Pi^2 \right]. \tag{3.23}$$

*Minimally Charged States*

The lowest energy state has,

$$\pi^x = \delta(x - x_0), \qquad \pi^y = \delta(y - y_0), \qquad \pi^z = \delta(z - z_0) \tag{3.24}$$

with some $x_0$, $y_0$, $z_0$. It corresponds to

$$\dot{\phi} = \frac{1}{\mu_0 \ell^x \ell^y \ell^z} \left[ \ell^x \delta(x - x_0) + \ell^y \delta(y - y_0) + \ell^z \delta(z - z_0) - 2 \right] \qquad (3.25)$$

The minimal energy of the charged mode is

$$H = \frac{1}{2\mu_0 \ell^x \ell^y \ell^z} \left[ (\ell^x + \ell^y + \ell^z)\delta(0) - 2 \right] . \qquad (3.26)$$

We see that quantum mechanically the momentum modes have energy of order $\delta(0) = \frac{1}{a}$ (see [3] for more discussion). The classically zero-energy configurations give rise to infinitely heavy modes in the continuum limit. The momentum dipole global symmetry (3.8) is restored quantum mechanically. This is qualitatively similar to the $\phi$-theory in one dimension lower (1.1) [3].

*General Charged States*

More general momentum modes have

$$\begin{aligned}
Q_{yz}(x) &= \pi^x(x) = \sum_\alpha N_{x\,\alpha} \delta(x - x_\alpha) , \\
Q_{zx}(y) &= \pi^y(y) = \sum_\beta N_{y\,\beta} \delta(y - y_\beta) , \\
Q_{xy}(z) &= \pi^z(z) = \sum_\alpha N_{z\,\gamma} \delta(z - z_\gamma) , \\
N &\equiv \sum_\alpha N_{x\,\alpha} = \sum_\beta N_{y\,\beta} = \sum_\gamma N_{z\,\gamma} ,
\end{aligned} \qquad (3.27)$$

where the $N$'s are integers and $\{x_\alpha\}, \{y_\beta\}, \{z_\gamma\}$ are a finite set of points on the $x, y, z$ axes, respectively. On a lattice, there are $L^x + L^y + L^z - 2$ different charged sectors. The minimal energy with these charges is

$$H = \frac{1}{2\mu_0 \ell^x \ell^y \ell^z} \left[ \ell^x \delta(0) \sum_\alpha N_{x\,\alpha}^2 + \ell^y \delta(0) \sum_\beta N_{y\,\beta}^2 + \ell^z \delta(0) \sum_\gamma N_{z\,\gamma}^2 - 2N^2 \right] , \qquad (3.28)$$

which is of order $\frac{1}{a}$.

## 3.5   Winding Modes

Next, we discuss states that are charged under the winding dipole symmetry (3.12).

The winding configurations can be obtained from linear combinations of (3.6):

$$\phi(t,x,y,z) = 2\pi \left[ \frac{x}{\ell^x} \sum_\beta W^y_{x\beta} \Theta(y - y_\beta) + \frac{y}{\ell^y} \sum_\alpha W^x_{y\alpha} \Theta(x - x_\alpha) - W^{xy} \frac{xy}{\ell^x \ell^y} \right]$$

$$+ 2\pi \left[ \frac{x}{\ell^x} \sum_\gamma W^z_{x\gamma} \Theta(z - z_\gamma) + \frac{z}{\ell^z} \sum_\alpha W^x_{z\alpha} \Theta(x - x_\alpha) - W^{zx} \frac{zx}{\ell^z \ell^x} \right] \qquad (3.29)$$

$$+ 2\pi \left[ \frac{z}{\ell^z} \sum_\beta W^y_{z\beta} \Theta(y - y_\beta) + \frac{y}{\ell^y} \sum_\gamma W^z_{y\gamma} \Theta(z - z_\gamma) - W^{yz} \frac{yz}{\ell^y \ell^z} \right]$$

where $W^i_{j\alpha} \in \mathbb{Z}$ and $W^{ij} = \sum_\alpha W^j_{i\alpha} = \sum_\beta W^i_{j\beta}$.

The winding dipole charge is

$$Q(\mathcal{C}^{ij}_i) = \frac{1}{2\pi} \oint dx^i \partial_k \partial_i \phi = \sum_\gamma W^k_{i\gamma} \delta(x^k - x^k_\gamma) \,, \qquad (3.30)$$

where $\mathcal{C}^{ij}_i$ is any closed curve on the $ij$-torus that wraps around the $i$ cycle once but not the $j$ cycle. They obey

$$\oint dx^k Q(\mathcal{C}^{ij}_i) = \oint dx^i Q(\mathcal{C}^{kj}_k) = W^{ik} \,. \qquad (3.31)$$

The Hamiltonian for this winding mode can be computed in a similar way as in [3]

$$H = \frac{2\pi^2 \ell^z}{\mu \ell^x \ell^y} \left[ \ell^x \sum_\alpha (W^x_{y\alpha})^2 \delta(0) + \ell^y \sum_\beta (W^y_{x\beta})^2 \delta(0) - (W^{xy})^2 \right]$$

$$+ \frac{2\pi^2 \ell^x}{\mu \ell^y \ell^z} \left[ \ell^y \sum_\beta (W^y_{z\beta})^2 \delta(0) + \ell^z \sum_\gamma (W^z_{y\gamma})^2 \delta(0) - (W^{yz})^2 \right] \qquad (3.32)$$

$$+ \frac{2\pi^2 \ell^y}{\mu \ell^z \ell^x} \left[ \ell^z \sum_\gamma (W^z_{x\gamma})^2 \delta(0) + \ell^x \sum_\alpha (W^x_{z\alpha})^2 \delta(0) - (W^{zx})^2 \right] \,.$$

We find that the winding modes have energy of order $\frac{1}{a}$, which diverges in the continuum limit.

## 3.6  Robustness and Universality

We end this section by mentioning two subtle issues that were discussed in the $2 + 1$-dimensional version of this model in [3].

First is the issue of robustness. As we saw, the theory has a large symmetry and in the continuum limit, all the charged states carry high energy (of order $1/a$ or higher) under this symmetry. Therefore, operators carrying charges under those symmetries are irrelevant in the low-energy theory. As a result, the model is robust under small enough deformations that violate this symmetry.

Second is the universality of the computation of the energies of these charged states.[6] We argued in [3] that analyzing them using the minimal Lagrangian leads to correct qualitative conclusions, but the detailed quantitative answers could be modified by some higher derivative terms. Since the discussion of these two issues is identical to that in [3], we will not repeat it here.

# 4    The $\hat{\phi}$-Theory

In this section we discuss a $3 + 1$-dimensional continuum field theory of $\hat{\phi}$ with tensor global symmetries (2.20) and (2.23). It is the continuum limit of a lattice model that we will introduce.

## 4.1    The Lattice Model

On a three-dimensional spatial, cubic lattice, there are three $U(1)$ phases at every site $s = (\hat{x}, \hat{y}, \hat{z})$,

$$e^{i\hat{\phi}_s^{x(yz)}}, \quad e^{i\hat{\phi}_s^{y(zx)}}, \quad e^{i\hat{\phi}_s^{z(xy)}}, \tag{4.1}$$

subject to the constraint $e^{i(\hat{\phi}_s^{x(yz)} + \hat{\phi}_s^{y(zx)} + \hat{\phi}_s^{z(xy)})} = 1$.

Let $\pi_s^{k(ij)}$ be the conjugate momenta of $\hat{\phi}_s^{k(ij)}$. Here we slightly abuse the notation because the momenta $\pi_s^{k(ij)}$ do not sum to zero. Instead, the above constraint implies a gauge ambiguity:

$$\pi_s^{x(yz)} \sim \pi_s^{x(yz)} + c, \quad \pi_s^{y(zx)} \sim \pi_s^{y(zx)} + c, \quad \pi_s^{z(xy)} \sim \pi_s^{z(xy)} + c, \tag{4.2}$$

separately at each site.

---

[6]We thank P. Gorantla and H.T. Lam for useful discussions about the universality of these models.

The Hamiltonian is

$$H = \hat{u} \sum_s \left[ (\pi_s^{x(yz)} - \pi_s^{y(zx)})^2 + (\pi_s^{y(zx)} - \pi_s^{z(xy)})^2 + (\pi_s^{z(xy)} - \pi_s^{x(yz)})^2 \right]$$
$$- \hat{K} \sum_s \left[ \cos(\hat{\phi}_{s+(1,0,0)}^{x(yz)} - \hat{\phi}_s^{x(yz)}) + \cos(\hat{\phi}_{s+(0,1,0)}^{y(zx)} - \hat{\phi}_s^{y(zx)}) + \cos(\hat{\phi}_{s+(0,0,1)}^{z(xy)} - \hat{\phi}_s^{z(xy)}) \right] . \tag{4.3}$$

This system has a large number of $U(1)$ global symmetries. For every point $\hat{z}_0$, there is $U(1)_{\hat{z}_0}$ global symmetry that acts as

$$\hat{\phi}_s^{x(yz)} \to \hat{\phi}_s^{x(yz)} + \varphi , \quad \hat{\phi}_s^{y(zx)} \to \hat{\phi}_s^{y(zx)} - \varphi , \quad \hat{\phi}_s^{z(xy)} \to \hat{\phi}_s^{z(xy)} ,$$
$$\forall \ s = (\hat{x}, \hat{y}, \hat{z}) \ \text{ with } \ \hat{z} = \hat{z}_0 \tag{4.4}$$

where $\varphi \in [0, 2\pi)$. Similarly we have $U(1)_{\hat{x}_0}$ and $U(1)_{\hat{y}_0}$ for every point $\hat{x}_0$ and $\hat{y}_0$, respectively. Note that the composition of all the $U(1)_{\hat{x}_0}, U(1)_{\hat{y}_0}, U(1)_{\hat{z}_0}$ is trivial. Therefore on a lattice, we have $L^x + L^y + L^z - 1$ such $U(1)$ global symmetries.

## 4.2 Continuum Lagrangian

The continuum limit of the lattice model discussed above is a theory of $\hat{\phi}^{i(jk)}$ in the **2** of $S_4$ with Lagrangian

$$\mathcal{L} = \frac{\hat{\mu}_0}{12} (\partial_0 \hat{\phi}^{i(jk)})^2 - \frac{\hat{\mu}}{4} (\partial_k \hat{\phi}^{k(ij)})^2 , \tag{4.5}$$

subject to the constraint $\hat{\phi}^{x(yz)} + \hat{\phi}^{y(zx)} + \hat{\phi}^{z(xy)} = 0$. Here the coefficient $\hat{\mu}_0, \hat{\mu}$ have mass dimension 1.

A field in the **2** can also be expressed as $\hat{\phi}^{[ij]k}$ (see Appendix A). It is related to $\hat{\phi}^{k(ij)}$ by:

$$\hat{\phi}^{k(ij)} = \hat{\phi}^{[ki]j} + \hat{\phi}^{[kj]i} ,$$
$$\hat{\phi}^{[ij]k} = \frac{1}{3} (\hat{\phi}^{i(jk)} - \hat{\phi}^{j(ki)}) . \tag{4.6}$$

We have $\hat{\phi}_{i(jk)} \hat{\phi}^{i(jk)} = 3 \hat{\phi}_{[ij]k} \hat{\phi}^{[ij]k}$. In the $\hat{\phi}^{[ij]k}$ basis, the Lagrangian is

$$\mathcal{L} = \frac{\hat{\mu}_0}{4} (\partial_0 \hat{\phi}^{[ij]k})^2 - \frac{\hat{\mu}}{4} \left[ \partial_k (\hat{\phi}^{[ki]j} + \hat{\phi}^{[kj]i}) \right]^2$$
$$= \frac{\hat{\mu}_0}{2} \left[ (\partial_0 \hat{\phi}^{[xy]z})^2 + (\partial_0 \hat{\phi}^{[yz]x})^2 + (\partial_0 \hat{\phi}^{[zx]y})^2 \right] \tag{4.7}$$
$$- \frac{\hat{\mu}}{2} \left\{ \left[ \partial_z (\hat{\phi}^{[zx]y} - \hat{\phi}^{[yz]x}) \right]^2 + \left[ \partial_y (\hat{\phi}^{[yz]x} - \hat{\phi}^{[xy]z}) \right]^2 + \left[ \partial_x (\hat{\phi}^{[xy]z} - \hat{\phi}^{[zx]y}) \right]^2 \right\} ,$$

subject to the constraint $\hat{\phi}^{[xy]z} + \hat{\phi}^{[yz]x} + \hat{\phi}^{[zx]y} = 0$. For clarity, we will write many of our expressions in both the $\hat{\phi}^{i(jk)}$ and the $\hat{\phi}^{[ij]k}$ bases below.

The fields $\hat{\phi}^{i(jk)}$ are point-wise $2\pi$-periodic in a way compatible with the constraint $\hat{\phi}^{x(yz)} + \hat{\phi}^{y(zx)} + \hat{\phi}^{z(xy)} = 0$. Locally, we impose the following three gauge symmetries:[7]

$$
\begin{aligned}
\hat{\phi}^{x(yz)} &\sim \hat{\phi}^{x(yz)}, & \hat{\phi}^{y(zx)} &\sim \hat{\phi}^{y(zx)} + 2\pi w^x(x), & \hat{\phi}^{z(xy)} &\sim \hat{\phi}^{z(xy)} - 2\pi w^x(x), \\
\hat{\phi}^{x(yz)} &\sim \hat{\phi}^{x(yz)} - 2\pi w^y(y), & \hat{\phi}^{y(zx)} &\sim \hat{\phi}^{y(zx)}, & \hat{\phi}^{z(xy)} &\sim \hat{\phi}^{z(xy)} + 2\pi w^y(y), \\
\hat{\phi}^{x(yz)} &\sim \hat{\phi}^{x(yz)} + 2\pi w^z(z), & \hat{\phi}^{y(zx)} &\sim \hat{\phi}^{y(zx)} - 2\pi w^z(z), & \hat{\phi}^{z(xy)} &\sim \hat{\phi}^{z(xy)},
\end{aligned}
\tag{4.9}
$$

where $w^i(x^i) \in \mathbb{Z}$ is a discontinuous, integer-valued function in $x^i$. It follows that while $e^{i\hat{\phi}^{k(ij)}}, \partial_k \hat{\phi}^{k(ij)}$ are well-defined, operators such as $\partial_z \hat{\phi}^{x(yz)}$ are not. Note that these identifications leave the Lagrangian invariant and are compatible with the constraint $\hat{\phi}^{x(yz)} + \hat{\phi}^{y(zx)} + \hat{\phi}^{z(xy)} = 0$.

## 4.3  Global Symmetries and Their Charges

We now discuss the global symmetries in the continuum $\hat{\phi}$-theory.

### 4.3.1  Momentum Tensor Symmetry

The equation of motion in the $\hat{\phi}^{i(jk)}$ basis[8]

$$
\hat{\mu}_0 \partial_0^2 \hat{\phi}^{i(jk)} = \hat{\mu} \left[ 2\partial_i^2 \hat{\phi}^{i(jk)} - \partial_j^2 \hat{\phi}^{j(ki)} - \partial_k^2 \hat{\phi}^{k(ij)} \right], \quad \text{(no sum in } i, j, k) \tag{4.10}
$$

or in the $\hat{\phi}^{[ij]k}$ basis

$$
\hat{\mu}_0 \partial_0^2 \hat{\phi}^{[ij]k} = \hat{\mu} \left[ \partial_i^2 (\hat{\phi}^{[ij]k} + \hat{\phi}^{[ik]j}) - \partial_j^2 (\hat{\phi}^{[ji]k} + \hat{\phi}^{[jk]i}) \right]. \tag{4.11}
$$

---

[7]In the $\hat{\phi}^{[ij]k}$ basis, this gauge symmetry becomes

$$
\hat{\phi}^{[yz]x} \sim \hat{\phi}^{[yz]x} + \frac{4\pi}{3} w^x(x), \quad \hat{\phi}^{[zx]y} \sim \hat{\phi}^{[zx]y} - \frac{2\pi}{3} w^x(x), \quad \hat{\phi}^{[xy]z} \sim \hat{\phi}^{[xy]z} - \frac{2\pi}{3} w^x(x). \tag{4.8}
$$

and so on.

[8]It is important to take the constraint $\hat{\phi}^{i(jk)} + \hat{\phi}^{j(ki)} + \hat{\phi}^{k(ij)} = 0$ into account when deriving the equation of motion.

These are recognized as the conservation equation (2.20) for the tensor global symmetry (5.29) whose current is in $(\mathbf{2}, \mathbf{3}')$:

$$
\begin{aligned}
J_0^{i(jk)} &= \hat{\mu}_0 \, \partial_0 \hat{\phi}^{i(jk)} \,, \\
J^{ij} &= \hat{\mu} \, \partial_k \hat{\phi}^{k(ij)} \,.
\end{aligned}
\tag{4.12}
$$

or

$$
\begin{aligned}
J_0^{[ij]k} &= \hat{\mu}_0 \, \partial_0 \hat{\phi}^{[ij]k} \,, \\
J^{ij} &= \hat{\mu} \, \partial_k (\hat{\phi}^{[ki]j} + \hat{\phi}^{[kj]i}) \,.
\end{aligned}
\tag{4.13}
$$

We will refer to the symmetry generated by this current as the *momentum tensor symmetry*. This is the continuum version of the symmetry (4.4) on the lattice.

The charge operator is

$$
Q^{[ij]}(x^k) = \hat{\mu}_0 \oint dx^i \oint dx^j \, \partial_0 \hat{\phi}^{[ij]k} \,, \quad \text{(no sum in } i, j\text{)} \,.
\tag{4.14}
$$

Note that $\oint dz Q^{xy} + \oint dx Q^{yz} + \oint dy Q^{xz} = 0$. $Q^{[xy]}(z)$ implements

$$
\hat{\phi}^{x(yz)} \to \hat{\phi}^{x(yz)} + f^z(z) \,, \quad \hat{\phi}^{y(zx)} \to \hat{\phi}^{y(zx)} - f^z(z) \,, \quad \hat{\phi}^{z(xy)} \to \hat{\phi}^{z(xy)} \,.
\tag{4.15}
$$

The integer part of the momentum tensor global symmetry is gauged (4.9), so that the global form of the symmetry is $U(1)$ as opposed to $\mathbb{R}$.

### 4.3.2 Winding Tensor Symmetry

Consider the following currents in the $(\mathbf{3}', \mathbf{2})$ in the $\hat{\phi}^{i(jk)}$ basis

$$
\begin{aligned}
J_0^{ij} &= \frac{1}{2\pi} \partial_k \hat{\phi}^{k(ij)} \,, \\
J^{i(jk)} &= \frac{1}{2\pi} \partial_0 \hat{\phi}^{i(jk)} \,,
\end{aligned}
\tag{4.16}
$$

or in the $\hat{\phi}^{[ij]k}$ basis

$$
\begin{aligned}
J_0^{ij} &= \frac{1}{2\pi} \partial_k (\hat{\phi}^{[ki]j} + \hat{\phi}^{[kj]i}) \,, \\
J^{[ij]k} &= \frac{1}{2\pi} \partial_0 \hat{\phi}^{[ij]k} \,.
\end{aligned}
\tag{4.17}
$$

They obey the conservation equation of the $(\mathbf{3}', \mathbf{2})$ tensor global symmetry (2.23)

$$
\partial_0 J_0^{ij} = \partial_k (J^{[ki]j} + J^{[kj]i}) \,.
\tag{4.18}
$$

Since $\hat{\phi}^{k(ij)} + \hat{\phi}^{i(jk)} + \hat{\phi}^{j(ki)} = 0$, the current obeys the differential constraint (2.24)

$$\partial_i \partial_j J_0^{ij} = 0 \,. \tag{4.19}$$

We will refer to the symmetry generated by this current as the *winding tensor symmetry*. This symmetry is not present on the lattice.

The charge operator is

$$Q^k(x^i, x^j) = \oint dx^k \, J_0^{ij} = \frac{1}{2\pi} \oint dx^k \, \partial_k \hat{\phi}^{k(ij)} \,. \tag{4.20}$$

The differential condition (4.19) implies that $Q^k(x^i, x^j)$ is a function of $x^i$ plus another function of $x^j$.

## 4.4  Momentum Modes

We now discuss states that are charged under the momentum tensor global symmetry (4.12).

We start with plane wave solutions of the equation of motions in $\mathbb{R}^{3,1}$,

$$\hat{\phi}^{[ij]k} = C^{[ij]k} e^{i\omega t + ik_i x^i} \,, \tag{4.21}$$

with constant $C^{[ij]k}$ in the **2**. The dispersion relation is

$$\frac{\hat{\mu}_0^2}{\hat{\mu}^2} \omega^4 - 2\frac{\hat{\mu}_0}{\hat{\mu}} \omega^2 (k_x^2 + k_y^2 + k_z^2) + 3(k_x^2 k_y^2 + k_x^2 k_z^2 + k_y^2 k_z^2) = 0 \,, \tag{4.22}$$

leading to

$$\omega_\pm^2 = \frac{\hat{\mu}}{\hat{\mu}_0} \left[ (k_x^2 + k_y^2 + k_z^2) \pm \sqrt{(k_x^2 + k_y^2 + k_z^2)^2 - 3(k_x^2 k_y^2 + k_x^2 k_z^2 + k_y^2 k_z^2)} \right] \,, \tag{4.23}$$

For either solution $\omega_\pm^2$, the fields are

$$\hat{\phi}^{[ij]k} = C(k_i^2 - k_j^2) \left( \frac{\hat{\mu}_0}{\hat{\mu}} \omega^2 - 3k_k^2 \right) e^{i\omega t + ik_i x^i} \,, \tag{4.24}$$

with constant $C$ and for both signs in (4.23).

The limit where two of the components of the momenta go to zero and the third one is

generic, say $k_x, k_y \to 0$, is interesting. Here for the branch with the plus sign,

$$\omega_+^2 = \frac{2\hat{\mu}_0}{\hat{\mu}} k_z^2 \,, \qquad (4.25)$$

we can take the limit of the solution (4.24) above

$$\hat{\phi}^{[xy]z} = 0 \,, \quad \hat{\phi}^{[yz]x} = -\hat{\phi}^{[zx]y} = Ce^{i\omega_+ t + ik_z z} \,. \qquad (4.26)$$

In the branch with the minus sign, the energy is zero

$$\omega_- = 0 \,. \qquad (4.27)$$

We expand the solution (4.24) for small $k_x, k_y$ and divide by some common factor to obtain

$$\hat{\phi}^{[xy]z} = 2Ce^{ik_z z} \,, \quad \hat{\phi}^{[yz]x} = \hat{\phi}^{[zx]y} = -Ce^{ik_z z} \,. \qquad (4.28)$$

This means that we have zero energy states with arbitrary $k_z$ as long as they have vanishing momentum in the $x$ and $y$ directions. These modes are spread in $x$ and $y$, but can have arbitrary $z$ dependence.

We can state the previous result as follows. The classical Lagrangian of the $\hat{\phi}$ theory admits the following classical zero-energy solutions:

$$\begin{aligned}
\hat{\phi}^{x(yz)} &= \hat{\phi}_y(y) - \hat{\phi}_z(z) \,, \\
\hat{\phi}^{y(zx)} &= \hat{\phi}_z(z) - \hat{\phi}_x(x) \,, \\
\hat{\phi}^{z(xy)} &= \hat{\phi}_x(x) - \hat{\phi}_y(y) \,,
\end{aligned} \qquad (4.29)$$

labeled by three functions $\hat{\phi}_i(x^i)$. They are time independent because they have vanishing energy. These classical configurations are related to each other by the momentum tensor symmetry (4.15). This explains the classical infinite degeneracy of the ground states.

In order to quantize these modes, we give them time dependence $\hat{\phi}_i(t, x^i)$ and study their effective Lagrangian. The gauge symmetry (4.9) implies that $\hat{\phi}_i$ is pointwise $2\pi$-periodic, $\hat{\phi}_i(t, x^i) \sim \hat{\phi}_i(t, x^i) + 2\pi w^i(x^i)$ with $w^i(x^i) \in \mathbb{Z}$. The $\hat{\phi}_i$'s share a common zero mode, giving rise to a gauge symmetry:

$$\hat{\phi}_x(t, x) \to \hat{\phi}_x(t, x) + c(t) \,, \quad \hat{\phi}_y(t, y) \to \hat{\phi}_y(t, y) + c(t) \,, \quad \hat{\phi}_z(t, z) \to \hat{\phi}_z(t, z) + c(t) \,. \quad (4.30)$$

The effective Lagrangian of these modes is

$$L = \frac{\hat{\mu}_0}{6} \oint dx \oint dy \oint dz \left[ (\dot{\hat{\phi}}_y - \dot{\hat{\phi}}_z)^2 + (\dot{\hat{\phi}}_z - \dot{\hat{\phi}}_x)^2 + (\dot{\hat{\phi}}_x - \dot{\hat{\phi}}_y)^2 \right]$$
$$= \frac{\hat{\mu}_0}{3} \ell^x \ell^y \ell^z \left[ \sum_i \frac{1}{\ell^i} \oint dx^i (\dot{\hat{\phi}}_i)^2 - \sum_{i \neq j} \frac{1}{2\ell^i \ell^j} \left( \oint dx^i \dot{\hat{\phi}}_i \right) \left( \oint dx^j \dot{\hat{\phi}}_j \right) \right] . \tag{4.31}$$

Let us quantize these modes. The momentum conjugate to $\hat{\phi}_i$ is

$$\pi^i(t, x^i) = \frac{\hat{\mu}_0}{3} \frac{\ell^x \ell^y \ell^z}{\ell^i} \left( 2\dot{\hat{\phi}}_i(t, x^i) - \frac{1}{\ell^j} \oint dx^j \dot{\hat{\phi}}_j - \frac{1}{\ell^k} \oint dx^k \dot{\hat{\phi}}_k \right) \qquad i \neq j \neq k . \tag{4.32}$$

The gauge symmetry (4.30) implies that these momenta are not independent

$$\oint dx\, \pi^x(t, x) + \oint dy\, \pi^y(t, y) + \oint dz\, \pi^z(t, z) = 0 , \tag{4.33}$$

The conserved charges are expressed simply in terms of these momenta

$$Q^{[jk]}(x^i) = \hat{\mu}_0 \oint dx^j dx^k \partial_0 \hat{\phi}^{[jk]i} = -\frac{\hat{\mu}_0}{3} \oint dx^j dx^k (2\dot{\hat{\phi}}_i - \dot{\hat{\phi}}_j - \dot{\hat{\phi}}_k) = -\pi^i(t, x^i) . \tag{4.34}$$

The Hamiltonian is

$$H = \sum_i \oint dx^i \pi^i \dot{\hat{\phi}}_i - L = \frac{3}{4\hat{\mu}_0 \ell^x \ell^y \ell^z} \sum_i \left[ \ell^i \oint dx^i (\pi^i)^2 - \frac{1}{3} \left( \oint dx^i \pi^i \right)^2 \right] \tag{4.35}$$

This can be checked by substituting the expression (4.32) for $\pi^i$ in terms of $\dot{\hat{\phi}}_i$.

### Minimally Charged States

The point-wise periodicity $\hat{\phi}_i(t, x^i) \sim \hat{\phi}_i(t, x^i) + 2\pi w^i(x^i)$ implies that $\pi^i$ is a linear combination of delta functions with integer coefficients. The lowest energy charged states are of the form

$$\pi^x = \delta(x - x_0), \quad \pi^y = -\delta(y - y_0), \quad \pi^z = 0 \tag{4.36}$$

with energy

$$H = \frac{3}{4\hat{\mu}_0} \frac{1}{\ell^z} \left[ \frac{\delta(0)}{\ell^x} + \frac{\delta(0)}{\ell^y} - \frac{2}{3\ell^x \ell^y} \right] . \tag{4.37}$$

### General Charged States

More general charged states are labeled by $n_{x\,\alpha}, n_{y\,\beta}, n_{z\,\gamma} \in \mathbb{Z}$:

$$Q^{[yz]}(x) = -\pi^x(x) = -\sum_\alpha n_{x\,\alpha}\delta(x - x_\alpha),$$

$$Q^{[zx]}(y) = -\pi^y(y) = -\sum_\beta n_{y\,\beta}\delta(y - y_\beta), \qquad (4.38)$$

$$Q^{[xy]}(z) = -\pi^z(z) = -\sum_\gamma n_{z\,\gamma}\delta(z - z_\gamma),$$

subject to the constraint (4.33)

$$\sum_\alpha n_{x\,\alpha} + \sum_\beta n_{y\,\beta} + \sum_\gamma n_{z\,\gamma} = 0. \qquad (4.39)$$

The minimal energy with these charges (4.38) is

$$H = \frac{3}{4\hat{\mu}_0 \ell^x \ell^y \ell^z}\left[\delta(0)\left(\ell^x \sum_\alpha n_{x\,\alpha}^2 + \ell^y \sum_\beta n_{y\,\beta}^2 + \ell^z \sum_\gamma n_{z\,\gamma}^2\right)\right.$$
$$\left. -\frac{1}{3}\left(\left(\sum_\alpha n_{x\,\alpha}\right)^2 + \left(\sum_\beta n_{y\,\beta}\right)^2 + \left(\sum_\gamma n_{z\,\gamma}\right)^2\right)\right]. \qquad (4.40)$$

These momentum modes have energy order $\frac{1}{a}$, which becomes infinite in the strict continuum limit.

## 4.5 Winding Modes

In this subsection we discuss states that are charged under the winding tensor symmetry (4.16).

The gauge symmetry (4.9) gives rise to the winding modes:

$$\hat{\phi}^{x(yz)} = 2\pi\frac{x}{\ell^x}\left(W_x^y(y) + W_x^z(z)\right) - 2\pi\frac{W_y^z(z)y}{\ell^y} - 2\pi\frac{W_z^y(y)z}{\ell^z},$$

$$\hat{\phi}^{y(zx)} = 2\pi\frac{y}{\ell^y}\left(W_y^z(z) + W_y^x(x)\right) - 2\pi\frac{W_z^x(x)z}{\ell^z} - 2\pi\frac{W_x^z(z)x}{\ell^x}, \qquad (4.41)$$

$$\hat{\phi}^{z(xy)} = 2\pi\frac{z}{\ell^z}\left(W_z^x(x) + W_z^y(y)\right) - 2\pi\frac{W_y^x(x)y}{\ell^y} - 2\pi\frac{W_x^y(y)x}{\ell^x},$$

where we have 6 such integer-valued $W_j^i(x^i) \in \mathbb{Z}$. These winding modes realize the charges

(4.20) of the winding tensor symmetry,

$$Q^z(x,y) = \frac{1}{2\pi} \oint dz \partial_z \hat{\phi}^{z(xy)} = W_z^x(x) + W_z^y(y) \,, \tag{4.42}$$

and similarly for the other two charges.

Consider two winding modes that differ by the following shift

$$\begin{aligned} W_z^x(x) &\to W_z^x(x) + 1 \,, \\ W_z^y(y) &\to W_z^y(y) - 1 \,. \end{aligned} \tag{4.43}$$

While the winding charge (4.42) is invariant under this shift, $\hat{\phi}^{k(ij)}$ changes by a momentum mode $\hat{\phi}_z(t,z)$ (use (4.41) and then (4.29)):

$$\begin{aligned} \hat{\phi}^{x(yz)} &\sim \hat{\phi}^{x(yz)} + 2\pi \frac{z}{\ell^z} \,, \\ \hat{\phi}^{y(zx)} &\sim \hat{\phi}^{y(zx)} - 2\pi \frac{z}{\ell^z} \,, \\ \hat{\phi}^{z(xy)} &\sim \hat{\phi}^{z(xy)} \,. \end{aligned} \tag{4.44}$$

Therefore, the difference between them is a mode we have already discussed and we can focus on just one of them. On a lattice, we are left with $2L^x + 2L^y + 2L^z - 3$ different winding sectors.

Let us compute the energy of the winding modes (4.41). We will focus on $W_z^x(x)$ and $W_z^y(y)$. Their contribution to the Hamiltonian is

$$\begin{aligned} H &= \frac{\hat{\mu}}{2} \oint dx dy dz (\partial_z \hat{\phi}^{z(xy)})^2 \\ &= \frac{2\pi^2 \hat{\mu}}{\ell^z} \left[ \ell^y \oint dx W_z^x(x)^2 + \ell^x \oint dy W_z^y(y)^2 + 2 \oint dx W_z^x(x) \oint dy W_z^y(y) \right] \,. \end{aligned} \tag{4.45}$$

There are similar contributions from the other $W$'s. Since the values of $W_j^i(x^i)$ are independent integers at every point in $x^i$, the energy of a generic winding mode is of order $a$. To see this more explicitly, we can introduce a lattice regularization with discretized space $\hat{x}^i = 1, 2, \cdots, L^i$. Then the Hamiltonian takes the form

$$H = \frac{2\pi^2 \hat{\mu}}{\ell^z} \left[ \ell^y a \sum_{\hat{x}=1}^{L^x} W_z^x(\hat{x})^2 + \ell^x a \sum_{\hat{y}=1}^{L^y} W_z^y(\hat{y})^2 + 2a^2 \sum_{\hat{x}=1}^{L^x} W_z^x(\hat{x}) \sum_{\hat{y}=1}^{L^y} W_z^y(\hat{y}) \right] \tag{4.46}$$

If we only have order one nonzero $W$'s (rather than order $1/a$ of them), then the energy of such winding mode is of order $a$.

The momentum and winding states of the $\phi$-theory have energies of order $1/a$ (Sections 3.4 and 3.5). The same is true for the momentum modes of the $\hat{\phi}$-theory (Section 4.4). Therefore, we can study the strict continuum limit in which these states are absent, or we can also include them in the Hilbert space. Being the lowest energy states with these charges, their analysis is meaningful.

This is not the case for the winding states of the $\hat{\phi}$-theory of this section. Their energy is or order $a$ – it vanishes in the continuum limit. Therefore, the spectrum of the theory must include these winding states.

### An Important Comment

The fact that the winding states have energy of order $a$, which vanishes in the continuum limit, leads us to an important comment.

Consider the configuration

$$
\hat{\phi}^{x(yz)} = -\hat{\phi}^{z(xy)} = 2\pi \left[ \frac{x}{\ell^x} \Theta(y - y_0) + \frac{y}{\ell^y} \Theta(x - x_0) - \frac{xy}{\ell^x \ell^y} \right]
$$
$$
\hat{\phi}^{y(xz)} = 0 \,.
$$
(4.47)

It seems like a valid configuration in our continuum field theory, because it is periodic when $\hat{\phi}$ is circle-valued. We are going to argue that it is not a valid configuration of the continuum theory.

The configuration (4.47) has

$$
\partial_x \hat{\phi}^{x(yz)} = 2\pi \left[ \frac{1}{\ell^x} \Theta(y - y_0) + \frac{y}{\ell^y} \delta(x - x_0) - \frac{y}{\ell^x \ell^y} \right] \,.
$$
(4.48)

The existence of the delta function means that its energy is of order $1/a$. Furthermore, its winding tensor charge (4.20)

$$
Q^x = \frac{1}{2\pi} \oint dx \partial_x \hat{\phi}^{x(yz)} = \Theta(y - y_0)
$$
(4.49)

is not single-valued along the $y$ direction. This reflects the fact that the underlying lattice theory violates the winding tensor symmetry at energies of order $1/a$.

Configurations like (4.47) are not present in the strict continuum limit. Their infinite action makes them irrelevant. Furthermore, we argue that we should not consider states in the Hilbert space constructed on top of such configurations because they do not carry a new conserved charge. In this respect, states built on top of these configurations are different from

the momentum and winding states of the $\phi$-theory (Sections 3.4 and 3.5) and the momentum states of the $\hat{\phi}$-theory (Section 4.4).

Note that we did not have such a subtlety in the $\phi$-theory (see Section 3). There, the winding dipole symmetry (3.12) was also absent on the lattice and was present only in the continuum limit. However, there the lowest states charged under the winding symmetry were at order $1/a$. Therefore, they were meaningful.

There is another way to state why a configuration like (4.47) should not be considered in the continuum field theory. We imposed on our continuum field theory the gauge symmetry (4.9) and then studied field configurations twisted by this gauge symmetry. The gauge symmetry on the lattice is larger. It includes arbitrary $2\pi$ shifts at every spacetime point preserving $\hat{\phi}_s^{x(yz)} + \hat{\phi}_s^{y(zx)} + \hat{\phi}_s^{z(xy)} = 0$. The configuration (4.47) is a twisted configuration by this larger gauge symmetry, but it is not a twisted configuration of the smaller gauge symmetry (4.9). To see that, note that the transition function at $y = \ell^y$

$$\hat{\phi}^{x(yz)}(t, x, y = \ell^y, z) = \hat{\phi}^{x(yz)}(t, x, y = 0, z) + 2\pi\Theta(x - x_0), \qquad (4.50)$$

is not one of the identifications in (4.9).

## 4.6   Robustness and Universality

Let us discuss deformations of the minimal Lagrangian (4.5) of the $\hat{\phi}$-theory.

We start with the issue of robustness. Without imposing any global symmetry, we can perturb the theory by the local operator $e^{i\hat{\phi}^{i(jk)}}$. Naively, such a term gives the field $\hat{\phi}$ a mass and gaps the system. But since this local operator is charged under the momentum tensor symmetry (4.15), it creates a momentum mode with energy of order $1/a$ (see Section 4.4). As a result, this operator is irrelevant in the low-energy theory. Therefore, in the continuum limit ($a \to 0$ with fixed system size $\ell^i$), the model is robust under such small deformations.

Next, let us discuss the universality of our computations for the energy of various charged states. For example, we can add to the minimal Lagrangian

$$g(\partial_x \partial_0 \hat{\phi}^{z(xy)})^2 . \qquad (4.51)$$

Since it has two more derivatives than the leading term $(\partial_0 \hat{\phi}^{z(xy)})^2$, we should scale $g \sim a^2$. Therefore it has no effects on a generic plane wave mode.

However, such a higher derivative term does affect the momentum modes. For these modes, $\pi \sim \partial_0 \hat{\phi}$ is a sum of delta functions and the additional derivatives in (4.51) are not suppressed. More precisely, the term (4.51) shifts the energy of the momentum modes by

$g/a^3 \sim 1/a$. Therefore, the quantitative value of the energy of the momentum modes in Section 4.4 is not universal and receives $1/a$ correction from the higher derivative terms. However, their qualitative behavior is universal. This similar to the momentum modes in the $\phi$-theory of Section 3 and in [3].

Next, consider the following higher derivative term

$$g(\partial_x \partial_z \hat{\phi}^{z(xy)})^2 \tag{4.52}$$

with $g$ of order $a^2$. Again, such a term has negligible effects on the generic plane waves, but it does affect the winding modes. For example, consider the following winding mode

$$
\begin{aligned}
\hat{\phi}^{x(yz)} &= 0 \,, \\
\hat{\phi}^{y(zx)} &= -2\pi \frac{z}{\ell^z} W(x) \,, \\
\hat{\phi}^{z(xy)} &= 2\pi \frac{z}{\ell^z} W(x)
\end{aligned}
\tag{4.53}
$$

where $W(x)$ is an integer-valued function. If $W(x)$ is nonzero only at finitely many points, then the energy of this state is order $a$. The term (4.52) shifts this energy by $g/a \sim a$. Therefore, the energy of these states remain zero in the continuum limit $a \to 0$ (with fixed $\ell^i$). States with order $1/a$ nonzero $W(x)$ have energy of order one and they receive corrections of order one from terms like (4.52). Therefore, the computation of their energy using the original Lagrangian (4.5) is not universal. To sum up, while the zero-energy states are not lifted by these higher derivative terms, the finite energy states do receive quantitative corrections. Nonetheless, the qualitative features of these charged modes are universal. This is similar to the electric states in the $2+1$-dimensional $U(1)$ gauge theory of $A$ in [3]. In Section 5, we will see that this is also similar to the electric states of the $3+1$ dimensional $A$-theory, which is in fact dual to this theory (see Section 5.8).

# 5    The $A$ Tensor Gauge Theory

In this section we gauge the $(\mathbf{R}_{\text{time}}, \mathbf{R}_{\text{space}}) = (\mathbf{1}, \mathbf{3}')$ dipole global symmetry (2.31). We will focus on the pure gauge theory without matter, which is one of the gapless fracton models.

The gauge fields $(A_0, A_{ij})$ are in the $(\mathbf{1}, \mathbf{3}')$ representations of $S_4$. The gauge transformation is

$$A_0 \to A_0 + \partial_0 \alpha \,, \qquad A_{ij} \to A_{ij} + \partial_i \partial_j \alpha \,, \tag{5.1}$$

where $\alpha$ is a point-wise $2\pi$-periodic scalar. The gauge parameter $\alpha$ takes values in the same bundle as $\phi$ and requires nontrivial transition functions (see Section 3).

We define the gauge invariant electric and magnetic field strengths $E_{ij}$ and $B_{[ij]k}$ as

$$E_{ij} = \partial_0 A_{ij} - \partial_i \partial_j A_0 \,,$$
$$B_{[ij]k} = \partial_i A_{jk} - \partial_j A_{ik} \,,$$

(5.2)

which are in the $\mathbf{3'}$ and $\mathbf{2}$ of $S_4$, respectively.

Let space be a 3-torus with lengths $\ell^x, \ell^y, \ell^z$. Below, we will repeatedly consider a large gauge transformation of the form (3.6)

$$\alpha = 2\pi \left[ \frac{x}{\ell^x} \Theta(y - y_0) + \frac{y}{\ell^y} \Theta(x - x_0) - \frac{xy}{\ell^x \ell^y} \right]$$

(5.3)

which gives rise to the gauge transformation

$$A_{xy}(t, x, y, z) \sim A_{xy}(t, x, y, z) + 2\pi \left[ \frac{1}{\ell^x} \delta(y - y_0) + \frac{1}{\ell^y} \delta(x - x_0) - \frac{1}{\ell^x \ell^y} \right] \,.$$

(5.4)

## 5.1 Lattice Tensor Gauge Theory

Let us discuss the lattice version of the $U(1)$ tensor gauge theory of $A$ [17, 8, 18, 19, 9, 12]. Instead of simply reviewing these papers, we will present here a Euclidean lattice version of these systems.

We start with a Euclidean lattice and label the sites by integers $(\hat{\tau}, \hat{x}, \hat{y}, \hat{z})$. As in standard lattice gauge theory, the gauge transformations are $U(1)$ phases on the sites $\eta(\hat{\tau}, \hat{x}, \hat{y}, \hat{z}) = e^{i\alpha(\hat{\tau}, \hat{x}, \hat{y}, \hat{z})}$. The gauge fields are $U(1)$ phases placed on the (Euclidean) temporal links $U_\tau$ and on the spatial plaquettes $U_{xy}, U_{xz}, U_{yz}$. We also write $U_\tau = e^{iaA_\tau}$ and $U_{ij} = e^{ia^2 A_{ij}}$ where $a$ is the lattice spacing. It is clear that $U_\tau$ is in the trivial representation of the cubic group and the plaquette elements $U_{ij}$ are in $\mathbf{3'}$ – the two indices are symmetric rather than antisymmetric. Note that there are no diagonal components of the gauge fields $U_{xx}, U_{yy}, U_{zz}$ associated with the sites. This theory is sometimes called the "hollow rank-2 $U(1)$ gauge theory" [19].

The gauge transformations act on them as

$$U_\tau(\hat{\tau}, \hat{x}, \hat{y}, \hat{z}) \to U_\tau(\hat{\tau}, \hat{x}, \hat{y}, \hat{z}) \eta(\hat{\tau}, \hat{x}, \hat{y}, \hat{z}) \eta(\hat{\tau} + 1, \hat{x}, \hat{y}, \hat{z})^{-1} \,,$$
$$U_{xy}(\hat{\tau}, \hat{x}, \hat{y}, \hat{z}) \to U_{xy}(\hat{\tau}, \hat{x}, \hat{y}, \hat{z}) \eta(\hat{\tau}, \hat{x}, \hat{y}, \hat{z}) \eta(\hat{\tau}, \hat{x} + 1, \hat{y}, \hat{z})^{-1} \eta(\hat{\tau}, \hat{x} + 1, \hat{y} + 1, \hat{z}) \eta(\hat{\tau}, \hat{x}, \hat{y} + 1, \hat{z})^{-1} \,,$$

(5.5)

and similarly for $U_{xz}$ and $U_{yz}$. The Euclidean time-like links have standard gauge transformation rules and the plaquette elements are multiplied by the 4 phases around the plaquette.

The lattice action can include many gauge invariant terms. The simplest ones are asso-

ciated with cubes in the time-space-space directions and in the space-space-space directions

$$L_{xy\tau}(\hat{\tau}, \hat{x}, \hat{y}, \hat{z}) = U_\tau(\hat{\tau}, \hat{x}, \hat{y}, \hat{z})U_\tau(\hat{\tau}, \hat{x}+1, \hat{y}, \hat{z})^{-1}U_\tau(\hat{\tau}, \hat{x}+1, \hat{y}+1, \hat{z})U_\tau(\hat{\tau}, \hat{x}, \hat{y}+1, \hat{z})^{-1}$$
$$U_{xy}(\hat{\tau}, \hat{x}, \hat{y}, \hat{z})^{-1}U_{xy}(\hat{\tau}+1, \hat{x}, \hat{y}, \hat{z})$$
$$L_{[zx]y}(\hat{\tau}, \hat{x}, \hat{y}, \hat{z}) = U_{xy}(\hat{\tau}, \hat{x}, \hat{y}, \hat{z}+1)U_{xy}(\hat{\tau}, \hat{x}, \hat{y}, \hat{z})^{-1}U_{yz}(\hat{\tau}, \hat{x}+1, \hat{y}, \hat{z})^{-1}U_{yz}(\hat{\tau}, \hat{x}, \hat{y}, \hat{z})$$
$$(5.6)$$

and similarly for the other directions. Terms of the first kind, which involve the time direction are the analogs of the square of the electric field and terms of the second kind are analogs of the square of the magnetic field.

In addition to the local gauge-invariant operators (5.6), there are other non-local, extended ones. One example is a "strip" along the $x$ direction:

$$\prod_{\hat{x}=1}^{L^x} U_{xz}(\hat{\tau}, \hat{x}, \hat{y}, \hat{z}) , \qquad (5.7)$$

More generally, the strip (5.7) can be made out of plaquettes extending between $\hat{z}$ and $\hat{z}+1$ and zigzagging along a path on the $xy$-plane. Similar operators exist using the other directions.

In the Hamiltonian formulation, we choose the temporal gauge to set all the $U_\tau$'s to 1. We introduce the electric field $E_p$ such that $\frac{2}{g_e^2}E_p$ is conjugate to the phase of the plaquette $U_p$, where $g_e$ is the electric coupling constant. $\frac{2}{g_e^2}E_p$ has integer eigenvalues. This definition of the lattice electric field differs from the continuum definition by a power of the lattice spacing, which can be added easily on dimensional grounds.

Gauss law is imposed as an operator equation

$$G(\hat{x}, \hat{y}, \hat{z}) = \sum_{p \ni (\hat{x}, \hat{y}, \hat{z})} \epsilon_p E_p = 0 \qquad (5.8)$$

where the sum is an oriented sum ($\epsilon_p = \pm 1$) over the 12 plaquettes $p$ that share a common site $(\hat{x}, \hat{y}, \hat{z})$.

One example of such a Hamiltonian is

$$H = \frac{1}{g_e^2} \sum_{\text{plaquettes}} E_p^2 + \frac{1}{g_m^2} \sum_{\text{cubes}} (L_{[xy]z} + L_{[yz]x} + L_{[zx]y} + c.c) . \qquad (5.9)$$

Instead of imposing Gauss law as an operator equation, we can alternatively impose it energetically by adding a term $\sum_{\text{sites}} G^2$ to the Hamiltonian.

The lattice model has an electric tensor symmetry whose conserved charge is proportional

to

$$\sum_{\hat{z}=1}^{L^z} E_{xy}(\hat{x}_0, \hat{y}_0, \hat{z}) \,, \tag{5.10}$$

for each point $(\hat{x}_0, \hat{y}_0)$ on the $xy$-plane. There are similar charges along the other directions. This charge commutes with the Hamiltonian. The electric tensor symmetry rotates the phases of the plaquette variables $U_{xy}$ at $(\hat{x}_0, \hat{y}_0)$ for all $\hat{z}$. Using Gauss law (5.8), the dependence of the conserved charge on $p$ is a function of $\hat{x}_0$ plus a function of $\hat{y}_0$.

## 5.2   Continuum Lagrangian

The Lagrangian for the pure tensor gauge theory without matter is

$$\mathcal{L} = \frac{1}{2g_e^2} E_{ij} E^{ij} - \frac{1}{2g_m^2} B_{[ij]k} B^{[ij]k} \,. \tag{5.11}$$

Note that the coupling constants $g_e, g_m$ have mass dimension 1. The equations of motion are

$$\frac{1}{g_e^2} \partial_0 E_{ij} = \frac{1}{g_m^2} \partial^k (B_{[ki]j} + B_{[kj]i}) \,,$$
$$\partial_i \partial_j E^{ij} = 0 \,, \tag{5.12}$$

where the second equation is Gauss law.

From the definition (5.2) of the electric and magnetic fields, we have

$$\partial_0 B_{[ki]j} = \partial_k E_{ij} - \partial_i E_{kj} \,, \tag{5.13}$$

which is analogous to the Bianchi identity in standard gauge theories.

## 5.3   Fluxes

Let us put the theory on a Euclidean 4-torus with lengths $\ell^\tau, \ell^x, \ell^y, \ell^z$. Consider gauge field configurations with a nontrivial transition function at $\tau = \ell^\tau$:

$$g_{(\tau)} = 2\pi \left[ \frac{x}{\ell^x} \Theta(y - y_0) + \frac{y}{\ell^y} \Theta(x - x_0) - \frac{xy}{\ell^x \ell^y} \right] \,. \tag{5.14}$$

We have $A_{xy}(\tau + \ell^\tau, x, y, z) = A_{xy}(\tau, x, y, z) + \partial_x \partial_y g_{(\tau)}$. Such configurations have nontrivial, quantized electric fluxes

$$e_{xy}(x_1, x_2) \equiv \oint d\tau \int_{x_1}^{x_2} dx \oint dy E_{xy} \in 2\pi\mathbb{Z} \,. \tag{5.15}$$

In particular, the flux can be nontrivial when the integral is over the whole $(\tau, x, y)$ spacetime. The Bianchi identity (5.13) implies that

$$\partial_z e_{xy}(x_1, x_2) = 0. \tag{5.16}$$

Therefore, the flux $e_{xy}$ only depends on $x_1, x_2$.

The magnetic flux is realized in a bundle with transition functions $g_{(x)} = 0$ at $x = \ell^x$, $g_{(y)} = 0$ at $y = \ell^y$, and

$$g_{(z)} = 2\pi \left[ \frac{y}{\ell^y} \Theta(x - x_0) + \frac{x}{\ell^x} \Theta(y - y_0) - \frac{xy}{\ell^x \ell^y} \right] \tag{5.17}$$

at $z = \ell^z$. This means that

$$A_{ij}(\tau, x, y, z = \ell^z) = A_{ij}(\tau, x, y, z = 0) + \partial_i \partial_j g_{(z)} \tag{5.18}$$

and $A_{ij}$ periodic around the other directions. The only nonperiodic boundary condition is

$$A_{xy}(\tau, x, y, z = \ell^z) = A_{xy}(\tau, x, y, z = 0) + 2\pi \left[ \frac{1}{\ell^y} \delta(x - x_0) + \frac{1}{\ell^x} \delta(y - y_0) - \frac{1}{\ell^x \ell^y} \right] \tag{5.19}$$

and therefore

$$\oint dz dx B_{[zx]y} = 2\pi \oint dx \left[ \frac{1}{\ell^y} \delta(x - x_0) + \frac{1}{\ell^x} \delta(y - y_0) - \frac{1}{\ell^x \ell^y} \right] = 2\pi \delta(y - y_0) \,,$$

$$\oint dy dz B_{[yz]x} = -2\pi \oint dy \left[ \frac{1}{\ell^y} \delta(x - x_0) + \frac{1}{\ell^x} \delta(y - y_0) - \frac{1}{\ell^x \ell^y} \right] = -2\pi \delta(x - x_0) \,, \tag{5.20}$$

$$\oint dx dy B_{[xy]z} = 0 \,.$$

By taking linear combinations of similar bundles with transition functions in other directions, we realize the more general magnetic flux

$$b_{[yz]x}(x_1, x_2) \equiv \int_{x_1}^{x_2} dx \oint dy \oint dz \, B_{[yz]x} \in 2\pi \mathbb{Z} \,, \tag{5.21}$$

and similarly for the other components of the magnetic field. In particular, the flux can be nontrivial when integrated over the whole space $(x, y, z)$. The Bianchi identity (5.13) implies that

$$\partial_\tau b_{[yz]x}(x_1, x_2) = 0 \,. \tag{5.22}$$

Therefore, the flux $b_{[yz]x}$ depends only on $x_1, x_2$. It is conserved.

The magnetic symmetry is absent on the lattice. However, the flux quantization of the continuum theory can be traced to the lattice. It is associated with products of observables that are constrained to be one on the lattice and the quantized value in the continuum arises from writing them as $e^{2\pi i n}$. It is crucial that the integer $n$ is meaningful in the continuum. The magnetic flux (5.21) corresponds to the product $\prod_{\hat{y},\hat{z}} L_{[yz]x} = 1$ on the lattice. Similarly, the electric flux (5.15) corresponds to the product $\prod_{\hat{\tau},\hat{y}} L_{xy\tau} = 1$ on the lattice.

## 5.4 Global Symmetries and Their Charges

We now discuss the global symmetries of this continuum tensor gauge theory.

### 5.4.1 Electric Tensor Symmetry

Let us define a current with $(\mathbf{R}_{\text{time}}, \mathbf{R}_{\text{space}}) = (\mathbf{3}', \mathbf{2})$ as

$$
\begin{aligned}
J_0^{ij} &= \frac{2}{g_e^2} E^{ij} \,, \\
J^{[ki]j} &= \frac{2}{g_m^2} B^{[ki]j} \,.
\end{aligned}
\tag{5.23}
$$

The equations of motion for $A_{ij}$ and $A_0$ (5.12) are recognized as the conservation equation (2.23) and the differential condition (2.24) for the $(\mathbf{3}', \mathbf{2})$ tensor global symmetry, respectively. The symmetry generated by (5.23) will be called the *electric tensor symmetry*.

The conserved charge for the electric tensor global symmetry is

$$
Q^k(x^i, x^j) = \frac{2}{g_e^2} \oint dx^k \, E_{ij}
\tag{5.24}
$$

and the symmetry operator is

$$
\mathcal{U}^k(\beta; x^i, x^j) = \exp\left[ i\beta \, Q^k(x^i, x^j) \right] = \exp\left[ i\frac{2\beta}{g_e^2} \oint dx^k E_{ij} \right] .
\tag{5.25}
$$

Naively, the charge generates $A_{ij} \to A_{ij} + c(x^i, x^j)$, but combining it with a gauge transformation $A_{ij} \to A_{ij} + \partial_i \partial_j \alpha$, we can let it generate

$$
A_{ij} \to A_{ij} + c_{ij}^i(x^i) + c_{ij}^j(x^j) \,.
\tag{5.26}
$$

The electric tensor global symmetry maps one configuration of $A_{ij}$ to another with the same electric and magnetic field strengths. This is similar to the electric one-form global symmetry

in the $U(1)$ Maxwell theory, which shifts the gauge field by a flat $U(1)$ connection [33].

The charged objects under the electric tensor symmetry are the gauge-invariant strip operators

$$W(z_1, z_2, \mathcal{C}^{xy}) = \exp\left[ i \int_{z_1}^{z_2} dz \oint_{\mathcal{C}^{xy}} (dx\, A_{xz} + dy\, A_{yz}) \right] , \qquad (5.27)$$

where $\mathcal{C}^{xy}$ is a closed curve in the $xy$-plane. This is the continuum version of the gauge-invariant operator (5.7) on the lattice. We will refer to this operator as the Wilson strip. Under the gauge transformation (5.4), only integer powers of the Wilson strip are gauge invariant. Similarly we define $W(x_1^k, x_2^k, \mathcal{C}^{ij})$ for the other directions with $\mathcal{C}^{ij}$ a curve on the $ij$-plane. (Recall our convention, $i \neq j \neq k$.)

At a fixed time, the line operator $\mathcal{U}^x(\beta; y_0, z_0)$ and the strip operator obey the equal-time commutation relation

$$\mathcal{U}^x(\beta; y_0, z_0)\, W(z_1, z_2, \mathcal{C}^{xy}) = e^{i\beta\, I(\mathcal{C}^{xy}, y_0)}\, W(z_1, z_2, \mathcal{C}^{xy})\, \mathcal{U}^x(\beta; y_0, z_0)\,, \qquad \text{if} \ \ z_1 < z_0 < z_2\,,$$
$$(5.28)$$

and they commute otherwise. Here $I(\mathcal{C}^{xy}, y_0)$ is the intersection number between the curve $\mathcal{C}^{xy}$ and the $y = y_0$ line on the $xy$-plane. The exponent $\beta$ is $2\pi$-periodic, since the charged objects have integral charges. This means that the global structure of the electric tensor global symmetry is $U(1)$ rather than $\mathbb{R}$. Similar commutation relations hold true for $\mathcal{U}$ and $W$ in the other directions.

### 5.4.2 Magnetic Tensor Symmetry

Let us define

$$\begin{aligned} J_0^{[ij]k} &= \frac{1}{2\pi} B^{[ij]k}\,, \\ J^{ij} &= \frac{1}{2\pi} E^{ij}\,. \end{aligned} \qquad (5.29)$$

Then the Bianchi identity (5.13) is recognized as the conservation equation (2.20) for the $(\mathbf{2}, \mathbf{3}')$ tensor global symmetry. We will refer to this symmetry as the *magnetic tensor symmetry*.

While the continuum theory has both the electric and magnetic tensor global symmetries, the latter is absent on the lattice.

The conserved charge operator for the magnetic tensor global symmetry is

$$Q^{[ij]}(x^k) = \frac{1}{2\pi} \oint dx^i \oint dx^j\, B^{[ij]k}\,, \qquad (\text{no sum in } i, j)\,. \qquad (5.30)$$

The symmetry operator is

$$\mathcal{U}^{ij}(\beta; x_1^k, x_2^k) = \exp\left[ i\frac{\beta}{2\pi} \int_{x_1^k}^{x_2^k} dx^k \oint dx^i \oint dx^j \, B^{[ij]k} \right] \qquad \text{(no sum in } i, j, k\text{)}. \qquad (5.31)$$

It is a "slab" of width $x_2^k - x_1^k$, which extends along the $i, j$ directions.

The magnetically charged objects under the magnetic tensor global symmetry are point-like monopole operators. The monopole operator $e^{i\hat{\phi}^{k(ij)}}$ can be written in terms of the dual field $\hat{\phi}^{k(ij)}$. See Section 5.8.

## 5.5    Defects as Fractons

Having discussed various extended operators defined at a fixed time, we now turn to observables that also extend in the time direction, i.e., defects. In the $U(1)$ tensor gauge theory where Gauss law is imposed as an operator equation, there is no dynamical charged particle in the spectrum. The defects capture in the low energy theory the physics of probe charged particles that are infinitely heavy. Constraints about the motion of particles are captured by constraints on the spacetime trajectories of the defects. Here these constraints arise from the gauge symmetry. In particular, we will see that the defects exhibit the characteristic behaviors of fractons.

The simplest defect is a single particle of gauge charge $+1$ at a fixed point in space $(x, y, z)$. It is captured by the gauge-invariant defect

$$\exp\left[ i \int_{-\infty}^{\infty} dt \, A_0(t, x, y, z) \right] . \qquad (5.32)$$

Importantly, a single particle cannot move in space by itself – it is immobile – because of gauge invariance. This is the hallmark of a fracton.

While a single particle cannot move in isolation, a pair of them with opposite charges – a dipole – can move collectively. Consider two particles with charges $\pm 1$ at fixed $x_1$ and $x_2$ moving in time along a curve $\mathcal{C}$ in the $(y, z, t)$ spacetime. This motion is described by the gauge-invariant defect

$$W(x_1, x_2, \mathcal{C}) = \exp\left[ i \int_{x_1}^{x_2} dx \int_{\mathcal{C}} (\, dt\partial_x A_0 + dy A_{xy} + dz A_{xz} \,) \right] \qquad (5.33)$$

Note that the integrand $\int_{\mathcal{C}} (\, dt\partial_x A_0 + dy A_{xy} + dz A_{xz} \,)$ is gauge-invariant for any curve $\mathcal{C}$ without endpoints, e.g. running from the far past to the far future. More generally, we can

have a pair of particles moving in directions transverse to their separation. The operators (5.27) are special cases of these defects where $\mathcal{C}$ is a closed curve independent of time.

By combining two defects of the form (5.33), one separated in the $x$ direction and the other in the $y$ direction, we can have two particles with charges $\pm 1$ at $(x_1, y_1)$ and $(x_2, y_2)$ moving together along the $z$ direction. They are represented by the defect

$$\exp\left[i \int_{\text{strip}} (\partial_x A_0 \, dxdt + \partial_y A_0 \, dydt + A_{yz} \, dydz + A_{xz} \, dxdz)\right] \tag{5.34}$$

where the strip is a direct product of line segments $\mathcal{C}$ between $(x_1, y_1)$ to $(x_2, y_2)$ on the $xy$-plane and a curve $z(t)$ on the $zt$-plane. More generally, by combining more defects of the kind (5.33), the line segments $\mathcal{C}$ can be replaced by a continuous curve extending from $(x_1, y_1)$ to $(x_2, y_2)$ on the $xy$-plane.

Finally, while a single fracton cannot move by itself, it can move at the price of creating several more fractons. For example, consider the following defect

$$\exp\left[i \int_{-\infty}^{\tau} dt \, A_0(t, x_0, y_0, z_0)\right]$$
$$\times \exp\left[-i \int_{x_0}^{x_1} \int_{y_0}^{y_1} dxdy \, A_{xy}(\tau, x, y, z_0)\right]$$
$$\times \exp\left[i \int_{\tau}^{\infty} dt \, A_0(t, x_1, y_0, z_0)\right] \exp\left[i \int_{\tau}^{\infty} dt \, A_0(t, x_0, y_1, z_0)\right] \exp\left[-i \int_{\tau}^{\infty} dt \, A_0(t, x_1, y_1, z_0)\right] .$$
$$\tag{5.35}$$

The defect in the first line represents a single fracton of charge $+1$ as $(x_0, y_0, z_0)$. Then, at time $\tau$ it is acted upon by the an operator, written in the second line. The result is three fractons; two charge $+1$ fractons at $(x_1, y_0, z_0)$ and $(x_0, y_1, z_0)$ and a charge $-1$ fracton at $(x_1, y_1, z_0)$. Their motion is described by the defect in the third line.

## 5.6 Electric Modes

In this section we analyze the perturbative spectrum of the theory.

Let us consider plane wave mode in $\mathbb{R}^{3,1}$ in the temporal gauge $A_0 = 0$:

$$A_{ij} = C_{ij} \, e^{i\omega t + i k_i x^i} , \tag{5.36}$$

with constant $C_{ij}$ in the $\mathbf{3'}$. The equations of motion give the dispersion relation [17, 12]

$$\omega^2 \left[ \frac{g_m^4}{g_e^4} \omega^4 - 2 \frac{g_m^2}{g_e^2} \omega^2 (k_x^2 + k_y^2 + k_z^2) + 3(k_x^2 k_y^2 + k_x^2 k_z^2 + k_y^2 k_z^2) \right] = 0 , \tag{5.37}$$

There are three solutions for $\omega^2$:

$$\omega_\pm^2 = \frac{g_e^2}{g_m^2}\left[(k_x^2 + k_y^2 + k_z^2) \pm \sqrt{(k_x^2 + k_y^2 + k_z^2)^2 - 3(k_x^2 k_y^2 + k_x^2 k_z^2 + k_y^2 k_z^2)}\right],$$

$$\omega_0^2 = 0.$$

(5.38)

For generic $k_i$, the $\omega_0 = 0$ solution can be gauged away by a residual, time-independent gauge transformation with $\alpha \sim e^{ik_x x + ik_y y + ik_z z}$ and it should not be considered physical. The other solutions with generic $k_i$ lead to a Fock space of states – "photons."

The situation is more subtle as we take some of the $k_i$s to zero. For example, consider plane waves with $k_x = k_y = 0$. The equations of motion reduce to

$$\frac{g_m^2}{g_e^2} \partial_0^2 A_{xy} = 2\partial_z^2 A_{xy}, \qquad \partial_0^2 A_{yz} = \partial_0^2 A_{xz} = 0.$$

(5.39)

Restricting to the zero-energy solution $\omega = 0$, we find *two* independent plane wave solutions with arbitrary $C_{yz}$ and $C_{xz}$. Equivalently, in position space, there are two families of solutions that are independent of $x, y$:

$$A_{xy} = 0, \qquad A_{yz} = F_{yz}^z(z), \qquad A_{xz} = F_{xz}^z(z),$$

(5.40)

for any functions $F_{yz}^z(z)$ and $F_{xz}^z(z)$. They can be thought of as the $k_x, k_y \to 0$ limit of the $\omega_-$ solution and the $\omega_0$ solution. However, when $k_x = k_y = 0$, neither solution (5.40) can be gauged away by a residual, time-independent gauge transformation (with finite support in $\mathbb{R}^3$).

Similarly, we have two families of zero-energy solutions for each of the $x$ and $y$ directions. All in all, we have six zero-energy solutions $F_{xy}^x(x), F_{xy}^y(y), F_{yz}^y(y), F_{yz}^z(z), F_{xz}^x(x), F_{xz}^z(z)$, each a function of one spatial coordinate.

These zero-energy solutions are a consequence of the electric tensor global symmetry (5.26), which maps one solution to another, while leaving the electric and magnetic fields invariant. For this reason we will refer to these modes as the electric modes.

We now quantize these classically zero-energy configurations on a spatial 3-torus with lengths $\ell^x, \ell^y, \ell^z$. For later convenience, we will normalize these modes as

$$A_{ij} = \frac{1}{\ell^j} f_{ij}^i(x^i) + \frac{1}{\ell^i} f_{ij}^j(x^j).$$

(5.41)

Let us focus on $A_{xy}(t, x, y, z) = \frac{1}{\ell^y} f_{xy}^x(t, x) + \frac{1}{\ell^x} f_{xy}^y(t, y)$. The quantization of the other 4 functions $f_{ij}^i$ can be done in parallel.

The quantization of these modes proceeds as in the $2+1$-dimensional tensor gauge theory $A$ (1.4). See Section 6.6 of [3]. In the end, the Hamiltonian for these modes is

$$H = \frac{g_e^2}{4\ell^z} \left[ \ell^y \oint dx (\bar{\Pi}_{xy}^x)^2 + \ell^x \oint dy (\bar{\Pi}_{xy}^y)^2 + 2 \oint dx \, \bar{\Pi}_{xy}^x \oint dy \, \bar{\Pi}_{xy}^y \right] , \qquad (5.42)$$

where $\bar{\Pi}_{xy}^x(x), \bar{\Pi}_{xy}^y(y)$ are the conjugate momenta.[9] They have integer eigenvalues, $\bar{\Pi}_{xy}^x(x), \bar{\Pi}_{xy}^y(y) \in \mathbb{Z}$ at each point $x$ and $y$. Furthermore, they are subject to an ambiguity

$$\left( \bar{\Pi}_{xy}^x(x) , \bar{\Pi}_{xy}^y(y) \right) \sim \left( \bar{\Pi}_{xy}^x(x) + 1 , \bar{\Pi}_{xy}^y(y) - 1 \right) . \qquad (5.44)$$

In fact, the charge of the electric tensor symmetry (5.24) is the sum of $\bar{\Pi}$'s

$$Q^z(x, y) = \frac{2}{g_e^2} \oint dz E_{xy} = \bar{\Pi}_{xy}^x(x) + \bar{\Pi}_{xy}^y(y) . \qquad (5.45)$$

Including the charges from the other directions, we have $2L^x + 2L^y + 2L^z - 3$ such charges on a lattice.

Let us discuss the energy of these modes. Since $\bar{\Pi}_{xy}^i(x^i)$ have independent integer eigenvalues at each point $x^i$, a generic electric mode has energy order $a$, which goes to zero in the continuum limit. This is similar to the electric modes of the tensor gauge theory (1.4) in $2 + 1$ dimensions [3].

## 5.7 Magnetic Modes

In this subsection we explore gauge field configurations in nontrivial bundles characterized by transition functions $g_{(i)}$. These configurations realize the magnetic tensor symmetry charges (5.30).

### Minimally Charged States

The simplest nontrivial bundle with minimal magnetic tensor symmetry charges is characterized by the transition function in (5.17). Let us find the lowest energy configuration in

---

[9]More precisely, $\bar{\Pi}_{xy}^x(x), \bar{\Pi}_{xy}^y(y)$ are the conjugate momenta for

$$\bar{f}_{xy}^x(t, x) = f_{xy}^x(t, x) + \frac{1}{\ell^x} \oint dy f_{xy}^y(t, y) , \quad \bar{f}_{xy}^y(t, y) = f_{xy}^y(t, y) + \frac{1}{\ell^y} \oint dx f_{xy}^x(t, x) . \qquad (5.43)$$

See [3] for more details.

this bundle. We start with a simple example of a gauge field in this bundle

$$A_{xy} = 2\pi \frac{z}{\ell z} \left[ \frac{1}{\ell y} \delta(x - x_0) + \frac{1}{\ell x} \delta(y - y_0) - \frac{1}{\ell x \ell y} \right] ,$$

$$A_{yz} = A_{xz} = 0 .$$

(5.46)

Its magnetic field is

$$B_{[zx]y} = -B_{[yz]x} = \frac{2\pi}{\ell z} \left[ \frac{1}{\ell y} \delta(x - x_0) + \frac{1}{\ell x} \delta(y - y_0) - \frac{1}{\ell x \ell y} \right] , \quad B_{[xy]z} = 0 .$$

(5.47)

which realizes one unit of the $(\mathbf{2}, \mathbf{3}')$ tensor global symmetry charge (5.30). Its energy is

$$\frac{1}{g_m^2} \oint dx dy dz \left( B_{[zx]y}^2 + B_{[yz]x}^2 + B_{[xy]z}^2 \right) = \frac{8\pi^2}{g_m^2} \frac{1}{\ell x \ell y \ell z} \left[ (\ell^x + \ell^y) \delta(0) - 1 \right] .$$

(5.48)

Every other configuration in this bundle can be written as a sum of (5.46) and another gauge field in the trivial bundle $a_{ij}$:

$$A_{xy} = 2\pi \frac{z}{\ell z} \left[ \frac{1}{\ell y} \delta(x - x_0) + \frac{1}{\ell x} \delta(y - y_0) - \frac{1}{\ell x \ell y} \right] + a_{xy} ,$$

$$A_{xz} = a_{xz} , \quad A_{yz} = a_{yz} .$$

(5.49)

The energy of this configuration is

$$\frac{1}{g_m^2} \oint dx dy dz \left[ (\partial_x a_{yz} - \partial_y a_{xz})^2 + \left( \partial_y a_{xz} - \frac{2\pi}{\ell z} \left( \frac{1}{\ell y} \delta(x - x_0) + \frac{1}{\ell x} \delta(y - y_0) - \frac{1}{\ell x \ell y} \right) \right)^2 \right.$$

$$\left. + \left( \partial_x a_{yz} - \frac{2\pi}{\ell z} \left( \frac{1}{\ell y} \delta(x - x_0) + \frac{1}{\ell x} \delta(y - y_0) - \frac{1}{\ell x \ell y} \right) \right)^2 \right] ,$$

(5.50)

where we assumed that at the minimum $a_{ij}$ are independent of $z$. The minimization of this energy is determined by the equation of motion for $a_{ij}$

$$\partial_x \left[ 2\partial_x a_{yz} - \partial_y a_{xz} - \frac{2\pi}{\ell z} \left( \frac{1}{\ell y} \delta(x - x_0) + \frac{1}{\ell x} \delta(y - y_0) - \frac{1}{\ell x \ell y} \right) \right] = 0 ,$$

$$\partial_y \left[ 2\partial_y a_{xz} - \partial_x a_{yz} - \frac{2\pi}{\ell z} \left( \frac{1}{\ell y} \delta(x - x_0) + \frac{1}{\ell x} \delta(y - y_0) - \frac{1}{\ell x \ell y} \right) \right] = 0 .$$

(5.51)

This is solved by

$$
\begin{aligned}
a_{yz} &= \frac{\pi}{\ell^z \ell^y}\left[\Theta(x-x_0) - \frac{x}{\ell^x}\right] + f^y_{yz}(y)\,, \\
a_{xz} &= \frac{\pi}{\ell^z \ell^x}\left[\Theta(y-y_0) - \frac{y}{\ell^y}\right] + f^x_{xz}(x)\,,
\end{aligned}
\tag{5.52}
$$

where $f^x_{xz}(x)$ and $f^y_{yz}(y)$ are two periodic functions that can be absorbed into the electric modes that we have already quantized in Section 5.6.

We conclude that up to a gauge transformation and additive zero energy configurations, the minimum energy configuration in this bundle is

$$
\begin{aligned}
A_{xy} &= 2\pi\frac{z}{\ell^z}\left[\frac{1}{\ell^y}\delta(x-x_0) + \frac{1}{\ell^x}\delta(y-y_0) - \frac{1}{\ell^x\ell^y}\right]\,, \\
A_{xz} &= \frac{\pi}{\ell^z\ell^x}\left[\Theta(y-y_0) - \frac{y}{\ell^y}\right]\,, \\
A_{yz} &= \frac{\pi}{\ell^z\ell^y}\left[\Theta(x-x_0) - \frac{x}{\ell^x}\right]\,.
\end{aligned}
\tag{5.53}
$$

Its energy is

$$
\frac{6\pi^2}{g_m^2}\frac{1}{\ell^x\ell^y\ell^z}\left[(\ell^x + \ell^y)\delta(0) - \frac{2}{3}\right]\,,
\tag{5.54}
$$

which is indeed smaller than the energy (5.48).

Note that the energy of this magnetic mode is of order $\frac{1}{a}$ and diverges in the continuum limit.

## General Charged States

Next, we consider linear combinations of the configurations in (5.53) with those in the other directions:

$$
\begin{aligned}
A_{ij} &= 2\pi\frac{x^k}{\ell^k}\left[\frac{1}{\ell^j}\sum_\alpha W^{ij}_{i\alpha}\delta(x^i - x^i_\alpha) + \frac{1}{\ell^i}\sum_\beta W^{ij}_{j\beta}\delta(x^j - x^j_\beta) - \frac{W^{ij}}{\ell^i\ell^j}\right] \\
&\quad + \frac{\pi}{\ell^i\ell^j}\left[\sum_\gamma W^{jk}_{k\gamma}\Theta(x^k - x^k_\gamma) - W^{jk}\frac{x^k}{\ell^k}\right] + \frac{\pi}{\ell^j\ell^i}\left[\sum_\gamma W^{ik}_{k\gamma}\Theta(x^k - x^k_\gamma) - W^{ik}\frac{x^k}{\ell^k}\right]\,, \\
W^{ij} &= \sum_\alpha W^{ij}_{i\alpha} = \sum_\beta W^{ij}_{j\beta}\,.
\end{aligned}
\tag{5.55}
$$

The transition function $g_{(k)}$ as we go along the $x^k$ direction is

$$g_{(k)} = 2\pi \left[ \sum_\alpha \frac{x^j}{\ell^j} W^{ij}_{i\alpha} \Theta(x^i - x^i_\alpha) + \sum_\beta \frac{x^i}{\ell^i} W^{ij}_{j\beta} \Theta(x^j - x^j_\beta) - W^{ij} \frac{x^i x^j}{\ell^i \ell^j} \right],$$

$$W^{ij} = \sum_\alpha W^{ij}_{i\alpha} = \sum_\beta W^{ij}_{j\beta}.$$

(5.56)

Not all these bundles are inequivalent. Consider a gauge transformation

$$\alpha(t, x, y, z) = 2\pi \frac{xy}{\ell^x \ell^y} \sum_\gamma w^z_\gamma \Theta(z - z_\gamma) + 2\pi \frac{xz}{\ell^x \ell^z} \sum_\beta w^y_\beta \Theta(y - y_\beta)$$

$$+ 2\pi \frac{yz}{\ell^y \ell^z} \sum_\alpha w^x_\alpha \Theta(x - x_\alpha) - 4\pi w \frac{xyz}{\ell^x \ell^y \ell^z},$$

(5.57)

with $w \equiv \sum_\alpha w^x_\alpha = \sum_\beta w^y_\beta = \sum_\gamma w^z_\gamma$ and all the $w$'s are integers. This gauge parameter does not have the appropriate transition functions discussed in Section 3. Rather it changes the transition functions by shifting the $W$'s by

$$W^{xy}_{x\alpha} \to W^{xy}_{x\alpha} + w^x_\alpha, \qquad W^{xy}_{y\beta} \to W^{xy}_{y\beta} + w^y_\beta,$$
$$W^{xz}_{x\alpha} \to W^{xz}_{x\alpha} + w^x_\alpha, \qquad W^{xz}_{z\gamma} \to W^{xz}_{z\gamma} + w^z_\gamma,$$
$$W^{yz}_{y\beta} \to W^{yz}_{y\beta} + w^y_\beta, \qquad W^{yz}_{z\gamma} \to W^{yz}_{z\gamma} + w^z_\gamma.$$

(5.58)

Hence two sets of $W$'s label the same bundle if they are related by (5.58).

The underlying lattice theory does not have the magnetic symmetry and does not have well-defined such bundles. These bundles and the corresponding symmetry are present only in the continuum theory. Yet, we can consider the points $x^i_\alpha$ to be chosen from a lattice with $L^i$ sites in the $x^i$ direction. Then, we have $2L^x + 2L^y + 2L^z - 3$ integers $W$'s, and $L^x + L^y + L^z - 2$ integer $w$'s. Therefore the number of distinct bundles is $L^x + L^y + L^z - 1$.

The magnetic field of (5.55) is

$$B_{[ij]k} = \frac{\pi}{\ell^i} \left[ \frac{1}{\ell^k} \sum_\beta W^{jk}_{j\beta} \delta(x^j - x^j_\beta) + \frac{2}{\ell^j} \sum_\gamma W^{jk}_{k\gamma} \delta(x^k - x^k_\gamma) - \frac{W^{jk}}{\ell^j \ell^k} \right]$$

$$- \frac{\pi}{\ell^j} \left[ \frac{1}{\ell^k} \sum_\alpha W^{ik}_{i\alpha} \delta(x^i - x^i_\alpha) + \frac{2}{\ell^i} \sum_\gamma W^{ik}_{k\gamma} \delta(x^k - x^k_\gamma) - \frac{W^{ik}}{\ell^i \ell^k} \right]$$

$$+ \frac{\pi}{\ell^k} \left[ \frac{1}{\ell^j} \sum_\alpha W^{ij}_{i\alpha} \delta(x^i - x^i_\alpha) - \frac{1}{\ell^i} \sum_\beta W^{ij}_{j\beta} \delta(x^j - x^j_\beta) \right]$$

(5.59)

The magnetic tensor symmetry charge is

$$
\begin{aligned}
Q^{[ij]}(x^k) = \frac{1}{2\pi} \oint dx^i \oint dx^j B_{[ij]k} &= \sum_\gamma \left( W^{jk}_{k\,\gamma} - W^{ik}_{k\,\gamma} \right) \delta(x^k - x^k_\gamma) \\
&\equiv -\sum_\gamma W_{k\,\gamma} \delta(x^k - x^k_\gamma) \,,
\end{aligned}
\tag{5.60}
$$

where we have defined $W_{k\gamma} \equiv W^{ik}_{k\,\gamma} - W^{jk}_{k\,\gamma}$ with $i,j,k$ cyclically ordered. The minimal energy with these charges is

$$
\begin{aligned}
H &= \frac{6\pi^2}{g_m^2 \ell^x \ell^y \ell^z} \sum_i \left[ \ell^i \oint dx^i \left( Q^{[jk]} \right)^2 - \frac{1}{3} \left( \oint dx^i Q^{[jk]} \right)^2 \right] \\
&= \frac{6\pi^2}{g_m^2 \ell^x \ell^y \ell^z} \left[ \delta(0) \left( \ell^x \sum_\alpha W_{x\,\alpha}^2 + \ell^y \sum_\beta W_{y\,\beta}^2 + \ell^z \sum_\gamma W_{z\,\gamma}^2 \right) \right. \\
&\qquad \left. - \frac{1}{3} \left( \left( \sum_\alpha W_{x\,\alpha} \right)^2 + \left( \sum_\beta W_{y\,\beta} \right)^2 + \left( \sum_\gamma W_{z\,\gamma} \right)^2 \right) \right] \,.
\end{aligned}
\tag{5.61}
$$

## 5.8   Duality Transformation

In this subsection we perform a duality transformation on the tensor gauge theory of $A$. We will arrive at the $\hat{\phi}$ theory of Section 4. This duality is similar to the duality between an ordinary $2+1$-dimensional gauge field $A_\mu$ and a compact real scalar $\varphi$.

The duality we present below is a continuum duality. It is related to the lattice duality in [17] in the same way as the continuum T-duality of the compact scalar in $1+1$ dimensions is related to the duality of the lattice $1+1$-dimensional XY-model. Our dual field $\hat{\phi}$ is circle-valued rather than an integer on the lattice. Also, $\hat{\phi}$ is in the two-dimensional representation of $S_4$ and hence the sum of its three components vanishes, while in the lattice version, the three components are subject to a gauge identification.

We work in Euclidean signature and denote the Euclidean time as $\tau$. We start with the Euclidean Lagrangian

$$
\begin{aligned}
\mathcal{L}_E &= \frac{1}{2g_e^2} E_{ij} E^{ij} + \frac{1}{2g_m^2} B_{[ij]k} B^{[ij]k} \\
&+ \frac{i}{2(2\pi)} \widetilde{B}^{ij} \left( \partial_\tau A_{ij} - \partial_i \partial_j A_\tau - E_{ij} \right) + \frac{i}{2(2\pi)} \widetilde{E}^{[ij]k} \left( \partial_i A_{jk} - \partial_j A_{ik} - B_{[ij]k} \right) \,,
\end{aligned}
\tag{5.62}
$$

where $E_{ij}, B_{[ij]k}, \widetilde{B}_{ij}, \widetilde{E}_{[ij]k}$ are independent fields in the appropriate representation of $S_4$.

They are not constrained by any differential condition. If we integrate out these fields, we find the original Lagrangian in terms of $(A_\tau, A_{ij})$.

Instead, we integrate out only $E_{ij}$ and $B_{[ij]k}$ to obtain $E^{ij} = \frac{ig_e^2}{4\pi}\widetilde{B}^{ij}$ and $B_{[ij]k} = \frac{ig_m^2}{4\pi}\widetilde{E}_{[ij]k}$. The Lagrangian then becomes

$$\mathcal{L}_E = \frac{g_e^2}{32\pi^2}\widetilde{B}_{ij}\widetilde{B}^{ij} + \frac{g_m^2}{32\pi^2}\widetilde{E}_{[ij]k}\widetilde{E}^{[ij]k} + \frac{i}{2(2\pi)}\widetilde{B}^{ij}\left(\partial_\tau A_{ij} - \partial_i\partial_j A_\tau\right) + \frac{i}{2(2\pi)}\widetilde{E}^{[ij]k}\left(\partial_i A_{jk} - \partial_j A_{ik}\right).$$
(5.63)

Next, we integrate out the original gauge fields $(A_\tau, A_{ij})$ to find the constraints

$$\partial_\tau\widetilde{B}^{ij} = -\partial_k(\widetilde{E}^{[ki]j} + \widetilde{E}^{[kj]i}),$$
$$\partial_i\partial_j\widetilde{B}^{ij} = 0.$$
(5.64)

They are solved locally in terms of a field $\hat{\phi}^{[ij]k}$ in the representation **2** of $S_4$:

$$\widetilde{B}^{ij} = -\partial_k(\hat{\phi}^{[ki]j} + \hat{\phi}^{[kj]i}),$$
$$\widetilde{E}^{[ij]k} = \partial_\tau\hat{\phi}^{[ij]k}.$$
(5.65)

The tensor gauge theory Lagrangian can now be written in terms of $\hat{\phi}^{[ij]k}$:

$$\mathcal{L}_E = \frac{g_m^2}{32\pi^2}(\partial_\tau\hat{\phi}^{[ij]k})^2 + \frac{g_e^2}{32\pi^2}\left[\partial_k(\hat{\phi}^{[ki]j} + \hat{\phi}^{[kj]i})\right]^2,$$
(5.66)

subject to the constraint $\hat{\phi}^{[xy]z} + \hat{\phi}^{[yz]x} + \hat{\phi}^{[zx]y} = 0$. Importantly, there is no gauge field in this dual description of the tensor gauge theory.

The nontrivial fluxes of $E_{ij}, B_{[ij]k}$ (see Section 5.3) mean that the periods of $\widetilde{B}^{ij}, \widetilde{E}^{[ij]k}$ are quantized, corresponding to the periodicities of $\hat{\phi}$ in (4.9).

Going back to the Lorentzian signature, we have

$$E^{ij} = \frac{g_e^2}{4\pi}\partial_k(\hat{\phi}^{[ki]j} + \hat{\phi}^{[kj]i}),$$
$$B_{[ij]k} = \frac{g_m^2}{4\pi}\partial_0\hat{\phi}^{[ij]k}$$
(5.67)

The Lorentzian Lagrangian is

$$\mathcal{L} = \frac{g_m^2}{32\pi^2}(\partial_0\hat{\phi}^{[ij]k})^2 - \frac{g_e^2}{32\pi^2}\left[\partial_k(\hat{\phi}^{[ki]j} + \hat{\phi}^{[kj]i})\right]^2.$$
(5.68)

Comparing with (4.5), the duality maps

$$\hat{\mu}_0 = \frac{g_m^2}{8\pi^2} \,, \qquad \hat{\mu} = \frac{g_e^2}{8\pi^2} \,. \tag{5.69}$$

Under the duality between the $A$ and the $\hat{\phi}$ theories, the winding modes of $\hat{\phi}$ are mapped to the electric modes in the $A$ theory. Indeed, their charges, (4.42) and (5.45), and their energies, (4.45) and (5.42), agree. Similarly, the momentum modes of $\hat{\phi}$ are mapped to the magnetic modes in the $A$ theory. Again, their charges, (4.38) and (5.60), and their energies, (4.40) and (5.61), agree. As in every duality transformation, the quantum effects on one side – the energies of the momentum modes of $\hat{\phi}$ and of the electric modes of $A$ – appear classically on the other side.

Finally, we summarize the analogy between the $3+1$-dimensional $A$ tensor gauge theory and $2+1$-dimensional ordinary gauge theory in Table 5.

## 5.9    Robustness and Universality

Since the $U(1)$ $A$-theory is dual to the $\hat{\phi}$-theory of Section 4, the issues of robustness and universality for the $\hat{\phi}$-theory in Section 4.6 directly applies to the $U(1)$ $A$-theory.

Let us comment more on the robustness issue. The microscopic global symmetry $G_{UV}$ is the electric tensor symmetry (5.23). By contrast, the global symmetry $G_{IR}$ of the continuum field theory not only has $G_{UV}$, but also includes the magnetic tensor symmetry (5.29). The latter is absent on the lattice.

In the continuum field theory, there is a $G_{UV}$-invariant, local (monopole) operator violating the magnetic symmetry. In the dual description using the $\hat{\phi}$ field, this local operator can be written as $e^{i\hat{\phi}^{i(jk)}}$. Naively, as in the famous Polyakov mechanism in $2+1$-dimensional $U(1)$ gauge theory [35], perturbations by this local operator would ruin the robustness of this gauge theory. However, this is not the case here, because this operator is irrelevant. One way to see this is that the magnetic modes created by this operator carry energy of order $1/a$ (see Section 4.4 and 5.7). Therefore the operator $e^{i\hat{\phi}^{i(jk)}}$ is irrelevant in the continuum limit ($a \to 0$ with fixed system size $\ell^i$). We conclude that the $U(1)$ $A$-theory is robust under small perturbation by this local operator.[10]

---

[10]Note that this conclusion depends crucially on the continuum limit we consider here. For example, the authors of [17] considered a different limit and argued that the lattice gauge theory of $A$ is not robust against perturbations by the monopole operator.

| | $(2+1)d$<br>$U(1)$ gauge theory | $(3+1)d$<br>$U(1)$ tensor gauge theory $A$ |
|---|---|---|
| gauge symmetry | $A_\mu \to A_\mu + \partial_\mu \alpha$ | $A_0 \to A_0 + \partial_0 \alpha$<br>$A_{ij} \to A_{ij} + \partial_i \partial_j \alpha$ |
| field strength | $E_i = \partial_0 A_i - \partial_i A_0$<br>$B_{xy} = \partial_x A_y - \partial_y A_x$ | $E_{ij} = \partial_0 A_{ij} - \partial_i \partial_j A_0$<br>$B_{[ij]k} = \partial_i A_{jk} - \partial_j A_{ik}$ |
| Lagrangian | $\frac{1}{g^2} E_i E^i - \frac{1}{g^2} B_{xy} B^{xy}$ | $\frac{1}{2g_e^2} E_{ij} E^{ij} - \frac{1}{2g_m^2} B_{[ij]k} B^{[ij]k}$ |
| flux | $\oint d\tau \oint dx^i E_i \in 2\pi\mathbb{Z}$<br>$\oint dx \oint dy B_{xy} \in 2\pi\mathbb{Z}$ | $\oint d\tau \int_{x_1^i}^{x_2^i} dx^i \oint dx^j E_{ij} \in 2\pi\mathbb{Z}$<br>$\int_{x_1^i}^{x_2^i} dx^i \oint dx^j \oint dx^k B_{[jk]i} \in 2\pi\mathbb{Z}$ |
| Gauss law | $\partial^i E_i = 0$ | $\partial_i \partial_j E^{ij} = 0$ |
| eom | $\partial_0 E_i = \partial^j B_{ij}$ | $\frac{1}{g_e^2} \partial_0 E_{ij} = \frac{1}{g_m^2} \partial^k (B_{[ki]j} + B_{[kj]i})$ |
| Bianchi identity | $\partial_0 B_{xy} = \partial_x E_y - \partial_y E_x$ | $\partial_0 B_{[ij]k} = \partial_i E_{jk} - \partial_j E_{ik}$ |
| electric symmetry | electric 1-form<br>$\exp\left[i\frac{2\beta}{g^2} \oint dx^i E_j\right]$ | electric tensor<br>$\exp\left[i\frac{2\beta}{g_e^2} \oint dx^k E_{ij}\right]$ |
| magnetic symmetry | magnetic 0-form<br>$\exp\left[i\frac{\beta}{2\pi} \oint dx \oint dy B_{xy}\right]$ | magnetic tensor<br>$\exp\left[i\frac{\beta}{2\pi} \int_{x_1^i}^{x_2^i} dx^i \oint dx^j \oint dx^k B_{[jk]i}\right]$ |
| electrically charged object | Wilson line<br>$\exp\left[i \oint dx^i A_i\right]$ | Wilson strip<br>$\exp\left[i \int_{x_1^k}^{x_2^k} dx^k \oint_C (dx^i A_{ik} + dx^j A_{jk})\right]$ |
| magnetically charged object | monopole<br>$\exp\left[i\varphi\right]$ | monopole<br>$\exp\left[i\hat{\phi}^{k(ij)}\right]$ |

Table 5: Analogy between the 3+1-dimensional $U(1)$ tensor gauge theory $A$ and the ordinary $2+1$-dimensional $U(1)$ gauge theory.

# 6 The $\hat{A}$ Tensor Gauge Theory

In this section we gauge the $(\mathbf{R}_{\text{time}}, \mathbf{R}_{\text{space}}) = (\mathbf{2}, \mathbf{3}')$ tensor global symmetry (2.20). We will focus on the pure gauge theory without matter. Certain aspects of this tensor gauge theory have been discussed in [8].

The gauge fields are $(\hat{A}_0^{i(jk)}, \hat{A}^{ij})$ in the $(\mathbf{2}, \mathbf{3}')$ of $S_4$. The gauge transformations are

$$\begin{aligned}
\hat{A}_0^{i(jk)} &\to \hat{A}_0^{i(jk)} + \partial_0 \hat{\alpha}^{i(jk)} \,, \\
\hat{A}^{ij} &\to \hat{A}^{ij} + \partial_k \hat{\alpha}^{k(ij)} \,.
\end{aligned} \tag{6.1}$$

where the gauge parameters $\hat{\alpha}^{i(jk)}$ are in the $\mathbf{2}$. The gauge parameters $\hat{\alpha}^{i(jk)}$ are point-wise $2\pi$-periodic, subject to the constraint that $\hat{\alpha}^{x(yz)} + \hat{\alpha}^{y(zx)} + \hat{\alpha}^{z(xy)} = 0$. Globally, this implies that the transition functions can have their own transition functions (see Section 6.3).

The gauge-invariant field strengths are

$$\begin{aligned}
\hat{E}^{ij} &= \partial_0 \hat{A}^{ij} - \partial_k \hat{A}_0^{k(ij)} \,, \\
\hat{B} &= \frac{1}{2} \partial_i \partial_j \hat{A}^{ij} \,,
\end{aligned} \tag{6.2}$$

which are in the $\mathbf{3}'$ and $\mathbf{1}$ of $S_4$, respectively.

## 6.1 Lattice Tensor Gauge Theory

In this subsection we discuss the $U(1)$ lattice tensor gauge theory of $\hat{A}$. We will present both the Lagrangian and Hamiltonian formulations of this lattice model.

For each site $(\hat{\tau}, \hat{x}, \hat{y}, \hat{z})$ on a Euclidean lattice, there are three gauge parameters $\hat{\eta}^{i(jk)}(\hat{\tau}, \hat{x}, \hat{y}, \hat{z}) = e^{i\hat{\alpha}^{i(jk)}(\hat{\tau}, \hat{x}, \hat{y}, \hat{z})}$ (with $i \neq j \neq k$) satisfying $\hat{\eta}^{x(yz)} \hat{\eta}^{y(zx)} \hat{\eta}^{z(xy)} = 1$ at every site. This means that the gauge parameter is in the $\mathbf{2}$ of $S_4$ in the notation of Appendix A.

The gauge fields are placed on the links. Associated with each temporal link, there are three gauge fields $\hat{U}_\tau^{i(jk)}(\hat{\tau}, \hat{x}, \hat{y}, \hat{z})$ satisfying $\hat{U}_\tau^{x(yz)} \hat{U}_\tau^{y(zx)} \hat{U}_\tau^{z(xy)} = 1$, i.e. they are the $\mathbf{2}$ of $S_4$. Associated with each spatial link along the $k$ direction, there is a gauge field $\hat{U}^{ij}$ in the $\mathbf{3}'$.

The gauge transformations act on them as

$$\begin{aligned}
\hat{U}_\tau^{i(jk)}(\hat{\tau}, \hat{x}, \hat{y}, \hat{z}) &\to \hat{U}_\tau^{i(jk)}(\hat{\tau}, \hat{x}, \hat{y}, \hat{z}) \, \hat{\eta}^{i(jk)}(\hat{\tau}, \hat{x}, \hat{y}, \hat{z}) \, \hat{\eta}^{i(jk)}(\hat{\tau}+1, \hat{x}, \hat{y}, \hat{z})^{-1} \,, \\
\hat{U}^{xy}(\hat{\tau}, \hat{x}, \hat{y}, \hat{z}) &\to \hat{U}^{xy}(\hat{\tau}, \hat{x}, \hat{y}, \hat{z}) \, \hat{\eta}^{z(xy)}(\hat{\tau}, \hat{x}, \hat{y}, \hat{z}) \, \hat{\eta}^{z(xy)}(\hat{\tau}, \hat{x}, \hat{y}, \hat{z}+1)^{-1} \,,
\end{aligned} \tag{6.3}$$

and similarly for $\hat{U}^{yz}$ and $\hat{U}^{zx}$.

Let us discuss the gauge invariant local terms in the action. The first kind is a plaquette on the $\tau z$-plane:

$$\hat{L}^{\tau z}(\hat{\tau}, \hat{x}, \hat{y}, \hat{z}) = \hat{U}^{xy}(\hat{\tau}, \hat{x}, \hat{y}, \hat{z}) \, \hat{U}^{z(xy)}_{\tau}(\hat{\tau}, \hat{x}, \hat{y}, \hat{z}+1) \, \hat{U}^{xy}(\hat{\tau}+1, \hat{x}, \hat{y}, \hat{z})^{-1} \, \hat{U}^{z(xy)}_{\tau}(\hat{\tau}, \hat{x}, \hat{y}, \hat{z})^{-1}$$
(6.4)

and similarly for $\hat{L}^{\tau x}$ and $\hat{L}^{\tau y}$. This term becomes the square of the electric field in the continuum limit. The second kind is a product of 12 spatial links around a cube in space at a fixed time:

$$\begin{aligned}
\hat{L}(\hat{\tau}, \hat{x}, \hat{y}, \hat{z}) &= \hat{U}^{yz}(\hat{\tau}, \hat{x}, \hat{y}, \hat{z}) \, \hat{U}^{zx}(\hat{\tau}, \hat{x}+1, \hat{y}, \hat{z})^{-1} \, \hat{U}^{yz}(\hat{\tau}, \hat{x}, \hat{y}+1, \hat{z})^{-1} \, \hat{U}^{zx}(\hat{\tau}, \hat{x}, \hat{y}, \hat{z}) \\
&\times \hat{U}^{yz}(\hat{\tau}, \hat{x}, \hat{y}, \hat{z}+1)^{-1} \, \hat{U}^{zx}(\hat{\tau}, \hat{x}+1, \hat{y}, \hat{z}+1) \, \hat{U}^{yz}(\hat{\tau}, \hat{x}, \hat{y}+1, \hat{z}+1) \, \hat{U}^{zx}(\hat{\tau}, \hat{x}, \hat{y}, \hat{z}+1)^{-1} \\
&\times \hat{U}^{xy}(\hat{\tau}, \hat{x}, \hat{y}, \hat{z}) \, \hat{U}^{xy}(\hat{\tau}, \hat{x}+1, \hat{y}, \hat{z})^{-1} \, \hat{U}^{xy}(\hat{\tau}, \hat{x}+1, \hat{y}+1, \hat{z}) \, \hat{U}^{xy}(\hat{\tau}, \hat{x}, \hat{y}+1, \hat{z})^{-1} \, .
\end{aligned}$$
(6.5)

This term becomes the square of the magnetic field in the continuum limit. The Lagrangian for this lattice model is a sum over the above terms.

In addition to the local, gauge-invariant operators (6.5), there are other non-local, extended ones. For example, we have a line operator along the $x^k$ direction.

$$\prod_{\hat{x}^k=1}^{L^k} \hat{U}^{ij} \, .$$
(6.6)

As in Section 5.1, in the Hamiltonian formulation, we choose the temporal gauge to set all the $\hat{U}^{i(jk)}_{\tau}$'s to 1. We introduce the electric field $\hat{E}^{ij}$ such that $\frac{2}{\hat{g}_e^2}\hat{E}^{ij}$ is conjugate to the phase of the spatial variable $\hat{U}^{ij}$ with $\hat{g}_e$ the electric coupling constant. It differs from the electric field in the continuum by a power of the lattice spacing, which can be added easily on dimensional grounds.

At every site, we impose Gauss law

$$\hat{G}^{[zx]y}(\hat{x}, \hat{y}, \hat{z}) = \hat{E}^{xy}(\hat{x}, \hat{y}, \hat{z}+1) - \hat{E}^{xy}(\hat{x}, \hat{y}, \hat{z}) - \hat{E}^{yz}(\hat{x}+1, \hat{y}, \hat{z}) + \hat{E}^{yz}(\hat{x}, \hat{y}, \hat{z}) = 0 \quad (6.7)$$

and similarly $\hat{G}^{[xy]z} = 0$ and $\hat{G}^{[yz]x} = 0$.

The Hamiltonian is a sum of $(\hat{E}^{ij})^2$ over all the links plus a sum of $\hat{L}$ over all the cubes, with Gauss law imposed by hand. Alternatively, we can impose Gauss law energetically by adding a term $\sum_{\text{sites}}[(\hat{G}^{[xy]z})^2 + (\hat{G}^{[zx]y})^2 + (\hat{G}^{[yz]x})^2]$ to the Hamiltonian.

The lattice model has an electric dipole symmetry whose conserved charges are propor-

tional to

$$\sum_{\hat{x}=1}^{L^x} \hat{E}^{zx}(\hat{x}, \hat{y}_0, \hat{z}_0), \quad \sum_{\hat{y}=1}^{L^y} \hat{E}^{zy}(\hat{x}_0, \hat{y}, \hat{z}_0).$$  (6.8)

There are 4 other charges associated with the other directions. They commute with the Hamiltonian. These two electric dipole symmetries rotate the phases of $\hat{U}^{ij}$ along a strip on the $zx$ and $yz$ planes, respectively.

## 6.2 Continuum Lagrangian

The $3 + 1$-dimensional Lagrangian for the pure tensor gauge theory of $\hat{A}$ is

$$\mathcal{L} = \frac{1}{2\hat{g}_e^2} \hat{E}_{ij} \hat{E}^{ij} - \frac{1}{\hat{g}_m^2} \hat{B}^2.$$  (6.9)

Note that $\hat{g}_e$ has mass dimension 0 and $\hat{g}_m$ has mass dimension 1. The equations of motion are

$$\frac{1}{\hat{g}_e^2} \partial_0 \hat{E}^{ij} = -\frac{1}{\hat{g}_m^2} \partial^i \partial^j \hat{B},$$

$$\partial^k \hat{E}^{ij} - \partial^i \hat{E}^{kj} = 0.$$  (6.10)

Equivalently, the second equation, which is Gauss law, can be written as $2\partial^k \hat{E}^{ij} - \partial^i \hat{E}^{kj} - \partial^j \hat{E}^{ki} = 0$.

There is also a Bianchi identity

$$\partial_0 \hat{B} = \frac{1}{2} \partial_i \partial_j \hat{E}^{ij}.$$  (6.11)

## 6.3 Fluxes

Let us put the theory on a Euclidean 4-torus with lengths $\ell^\tau, \ell^x, \ell^y, \ell^z$. Consider a bundle with transition functions $\hat{g}_{(\tau)}$ at $\tau = \ell^\tau$:

$$\hat{g}_{(\tau)}^{z(xy)} = -\hat{g}_{(\tau)}^{x(yz)} = \frac{2\pi z}{\ell^z}, \quad \hat{g}_{(\tau)}^{y(zx)} = 0.$$  (6.12)

This means that

$$\hat{A}^{xy}(\tau = \ell^\tau, x, y, z) = \hat{A}^{xy}(\tau = 0, x, y, z) + \partial_z \hat{g}_{(\tau)}^{z(xy)}.$$  (6.13)

Such gauge field configurations realize a nontrivial electric flux:

$$\hat{e}^{xy}(x,y) \equiv \oint d\tau \oint dz \, \hat{E}^{xy} \in 2\pi\mathbb{Z} \, . \tag{6.14}$$

The Bianchi identity (6.11) implies that

$$\partial_x \partial_y \hat{e}^{xy}(x,y) = 0 \tag{6.15}$$

and therefore the electric flux can be written as

$$\hat{e}^{xy}(x,y) = \hat{e}^{xy}_x(x) + \hat{e}^{xy}_y(y) \, . \tag{6.16}$$

Electric fluxes along the other directions are realized in similar bundles.

To realize the magnetic flux, we consider a bundle whose transition function at $x = \ell^x$ is

$$\begin{aligned}
\hat{g}^{x(yz)}_{(x)} &= -2\pi \left[ \frac{z}{\ell^z}\Theta(y - y_0) + \frac{y}{\ell^y}\Theta(z - z_0) - \frac{yz}{\ell^y \ell^z} \right] \, , \\
\hat{g}^{y(zx)}_{(x)} &= 0 \, , \\
\hat{g}^{z(xy)}_{(x)} &= -\hat{g}^{x(yz)}_{(x)} \, ,
\end{aligned} \tag{6.17}$$

the transition function at $y = \ell^y$ is

$$\begin{aligned}
\hat{g}^{x(yz)}_{(y)} &= 0 \, , \\
\hat{g}^{y(zx)}_{(y)} &= -2\pi \left[ \frac{z}{\ell^z}\Theta(x - x_0) + \frac{x}{\ell^x}\Theta(z - z_0) - \frac{xz}{\ell^x \ell^z} \right] \, , \\
\hat{g}^{z(xy)}_{(y)} &= -\hat{g}^{y(zx)}_{(y)} \, ,
\end{aligned} \tag{6.18}$$

and there is no nontrivial transition function at $z = \ell^z$, i.e. $\hat{g}^{k(ij)}_{(z)} = 0$. This means that as we go around the $x$ direction, the gauge field changes by

$$\hat{A}^{ij}(\tau, x = \ell^x, y, z) = \hat{A}^{ij}(\tau, x = 0, y, z) + \partial_k \hat{g}^{k(ij)}_{(x)} \, , \tag{6.19}$$

and similarly for the $y, z$ directions.

We should make some comments about the transition functions (6.17) and (6.18).

First, these transition functions have their own transition functions. For example, the transition functions $\hat{g}^{k(ij)}_{(x)}$ on the $yz$-plane have their own transition functions at $y = \ell^y$:

$$\hat{g}^{x(yz)}_{(x)} \to \hat{g}^{x(yz)}_{(x)} - 2\pi\Theta(z - z_0) \, , \ \ \hat{g}^{y(zx)}_{(x)} \to \hat{g}^{y(zx)}_{(x)} \, , \ \ \hat{g}^{z(xy)}_{(x)} \to \hat{g}^{z(xy)}_{(x)} + 2\pi\Theta(z - z_0) \, . \tag{6.20}$$

Such a need for transition functions for transition functions is standard in higher form gauge theories.

Second, we argued in Section 4.5 that the configuration (4.47) should not be included in the $\hat{\phi}$ continuum field theory. Its energy is of order $1/a$ and it is not protected by any global symmetry. In fact, it violates the global winding tensor symmetry of the light modes. However, here the transition functions (6.17) and (6.18) are similar to (4.47). Why should we include them? The point is that unlike the $\hat{\phi}$-theory, here these transition functions do not violate any global symmetry. Furthermore, as we will see below, the energy of the configurations with these transition functions are of the same order, $1/a$, and they are the lightest states carrying the global symmetry charge. We conclude that when studying singular configurations in the continuum $\hat{A}$ gauge theory, we must consider gauge transformations and transition functions that are not important in the continuum $\hat{\phi}$-theory.

Third, as always, the transition functions can change by performing non-periodic gauge transformations. For example, the transformation

$$
\hat{\alpha}^{x(yz)} = -\hat{\alpha}^{z(xy)} = 2\pi \left[ \frac{yz}{\ell^y \ell^z} \Theta(x - x_0) + \frac{zx}{\ell^z \ell^x} \Theta(y - y_0) + \frac{xy}{\ell^x \ell^y} \Theta(z - z_0) - 2 \frac{xyz}{\ell^x \ell^y \ell^z} \right] ,
$$
$$
\hat{\alpha}^{y(zx)} = 0
$$

(6.21)

exchanges $x$ with $z$ in (6.17) and (6.18). While changing the transition functions, this does not change the bundle.

Using the transition functions (6.17) and (6.18)

$$
\oint dy \oint dz \hat{B} = \oint dz \partial_x \partial_z \hat{g}_{(y)}^{z(xy)} = 2\pi \delta(x - x_0) ,
$$
$$
\oint dx \oint dy \hat{B} = \oint dy \partial_y \partial_z \hat{g}_{(x)}^{z(xy)} = 2\pi \delta(z - z_0) ,
$$
$$
\oint dz \oint dx \hat{B} = \oint dz \partial_y \partial_z \hat{g}_{(x)}^{z(xy)} = 2\pi \delta(y - y_0) .
$$

(6.22)

By taking linear combinations of such bundles, we realize the general magnetic flux

$$
\hat{b}^x \equiv \int_{x_1}^{x_2} dx \oint dy \oint dz \hat{B} \in 2\pi \mathbb{Z} ,
$$

(6.23)

and similarly for the other directions. The Bianchi identity (6.11) implies that

$$
\partial_\tau \hat{b} = 0 .
$$

(6.24)

Hence the magnetic flux is constant in time.

These fluxes correspond to operators that multiply to 1 on the lattice. The electric flux (6.14) corresponds to the product $\prod_{\hat{\tau},\hat{z}} \hat{L}^{\tau z} = 1$ on the lattice. Similarly, the magnetic flux (6.23) corresponds to the product $\prod_{\hat{y},\hat{z}} \hat{L} = 1$ on the lattice.

## 6.4 Global Symmetries and Their Charges

We now discuss the global symmetries of the tensor gauge theory of $\hat{A}$.

### 6.4.1 Electric Dipole Symmetry

The equation of motion (6.10) is recognized as the current conservation equation

$$\partial_0 J_0^{ij} = \partial^i \partial^j J \tag{6.25}$$

with currents

$$\begin{aligned} J_0^{ij} &= -\frac{2}{\hat{g}_e^2} \hat{E}^{ij}\,, \\ J &= \frac{2}{\hat{g}_m^2} \hat{B}\,. \end{aligned} \tag{6.26}$$

The second equation of (6.10) is an additional differential equation

$$\partial^k J_0^{ij} - \partial^i J_0^{kj} = 0\,, \tag{6.27}$$

imposed on $J_0^{ij}$. We will refer to (6.26) as the *electric dipole symmetry.* This is the continuum version of the lattice symmetry (6.8).

The charges are

$$Q(\mathcal{C}^{xy}, z) = -\frac{2}{\hat{g}_e^2} \oint_{\mathcal{C}^{xy} \in (x,y)} \left( dx \hat{E}^{zx} + dy \hat{E}^{zy} \right) \tag{6.28}$$

where $\mathcal{C}^{xy}$ is a closed curve on the $xy$-plane. The differential condition (6.27) implies that the charge is independent of small deformations of the curve $\mathcal{C}^{xy}$. The symmetry operator is a strip operator:

$$\mathcal{U}(\beta; z_1, z_2, \mathcal{C}^{xy}) = \exp\left[ -i \frac{2\beta}{\hat{g}_e^2} \int_{z_1}^{z_2} dz \oint_{\mathcal{C}^{xy}} \left( dx\, \hat{E}^{zx} + dy\, \hat{E}^{yz} \right) \right]\,. \tag{6.29}$$

Here the strip is the direct product of the segment $[z_1, z_2]$ and the curve $\mathcal{C}^{xy}$ on the $xy$-plane. Similarly, we have operators along the other directions. The electric dipole symmetry acts

on the gauge fields as

$$
\begin{aligned}
\hat{A}^{xy} &\to \hat{A}^{xy} + \hat{c}^{xy}_x(x) + \hat{c}^{xy}_y(y)\,, \\
\hat{A}^{zx} &\to \hat{A}^{zx} + \hat{c}^{zx}_z(z) + \hat{c}^{zx}_x(x)\,, \\
\hat{A}^{yz} &\to \hat{A}^{yz} + \hat{c}^{yz}_y(y) + \hat{c}^{yz}_z(z)\,,
\end{aligned}
\tag{6.30}
$$

parametrized by six functions $\hat{c}^{ij}_i(x^i)$ of one variable.

The electrically charged operator is a line operator

$$
\hat{W}^k(x^i, x^j) = \exp\left[i \oint dx^k\, \hat{A}^{ij}\right]\,.
\tag{6.31}
$$

This is the continuum version of the gauge-invariant operator (6.6) on the lattice. $\mathcal{U}$ and $\hat{W}^k$ obeys the following equal-time commutation relation

$$
\mathcal{U}(\beta; z_1, z_2, \mathcal{C}^{xy})\,\hat{W}^x(y_0, z_0) = e^{-i\beta\, I(\mathcal{C}^{xy}, y_0)}\,\hat{W}^x(y_0, z_0)\,\mathcal{U}(\beta; z_1, z_2, \mathcal{C}^{xy})\,, \quad \text{if } z_1 < z_0 < z_2\,,
\tag{6.32}
$$

and they commute otherwise. Here $I(\mathcal{C}^{xy}, y_0)$ is the intersection number between the curve $\mathcal{C}^{xy}$ and the $y = y_0$ line on the $xy$-plane.

Only integer powers of $\hat{W}^k$ are invariant under the large gauge transformation $\hat{\alpha}^{k(ij)} = -\hat{\alpha}^{i(jk)} = \frac{2\pi x^k}{\ell^k}$, $\hat{\alpha}^{j(ki)} = 0$. It then follows that the exponent $\beta$ is $2\pi$-periodic. Therefore, the global structure of the electric multipole global symmetry is $U(1)$ not $\mathbb{R}$.

We also have gauge invariant strip operators:

$$
\hat{P}(z_1, z_2, \mathcal{C}) = \exp\left[i \int_{z_1}^{z_2} dz \oint_{\mathcal{C}} \left(\partial_z \hat{A}^{yz} dx - \partial_z \hat{A}^{zx} dy - \partial_y \hat{A}^{xy} dy\right)\right]\,,
\tag{6.33}
$$

where $\mathcal{C}$ is a closed curve on the $xy$-plane.

### 6.4.2 Magnetic Dipole Symmetry

The Bianchi identity (6.11) is recognized as the current conservation equation

$$
\partial_0 J_0 = \frac{1}{2}\partial_i \partial_j J^{ij}
\tag{6.34}
$$

with currents

$$
\begin{aligned}
J_0 &= \frac{1}{2\pi}\hat{B}\,, \\
J^{ij} &= \frac{1}{2\pi}\hat{E}^{ij}\,.
\end{aligned}
\tag{6.35}
$$

We will refer to (6.35) as the *magnetic dipole symmetry*. This symmetry is absent on the lattice.

The conserved charge operator of the magnetic dipole global symmetry is

$$Q_{ij}(x^k) = \frac{1}{2\pi} \oint dx^i \oint dx^j \, \hat{B} \, . \tag{6.36}$$

The symmetry operator is a slab with finite width in the $k$ direction

$$\mathcal{U}_{ij}(\beta; x_1^k, x_2^k) = \exp\left[i\beta \int_{x_1^k}^{x_2^k} dx^k \, Q_{ij}(x^k)\right] = \exp\left[i\frac{\beta}{2\pi} \int_{x_1^k}^{x_2^k} dx^k \oint dx^i \oint dx^j \, \hat{B}\right] \, . \tag{6.37}$$

The magnetically charged objects under the magnetic dipole global symmetry are point-operators. They are monopole operators. The monopole operator $e^{i\phi}$ can be written in terms of the dual field $\phi$. See Section 6.8.

## 6.5 Defects as Lineons

There are three species of particles, each associated with a spatial direction. A charge $+1$, static particle associated with the $x^i$ direction is described by the following defect[11]

$$\exp\left[i \int_{-\infty}^{\infty} dt \hat{A}_0^{i(jk)}\right] \, . \tag{6.38}$$

A particle of species $x^i$ can move in the $x^i$-direction by itself. This motion is captured by the following line defect in spacetime

$$\hat{W}^i(x^j, x^k, \mathcal{C}) = \exp\left[i \int_{\mathcal{C}} \left(\hat{A}_0^{i(jk)} dt + \hat{A}^{jk} dx^i\right)\right] \, , \tag{6.39}$$

where $\mathcal{C}$ is a spacetime curve on the $(t, x^i)$-plane representing the motion of a particle along the $x^i$-direction. The particle by itself cannot turn in space; it is confined to move along the $x^i$-direction. This particle is the probe limit of the lineon.

A pair of lineons of species, say, $x$ with gauge charges $\pm 1$ separated in the $z$ direction can move collectively not only in the $x$ direction, but also the $y$ direction. This motion is

---

[11]We can study the Euclidean version of this defect and let it wind around the Euclidean time direction. Then, invariance under the large gauge transformation $\hat{\alpha}^{i(jk)} = -\hat{\alpha}^{j(ki)} = 2\pi\frac{\tau}{\ell^\tau}, \hat{\alpha}^{k(ij)} = 0$ quantizes the charge.

captured by the defect

$$\hat{P}(z_1, z_2, \mathcal{C}) = \exp\left[i \int_{z_1}^{z_2} dz \int_{\mathcal{C}} \left(\partial_z \hat{A}_0^{x(yz)} dt + \partial_z \hat{A}^{yz} dx - \partial_z \hat{A}^{zx} dy - \partial_y \hat{A}^{xy} dy\right)\right] \quad (6.40)$$

where $\mathcal{C}$ is a spacetime curve in $(t, x, y)$. We will refer to this dipole of lineons as a planon on the $(x, y)$-plane.

In the special case when $\mathcal{C}$ is at a fixed time, then the defects (6.39) and (6.40) reduce to the operators (6.31) and (6.33), respectively.

## 6.6  Electric Modes

In this subsection we study states that are charged under the electric dipole symmetry (6.26).

Consider plane wave modes in $\mathbb{R}^{3,1}$ in the temporal gauge $\hat{A}_0^{i(jk)} = 0$:

$$\hat{A}^{ij} = \hat{C}^{ij} e^{i\omega t + ik_i x^i}, \quad (6.41)$$

with $\hat{C}^{ij}$ in the **3'**. The dispersion relation is

$$\omega^4 \left[\frac{\hat{g}_m^2}{\hat{g}_e^2} \omega^2 - k_x^2 k_y^2 - k_x^2 k_z^2 - k_y^2 k_z^2\right] = 0. \quad (6.42)$$

There are three solutions for $\omega^2$.

Consider first the case of generic momenta. Two of the solutions have zero energy $\omega^2 = 0$. They are the two residual pure gauge modes. The remaining one is

$$\omega^2 = \frac{\hat{g}_e^2}{\hat{g}_m^2} \left(k_x^2 k_y^2 + k_x^2 k_z^2 + k_y^2 k_z^2\right). \quad (6.43)$$

It leads to a Fock space of "photons."

When two of the momenta, say $k_x$ and $k_y$, vanish, the energy is zero for all $k_z$. Let us study it in more detail. In this case, the equations of motion become degenerate, and we have three solutions for $\omega^2$ all having $\omega^2 = 0$.

The analysis of the gauge modes is different than for generic momenta. In order to preserve $k_x = k_y = 0$, the gauge transformation parameter must be independent of $x, y, t$ and therefore it leads to a single pure gauge mode

$$\hat{A}^{xy} = \partial_z \hat{\alpha}^{z(xy)}, \quad \hat{A}^{yz} = \hat{A}^{xz} = 0. \quad (6.44)$$

In position space, the remaining two zero-energy solutions are

$$\hat{A}^{xy} = 0, \quad \hat{A}^{yz} = \hat{F}_z^{yz}(z), \quad \hat{A}^{xz} = \hat{F}_z^{xz}(z), \tag{6.45}$$

for any functions $\hat{F}_z^{yz}(z), \hat{F}_z^{xz}(z)$. Combining all three directions, we have 6 zero-energy solutions, each a function of one variable.

These modes are acted by the electric dipole symmetry (6.30). Therefore we will refer to them as the electric modes.

Let us quantize these modes on a 3-torus with lengths $\ell^x, \ell^y, \ell^z$:

$$\hat{A}^{ij} = \frac{1}{\ell^k} \hat{f}_i^{ij}(t, x^i) + \frac{1}{\ell^k} \hat{f}_j^{ij}(t, x^j). \tag{6.46}$$

We will focus on $\hat{A}^{xy}$; the analysis of the other two components is similar. The Lagrangian for these momentum modes is

$$L = \frac{1}{\hat{g}_e^2} \frac{1}{\ell^z} \left[ \ell^y \oint dx (\dot{\hat{f}}_x^{xy})^2 + \ell^x \oint dy (\dot{\hat{f}}_y^{xy})^2 + 2 \oint dx \dot{\hat{f}}_x^{xy} \oint dy \dot{\hat{f}}_y^{xy} \right]. \tag{6.47}$$

The quantization of these modes is identical to that of the momentum modes of the $2 + 1$-dimensional $\phi$-theory (1.1). See Section 4.1 of [3] for details.

The conjugate momenta are

$$\begin{aligned}
\pi_x^{xy}(t, x) &= \frac{2}{\hat{g}_e^2 \ell^z} \left( \ell^y \dot{\hat{f}}_x^{xy}(t, x) + \oint dy \dot{\hat{f}}_y^{xy}(t, y) \right), \\
\pi_y^{xy}(t, y) &= \frac{2}{\hat{g}_e^2 \ell^z} \left( \ell^x \dot{\hat{f}}_y^{xy}(t, y) + \oint dx \dot{\hat{f}}_x^{xy}(t, x) \right).
\end{aligned} \tag{6.48}$$

They are subject to the constraint:

$$\oint dx \pi_x^{xy}(x) = \oint dy \pi_y^{xy}(y). \tag{6.49}$$

The point-wise periodicity of $\hat{f}_i^{xy}$ implies that their conjugate momenta $\pi_i^{xy}$ are linear combinations of delta functions with integer coefficients:

$$Q(\mathcal{C}_y^{xy}, x) = -\pi_x^{xy}(x) = \sum_\alpha N_\alpha^x \delta(x - x_\alpha), \quad Q(\mathcal{C}_x^{xy}, y) = -\pi_y^{xy}(y) = \sum_\beta N_\beta^y \delta(y - y_\beta),$$

$$N^{xy} \equiv \sum_\alpha N_\alpha^x = \sum_\beta N_\beta^y, \quad N_\alpha^x, N_\beta^y \in \mathbb{Z}. \tag{6.50}$$

Here $\{x_\alpha\}$ and $\{y_\beta\}$ are a finite set of points on the $x$ and $y$ axes, respectively. $\mathcal{C}_i^{xy}$ is a closed curve on the $xy$-plane that wraps around the $x^i$ direction once and does not wrap around the other direction. Note that the momenta are the charges $Q(\mathcal{C}_y^{xy}, x), Q(\mathcal{C}_x^{xy}, y)$ of the electric dipole symmetry.

The minimal energy with these charges is

$$H = \frac{\hat{g}_e^2 \ell^z}{4 \ell^x \ell^y} \left[ \ell^x \sum_\alpha (N_{y\alpha}^x)^2 \delta(0) + \ell^y \sum_\beta (N_{x\beta}^y)^2 \delta(0) - (N^{xy})^2 \right] , \tag{6.51}$$

which is order $\frac{1}{a}$. The charges and energies of the modes $\pi_i^{ij}$ associated with the other directions can be computed similarly.

## 6.7  Magnetic Modes

In this subsection we discuss states that are charged under the magnetic dipole symmetry (6.35).

*Minimally Charged States*

The bundle realizing the minimal magnetic dipole symmetry charge is characterized by the transition functions in (6.17) and (6.18). The minimum energy configuration in this bundle is:

$$\hat{A}^{xy} = 2\pi \left[ \frac{y}{\ell^y \ell^z} \Theta(x - x_0) + \frac{x}{\ell^x \ell^z} \Theta(y - y_0) + \frac{xy}{\ell^x \ell^y} \delta(z - z_0) - 2\frac{xy}{\ell^x \ell^y \ell^z} \right] ,$$
$$\hat{A}^{yz} = \hat{A}^{zx} = 0 . \tag{6.52}$$

Its magnetic field is

$$\hat{B} = \frac{2\pi}{\ell^x \ell^y \ell^z} \left[ \ell^x \delta(x - x_0) + \ell^y \delta(y - y_0) + \ell^z \delta(z - z_0) - 2 \right] . \tag{6.53}$$

As a check, note that it is consistent with (6.22).

The energy of this minimally charged state is

$$H = \frac{1}{\hat{g}_m^2} \oint dx \oint dy \oint dz \hat{B}^2 = \frac{4\pi^2}{\hat{g}_m^2 \ell^x \ell^y \ell^z} \left[ (\ell^x + \ell^y + \ell^z) \delta(0) - 2 \right] , \tag{6.54}$$

which is of order $\frac{1}{a}$.

*General Charged States*

A more general gauge field configuration carrying the magnetic dipole charges is

$$\hat{A}^{xy} = 2\pi \left[ \frac{y}{\ell^y \ell^z} \sum_\alpha W_{x\,\alpha} \Theta(x - x_\alpha) + \frac{x}{\ell^x \ell^z} \sum_\beta W_{y\,\beta} \Theta(y - y_\beta) + \frac{xy}{\ell^x \ell^y} \sum_\gamma W_{z\,\gamma} \delta(z - z_\gamma) - \frac{2W xy}{\ell^x \ell^y \ell^z} \right] ,$$

$$\hat{A}^{yz} = \hat{A}^{zx} = 0 \, ,$$

$$W_{x,\alpha}, W_{y\,\beta}, W_{z\,\gamma} \in \mathbb{Z} \, , \qquad W \equiv \sum_\alpha W_{x\,\alpha} = \sum_\beta W_{y\,\beta} = \sum_\gamma W_{z\,\gamma} \, .$$

$$(6.55)$$

Its bundle is characterized by the following transition functions. The transition functions at $x = \ell^x$ are

$$\hat{g}^{x(yz)}_{(x)} = -2\pi \left[ \frac{z}{\ell^z} \sum_\beta W_{y\,\beta} \Theta(y - y_\beta) + \frac{y}{\ell^y} \sum_\gamma W_{z\,\gamma} \Theta(z - z_\gamma) - W \frac{yz}{\ell^y \ell^z} \right] \, ,$$

$$\hat{g}^{y(zx)}_{(x)} = 0 \, ,$$

$$\hat{g}^{z(xy)}_{(x)} = -\hat{g}^{x(yz)}_{(x)}$$

$$(6.56)$$

The transition functions at $y = \ell^y$ are

$$\hat{g}^{x(yz)}_{(y)} = 0 \, ,$$

$$\hat{g}^{y(zx)}_{(y)} = -2\pi \left[ \frac{z}{\ell^z} \sum_\alpha W_{x\,\alpha} \Theta(x - x_\alpha) + \frac{x}{\ell^x} \sum_\gamma W_{z\,\gamma} \Theta(z - z_\gamma) - W \frac{xz}{\ell^x \ell^z} \right] \, ,$$

$$\hat{g}^{z(xy)}_{(y)} = -\hat{g}^{y(zx)}_{(y)} \, .$$

$$(6.57)$$

And the transition functions at $z = \ell^z$ are trivial, i.e. $\hat{g}^{k(ij)}_{(z)} = 0$. The bundle is labeled by the integers $W_{x,\alpha}, W_{y\,\beta}, W_{z\,\gamma}$.

As in the $A$ theory, the underlying lattice theory here also does not have the magnetic symmetry and such bundles. Nonetheless, we can consider the points $x^i_\alpha$ to be chosen from a lattice with $L^i$ sites in the $x^i$ direction. Then, we have $L^x + L^y + L^z - 2$ distinct bundles where the $-2$ comes from the constraints in (6.55).

The magnetic field is

$$\hat{B} = \frac{2\pi}{\ell^x \ell^y \ell^z} \left[ \ell^x \sum_\alpha W_{x\,\alpha} \delta(x - x_\alpha) + \ell^y \sum_\beta W_{y\,\beta} \delta(y - y_\beta) + \ell^z \sum_\gamma W_{z\,\gamma} \delta(z - z_\gamma) - 2W \right]$$

$$(6.58)$$

This realizes the general magnetic dipole symmetry charges

$$Q_{yz}(x) = \frac{1}{2\pi} \oint dy \oint dz \hat{B} = \sum_\alpha W_{x\,\alpha} \delta(x - x_\alpha)\,,$$

$$Q_{zx}(y) = \frac{1}{2\pi} \oint dz \oint dx \hat{B} = \sum_\beta W_{y\,\beta} \delta(y - y_\beta)\,, \qquad (6.59)$$

$$Q_{xy}(z) = \frac{1}{2\pi} \oint dx \oint dy \hat{B} = \sum_\gamma W_{z\,\gamma} \delta(z - z_\gamma)\,.$$

(6.55) is the the minimum energy configuration with these charges. Its energy is

$$H = \frac{4\pi^2}{\hat{g}_m^2 \ell^x \ell^y \ell^z} \left[ \ell^x \delta(0) \sum_\alpha W_{x\,\alpha}^2 + \ell^y \delta(0) \sum_\beta W_{y\,\beta}^2 + \ell^z \delta(0) \sum_\gamma W_{z\,\gamma}^2 - 2W^2 \right]\,, \qquad (6.60)$$

which is of order $\frac{1}{a}$.

## 6.8   Duality Transformation

In this subsection we will perform a duality transformation on the $U(1)$ tensor gauge theory of $\hat{A}$ and show that it is dual to the non-gauge theory of $\phi$ in Section 3.

Let us rewrite the Euclidean Lagrangian as

$$\begin{aligned}
\mathcal{L}_E = {} & \frac{1}{2\hat{g}_e^2} \hat{E}_{ij} \hat{E}^{ij} + \frac{1}{\hat{g}_m^2} \hat{B}^2 \\
& + \frac{i}{2(2\pi)} \check{B}_{ij} \left( \partial_\tau \hat{A}^{ij} - \partial_k \hat{A}_\tau^{k(ij)} - \hat{E}^{ij} \right) + \frac{i}{2\pi} \check{E} \left( \frac{1}{2} \partial_i \partial_j \hat{A}^{ij} - \hat{B} \right)
\end{aligned} \qquad (6.61)$$

where now $\hat{E}^{ij}, \hat{B}, \check{B}_{ij}, \check{E}$ are independent fields.

If we integrate out the Lagrange multipliers $\check{B}_{ij}, \check{E}$, we recover the original Lagrangian (6.9). Instead, we integrate out $\hat{E}^{ij}, \hat{B}$ to obtain $\hat{E}^{ij} = i\frac{\hat{g}_e^2}{4\pi} \check{B}^{ij}$ and $\hat{B} = i\frac{\hat{g}_m^2}{4\pi} \check{E}$. The Lagrangian becomes

$$\begin{aligned}
\mathcal{L}_E = {} & \frac{\hat{g}_e^2}{32\pi^2} \check{B}^{ij} \check{B}_{ij} + \frac{\hat{g}_m^2}{16\pi^2} \check{E}^2 \\
& + \frac{i}{2(2\pi)} \check{B}_{ij} \left( \partial_\tau \hat{A}^{ij} - \partial_k \hat{A}_\tau^{k(ij)} \right) + \frac{i}{2(2\pi)} \check{E} \partial_i \partial_j \hat{A}^{ij}\,.
\end{aligned} \qquad (6.62)$$

Next, we integrate out $\hat{A}_\tau^{i(jk)}, \hat{A}^{ij}$ to find the constraints

$$\partial_\tau \check{B}_{ij} = \partial_i \partial_j \check{E} \,,$$
$$\partial_k \check{B}_{ij} - \partial_i \check{B}_{kj} = 0 \,. \tag{6.63}$$

These constraints are locally solved by a real scalar $\phi$

$$\check{B}_{ij} = \partial_i \partial_j \phi \,,$$
$$\check{E} = \partial_\tau \phi \,. \tag{6.64}$$

The Euclidean Lagrangian written in terms of $\phi$ is then

$$\mathcal{L}_E = \frac{\hat{g}_m^2}{16\pi^2} (\partial_\tau \phi)^2 + \frac{\hat{g}_e^2}{32\pi^2} (\partial_i \partial_j \phi)^2 \,. \tag{6.65}$$

The nontrivial fluxes of $\hat{E}^{ij}, \hat{B}$ (see Section 6.3) mean that the periods of $\check{B}_{ij}, \check{E}$ are quantized, corresponding to the periodicities of $\phi$ in (3.5).

When we Wick rotate to the Lorentzian signature, we have

$$\hat{E}^{ij} = -\frac{\hat{g}_e^2}{4\pi} \partial^i \partial^j \phi \,,$$
$$\hat{B} = \frac{\hat{g}_m^2}{4\pi} \partial_0 \phi \,, \tag{6.66}$$

and the Lagrangian is

$$\mathcal{L} = \frac{\hat{g}_m^2}{16\pi^2} (\partial_0 \phi)^2 - \frac{\hat{g}_e^2}{32\pi^2} (\partial_i \partial_j \phi)^2 \,. \tag{6.67}$$

Comparing with (3.3), the duality map is

$$\mu_0 = \frac{\hat{g}_m^2}{8\pi^2} \,, \qquad \frac{1}{\mu} = \frac{\hat{g}_e^2}{8\pi^2} \,. \tag{6.68}$$

Under the duality, the momentum modes of $\phi$ are mapped to the magnetic modes of $\hat{A}$. Indeed, their charges (3.27) and (6.59) and their energies (3.28) and (6.60) match. The winding modes of $\phi$ are mapped to the electric modes of $\hat{A}$. Again, their charges (3.30) and (6.50) and their energies (3.32) and (6.51) match.

Finally, we summarize the analogy between the $3+1$-dimensional $\hat{A}$ tensor gauge theory and $2+1$-dimensional ordinary gauge theory in Table 6.

| | $(2+1)d$ $U(1)$ gauge theory | $(3+1)d$ $U(1)$ tensor gauge theory $\hat{A}$ |
|---|---|---|
| gauge symmetry | $A_\mu \to A_\mu + \partial_\mu \alpha$ | $\hat{A}_0^{k(ij)} \to \hat{A}_0^{k(ij)} + \partial_0 \hat{\alpha}^{k(ij)}$ <br> $\hat{A}^{ij} \to \hat{A}^{ij} + \partial_k \hat{\alpha}^{k(ij)}$ |
| field strength | $E_i = \partial_0 A_i - \partial_i A_0$ <br> $B_{xy} = \partial_x A_y - \partial_y A_x$ | $\hat{E}_{ij} = \partial_0 \hat{A}^{ij} - \partial_k \hat{A}_0^{k(ij)}$ <br> $\hat{B} = \frac{1}{2}\partial_i \partial_j \hat{A}^{ij}$ |
| Lagrangian | $\frac{1}{g^2} E_i E^i - \frac{1}{g^2} B_{xy} B^{xy}$ | $\frac{1}{2\hat{g}_e^2} \hat{E}_{ij}\hat{E}^{ij} - \frac{1}{\hat{g}_m^2}\hat{B}^2$ |
| flux | $\oint d\tau \oint dx^i E_i \in 2\pi\mathbb{Z}$ <br> $\oint dx \oint dy B_{xy} \in 2\pi\mathbb{Z}$ | $\oint d\tau \oint dx^k \hat{E}^{ij} \in 2\pi\mathbb{Z}$ <br> $\int_{x_1^i}^{x_2^i} dx^i \oint dx^j \oint dx^k \hat{B} \in 2\pi\mathbb{Z}$ |
| Gauss law | $\partial^i E_i = 0$ | $\partial^k \hat{E}^{ij} - \partial^i \hat{E}^{kj} = 0$ |
| eom | $\partial_0 E_i = \partial^j B_{ij}$ | $\frac{1}{\hat{g}_e^2}\partial_0 \hat{E}^{ij} = -\frac{1}{\hat{g}_m^2}\partial^i \partial^j \hat{B}\,,$ |
| Bianchi identity | $\partial_0 B_{xy} = \partial_x E_y - \partial_y E_x$ | $\partial_0 \hat{B} = \frac{1}{2}\partial_i \partial_j \hat{E}^{ij}$ |
| electric symmetry | electric 1-form <br> $\exp\left[i\frac{2\beta}{g^2}\oint dx^i E_j\right]$ | electric dipole <br> $\exp\left[-i\frac{2\beta}{\hat{g}_e^2}\int_{x_1^k}^{x_2^k} dx^k \oint_{\mathcal{C}}\left(dx^i \hat{E}^{ki} + dx^j \hat{E}^{kj}\right)\right]$ |
| magnetic symmetry | magnetic 0-form <br> $\exp\left[i\frac{\beta}{2\pi}\oint dx \oint dy B_{xy}\right]$ | magnetic dipole <br> $\exp\left[i\frac{\beta}{2\pi}\int_{x_1^i}^{x_2^i} dx^i \oint dx^j \oint dx^k \hat{B}\right]$ |
| electrically charged object | Wilson line <br> $\exp\left[i\oint dx^i A_i\right]$ | Wilson line <br> $\exp\left[i\oint dx^k \hat{A}^{ij}\right]$ |
| magnetically charged object | monopole <br> $\exp\left[i\varphi\right]$ | monopole <br> $\exp\left[i\phi\right]$ |

Table 6: Analogy between the 3+1-dimensional $U(1)$ tensor gauge theory $\hat{A}$ and the ordinary $2+1$-dimensional $U(1)$ gauge theory.

## 6.9 Robustness and Universality

As we saw in Section 6.8, the $U(1)$ $\hat{A}$ gauge theory is dual to the $\phi$-theory of Section 3. Therefore the same conclusions on robustness and universality in Section 3.6 hold true for the $\hat{A}$ theory as well.

# Acknowledgements

We thank X. Chen, M. Cheng, M. Fisher, A. Gromov, M. Hermele, P.-S. Hsin, A. Kitaev, D. Radicevic, L. Radzihovsky, S. Sachdev, D. Simmons-Duffin, S. Shenker, K. Slagle, D. Stanford for helpful discussions. We also thank P. Gorantla, H.T. Lam, D. Radicevic and T. Rudelius for comments on the draft of this paper. The work of N.S. was supported in part by DOE grant DE$-$SC0009988. NS and SHS were also supported by the Simons Collaboration on Ultra-Quantum Matter, which is a grant from the Simons Foundation (651440, NS). Opinions and conclusions expressed here are those of the authors and do not necessarily reflect the views of funding agencies.

# A    Cubic Group and Our Notations

The symmetry group of the cubic lattice (up to translations) is the *cubic group*, which consists of 48 elements. We will focus on the group of orientation-preserving symmetries of the cube, which is isomorphic to the permutation group of four objects $S_4$.

The irreducible representations of $S_4$ are the trivial representation $\mathbf{1}$, the sign representation $\mathbf{1}'$, a two-dimensional irreducible representation $\mathbf{2}$, the standard representation $\mathbf{3}$, and another three-dimensional irreducible representation $\mathbf{3}'$. $\mathbf{3}'$ is the tensor product of the sign representation and the standard representation, $\mathbf{3}' = \mathbf{1}' \otimes \mathbf{3}$.

It is convenient to embed $S_4 \subset SO(3)$ and decompose the known $SO(3)$ irreducible representations in terms of $S_4$ representations. The first few are

$$
\begin{aligned}
SO(3) \quad &\supset \quad S_4 \\
\mathbf{1} \quad &= \quad \mathbf{1} \\
\mathbf{3} \quad &= \quad \mathbf{3} \\
\mathbf{5} \quad &= \quad \mathbf{2} \oplus \mathbf{3}' \\
\mathbf{7} \quad &= \quad \mathbf{1}' \oplus \mathbf{3} \oplus \mathbf{3}' \\
\mathbf{9} \quad &= \quad \mathbf{1} \oplus \mathbf{2} \oplus \mathbf{3} \oplus \mathbf{3}'
\end{aligned}
\tag{A.1}
$$

We will label the components of $S_4$ representations using $SO(3)$ vector indices as follows. The three-dimensional standard representation of $S_4$ carries an $SO(3)$ vector index $i$, or equivalently, an antisymmetric pair of indices $[jk]$.[12] Similarly, the irreducible representations of $S_4$ can be expressed in terms of the following tensors:

$$
\begin{aligned}
\mathbf{1} \quad &: \quad S \\
\mathbf{1'} \quad &: \quad T_{(ijk)} \quad , \quad i \neq j \neq k \\
\mathbf{2} \quad &: \quad B_{[ij]k} \quad , \quad i \neq j \neq k \quad , \quad B_{[ij]k} + B_{[jk]i} + B_{[ki]j} = 0 \\
& \qquad B_{i(jk)} \quad , \quad i \neq j \neq k \quad , \quad B_{i(jk)} + B_{j(ki)} + B_{k(ij)} = 0 \\
\mathbf{3} \quad &: \quad V_i \\
\mathbf{3'} \quad &: \quad E_{ij} \quad , \quad i \neq j \qquad , \quad E_{ij} = E_{ji}
\end{aligned}
\tag{A.2}
$$

In the above we have two different expressions, $B_{[ij]k}$ and $B_{i(jk)}$, for the irreducible representation $\mathbf{2}$ of $S_4$. In the first expression, $B_{[ij]k}$ is the component of $\mathbf{2}$ in the tensor product $\mathbf{3} \otimes \mathbf{3} = \mathbf{1} \oplus \mathbf{2} \oplus \mathbf{3} \oplus \mathbf{3'}$. In the second expression, $B_{i(jk)}$ is the component of $\mathbf{2}$ in the tensor product $\mathbf{3} \otimes \mathbf{3'} = \mathbf{1'} \oplus \mathbf{2} \oplus \mathbf{3} \oplus \mathbf{3'}$. The two bases of tensors are related as[13]

$$
\begin{aligned}
B_{i(jk)} &= B_{[ij]k} + B_{[ik]j} \,, \\
B_{[ij]k} &= \frac{1}{3} \left( B_{i(jk)} - B_{j(ik)} \right) .
\end{aligned}
\tag{A.3}
$$

In most of this paper, the indices $i, j, k$ in every expression are not equal, $i \neq j \neq k$ (see (A.2) for example). Equivalently, components of a tensor with repeated indices are set to be zero, e.g. $E_{ii} = 0$ and $B_{ijj} = 0$ (no sum). The indices $i, j, k$ can be freely lowered or raised. Repeated indices in an expression are summed over unless otherwise stated. For example, $E_{ij}E^{ij} = 2E_{xy}^2 + 2E_{yz}^2 + 2E_{xz}^2$. As in this expression, we will often use $x, y, z$ both as coordinates and as the indices of a tensor.

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
