# Peer review of "Exotic $U(1)$ Symmetries, Duality, and Fractons in 3+1-Dimensional Quantum Field Theory"

_SciPost Physics_

## Round 2 · Referee Report · Anonymous (Referee 1) · 2020-8-3

Report

Fracton phases of matter have attracted significant attention in recent years due to their unconventional properties including subsystem symmetries, mobility restricted excitations, and subextensive ground state degeneracies. The strong dependence of fracton models on features of the lattice challenges the conventional wisdom that the continuum limit of condensed matter systems should admit a quantum field theory description. At the same time, higher rank U(1) gauge theories have been formulated that exhibit dipole (and higher moment) conservation laws, similar to fractonic charge mobility constraints. In this work (the second installment of a trilogy) the authors formulate a pair of 3+1 dimensional U(1) lattice theories with subsystem symmetries and describe their continuum limit. They also describe a pair of gauge theories, related to the aforementioned theories via a gauging duality, that support fracton and lineon defects respectively.

Although quite long, I found the paper to be clearly written and presented. In particular the role of various symmetries, including lattice and subsystem symmetries, in the physics of the models introduced is made clear. The separation of this paper form the others in the series also makes sense, given that a lot of content is covered in this installment and the focus on U(1) theories in 3+1D comes across as being quite natural. I am happy to recommend this paper for publication. I have collected a few minor comments and questions upon reading through the paper, see below.

In the intro it would be nice to see some explanation of the motivation/ anticipated connection of the field theories to X-cube. Quite a few references are included and amongst them the connection of similar lattice U(1) theories to X-cube via Higgsing is explored. Likely this is explained in greater depth in the follow up paper about the Z_N fracton gauge theories, but it could be beneficial to explain that motivation here.

Could the authors comment on how their multipole symmetry relates to the concept of a multipole algebra that has appeared in literature, particularly ref.14?

How important actually is cubic lattice rotation symmetry for fracton field theories? Many fracton models, particularly the most interesting type-II variety, do not have this symmetry. Is it mainly used here as the goal is eventually a field theory description of the X-cube model, which does have this symmetry?

On page 20 the authors remark about the winding dipole symmetry: “Note that this symmetry is not present on the lattice.”
Could they comment explicitly here about the remnants of this symmetry on the lattice system, and how it disappears with cutoff. I think this may have been commented upon elsewhere but it would be good to reinforce it here.

Related comment on page 39 “These fluxes correspond to observables that are one on the lattice” the magnetic fluxes in the two gauge theories, which generate the magnetic symmetries, appear as constraints on the lattice operators. Do the magnetic symmetries, which are said to be absent on the lattice, simply correspond to these constraints?

Page 42 “a single particle cannot move in space - it is immobile - because of gauge Invariance” could the authors be a little more explicit about what immobile means here. Is it that there is no process that moves the single particle to a translated version of itself at another position in space? It seems it would be possible to ``move” the charge by creating several more charges (as is usual for fractons).

Could the authors clarify that the transition function g introduced on page 38 is for the gauge parameter \alpha?

Are the gauge theories introduced stable to confinement induced by instanton/monopole proliferation? Ref.16 suggested the U(1) version of X-cube may be unstable.

It is mentioned that the gauge theories are related to those studied by Slagle and Kim in Ref.7. I am also aware of a follow up work by Slagle and collaborators: arXiv:1812.01613. Is it known how the present work relates to that follow up? In particular there, and in other works, a structure known as foliation is employed. Is any relation of such foliation structures and the discontinuous field configurations in single variables (which seem to be a key new ingredient here) known?

The pair of gauge theories in this paper appear quite related to the X-cube model formulated in different bases. Upon higgsing to Z_N it seems the two gauge theories would be related/equivalent formulations of the same (continuum) X-cube model? Are they already related in some analogous way before Higgsing? Or is this relation only made possible since the dual of Z_N is Z_N (as opposed to the dual of U(1) being Z rather than U(1)).

Typos:
-the position of mu_0 in dispersion eq.1.2
-“An example realizes the” page 12
-“we will repeatedly a large gauge” page 35
-“minimumq” page 45
-“in terms of and” page 49
  • validity: high
  • significance: good
  • originality: good
  • clarity: high
  • formatting: good
  • grammar: excellent

Author:  Shu-Heng Shao  on 2020-09-08  [id 951]

(in reply to Report 1 on 2020-08-03)
Category:
answer to question

We thank the referee for the helpful suggestions and comments. We have added extensive discussions on the universality and robustness of the models studied in this paper. These additions are in the introduction, Section 3.6, Section 4.6, Section 5.9, and Section 6.9.

Below are our replies to the referee's comments and the corresponding changes we have made.

(1) "In the intro it would be nice to see some explanation of the motivation/ anticipated connection of the field theories to X-cube."

We have addressed this comment in the second to last paragraph of the introduction.

(2) "Could the authors comment on how their multipole symmetry relates to the concept of a multipole algebra that has appeared in literature, particularly ref.14?"

We have added the second paragraph on p16 to address this comment.

(3) "How important actually is cubic lattice rotation symmetry for fracton field theories? Many fracton models, particularly the most interesting type-II variety, do not have this symmetry. Is it mainly used here as the goal is eventually a field theory description of the X-cube model, which does have this symmetry?"

It is generally helpful to organize the equations and the fields using the global symmetry of the system. The models in our paper have the cubic group symmetry. In some other models with a different global symmetry, we would use that symmetry to organize the presentation.

(4) "On page 20 the authors remark about the winding dipole symmetry: 'Note that this symmetry is not present on the lattice.'
Could they comment explicitly here about the remnants of this symmetry on the lattice system, and how it disappears with cutoff. I think this may have been commented upon elsewhere but it would be good to reinforce it here. Related comment on page 39 'These fluxes correspond to observables that are one on the lattice' the magnetic fluxes in the two gauge theories, which generate the magnetic symmetries, appear as constraints on the lattice operators. Do the magnetic symmetries, which are said to be absent on the lattice, simply correspond to these constraints?"

We have added some discussions above (3.14).

(5) "Page 42 'a single particle cannot move in space - it is immobile - because of gauge Invariance' could the authors be a little more explicit about what immobile means here. Is it that there is no process that moves the single particle to a translated version of itself at another position in space? It seems it would be possible to move? the charge by creating several more charges (as is usual for fractons)."

We have clarified many discussions in Section 5.5 accordingly. In particular, we added a paragraph at the end of Section 5.5 to address the last question above.

(6) "Could the authors clarify that the transition function g introduced on page 38 is for the gauge parameter \alpha?"

The g in (5.14) is the transition function for the gauge fields, not for the gauge parameter \alpha.

(7) "Are the gauge theories introduced stable to confinement induced by instanton/monopole proliferation? Ref.16 suggested the U(1) version of X-cube may be unstable."

We have added extensive discussions related to this question in the new subsections: Section 3.6, Section 4.6, Section 5.9, and Section 6.9.

(8) "It is mentioned that the gauge theories are related to those studied by Slagle and Kim in Ref.7. I am also aware of a follow up work by Slagle and collaborators: arXiv:1812.01613. Is it known how the present work relates to that follow up? In particular there, and in other works, a structure known as foliation is employed. Is any relation of such foliation structures and the discontinuous field configurations in single variables (which seem to be a key new ingredient here) known?"

We have some partial understanding of the connection to these references and some related work in progress, but these are perhaps outside the scope of the current paper and will be left for future investigations.

(9) "The pair of gauge theories in this paper appear quite related to the X-cube model formulated in different bases. Upon higgsing to Z_N it seems the two gauge theories would be related/equivalent formulations of the same (continuum) X-cube model?"

Yes, that's correct. This point is discussed in 2004.06115.

(10) "Are they already related in some analogous way before Higgsing? Or is this relation only made possible since the dual of Z_N is Z_N (as opposed to the dual of U(1) being Z rather than U(1))."

The A and the \hat A theories are definitely not the same before Higgsing. Among other things, they have different global symmetry as discussed in the current paper.

(11) We thank the referee for pointing out the typos. We have fixed them in the latest version.

---

## Round 2 · Referee Report · Anonymous (Referee 2) · 2020-9-3

Strengths

  1. The paper is an exciting contribution that brings together research on fracton phases with QFT, addressing some of the most interesting current questions in fracton physics.

  2. The paper is generally clearly written, with many details provided so that interested readers can reproduce the results for themselves and add a new approach to their toolbox.

Weaknesses

I do not find any significant weaknesses in this paper.

Report

This paper addresses one of the most exciting and interesting questions in current research on fracton phases of matter, namely the extent and nature of the relationship between fracton phases on the one hand, and continuum quantum field theory (or generalizations thereof) on the other. The authors come at this question from the viewpoint of understanding how to extend continuum QFT to accommodate fracton phases, and this paper makes very significant progress toward this goal by constructing and studying a set of continuum theories related to fracton phases. The example theories and their analysis flows from a discussion of exotic symmetries, which provides a clear unifying framework.

The paper is generally clearly written, with many details of the analysis presented so that interested readers can reproduce the results and learn how to apply and generalize similar approaches. This work is having a high impact within the emerging subject of fractons -- and its intersection with QFT -- and I strongly recommend publication. I have only few minor questions and suggested changes/clarifications for the authors to consider.

The paper certainly satisfies the SciPost Physics acceptance criteria. In particular, the paper satisfies at least Expectation #3 by opening a new pathway with much potential for follow-up work (namely, further applications of these and related continuum theories to fracton physics), and also Expectation #4 by providing a novel and synergetic link between different research areas (namely, QFT and the theory of fracton phases).

Requested changes

  1. One question: given a symmetry of the kind considered, is there any general understanding of geometry of the subspace C on which the conserved charges Q(C) can be constructed? From reading the paper it looks like this needs to be analyzed on a case-by-case basis, which is fine, but it would be interesting to know if the authors are aware of a more systematic approach to this question. Depending on what the authors know about this question, they may wish to add some remarks on it to the manuscript -- I leave it up to them whether this should be done.

  2. Just below Eq. 2.27, the authors make the statement that "only the sum of their zero modes is physical." I'm not sure what "zero modes" is referring to here; this should be explained/clarified.

  3. In the first sentence of 3.1, the word "with" is repeated twice in a row.

  4. In 3.4, it may be helpful to explain why the momentum modes have charges that are sums of delta functions. My understanding is that this follows from the fact that these modes can be created by acting with 2pi-periodic exponentials of the phi^i fields, which are invariant under the gauge redundancy of 3.18.

  • validity: top
  • significance: top
  • originality: high
  • clarity: high
  • formatting: excellent
  • grammar: excellent

Author:  Shu-Heng Shao  on 2020-09-08  [id 952]

(in reply to Report 2 on 2020-09-03)
Category:
answer to question

We thank the referee for the comments and suggestions. Below are our replies to the questions.

  1. "One question: given a symmetry of the kind considered, is there any general understanding of geometry of the subspace C on which the conserved charges Q(C) can be constructed? From reading the paper it looks like this needs to be analyzed on a case-by-case basis, which is fine, but it would be interesting to know if the authors are aware of a more systematic approach to this question. Depending on what the authors know about this question, they may wish to add some remarks on it to the manuscript -- I leave it up to them whether this should be done."

We do not know of a general, systematic algorithm to identify the subspace C. In the current paper (and the other two papers in the series), we work it out based on a case-by-case analysis.

  1. "Just below Eq. 2.27, the authors make the statement that "only the sum of their zero modes is physical." I'm not sure what "zero modes" is referring to here; this should be explained/clarified."

We meant the constant modes. We have changed "zero modes" to "constant modes" there.

  1. "In the first sentence of 3.1, the word "with" is repeated twice in a row."

We thank the referee for pointing out this typo. We have fixed it in the latest version.

  1. "In 3.4, it may be helpful to explain why the momentum modes have charges that are sums of delta functions. My understanding is that this follows from the fact that these modes can be created by acting with 2pi-periodic exponentials of the phi^i fields, which are invariant under the gauge redundancy of 3.18."

We have added a sentence above (3.24) to explain this.

---

## Editorial Decision

resubmitted